# Integrated proteogenomic characterization of localized prostate cancer identifies biological insights and subtype-specific therapeutic strategies

Wei Ou[1,6], Xin-Xin Zhang [1,6], Bin Li[2,6], Ying Tuo[3,6], Ren-Xuan Lin[1,6], Peng-Fei Liu[4], Jian-Ping Guo [5], Hio-Cheng Un[1], Ming-Hao Li[1], Jia-Hao Lei[1], Xiao-Jing Gao[4], Fu-Fu Zheng[1], Ling-Wu Chen [1]✉, Ling-Li Long [2]✉ & Zong-Ren Wang [1]✉

Localized prostate cancer (PCa) is highly variable in their response to therapies. Although a fraction of this heterogeneity can be explained by clinical factors or genomic and transcriptomic profiling, the proteomic-based profiling of aggressive PCa remains poorly understood. Here, we profiled the genome, transcriptome, proteome and phosphoproteome of 145 cases of localized PCa in Chinese patients. Proteome-based stratification of localized PCa revealed three subtypes with distinct molecular features: immune subgroup, arachidonic acid metabolic subgroup and sialic acid metabolic subgroup with highest biochemical recurrence (BCR) rates. Further, we nominated NANS protein, a key enzyme in sialic acid synthesis as a potential prognostic biomarker for aggressive PCa and validated in two independent cohorts. Finally, taking advantage of cell-derived orthotopic transplanted mouse models, single-cell RNA sequencing (scRNA-seq) and immunofluorescence analysis, we revealed that targeting NANS can reverse the immunosuppressive microenvironment through restricting the sialoglycan-sialic acid-recognizing immunoglobulin superfamily lectin (Siglec) axis, thereby inhibiting tumor growth of PCa. In sum, we integrate multi-omic data to refine molecular subtyping of localized PCa, and identify NANS as a potential prognostic biomarker and therapeutic option for aggressive PCa.

Prostate cancer (PCa) is the second most common cancer in men and ranks the fifth leading cause of male cancer death worldwide[1]. Although approximately 75% of newly diagnosed patients are in the localized stage with a curable strategy by radical surgery or radiotherapy[2], part of them will develop biochemical recurrence (BCR) after local treatment, reflecting its remarkable intrinsic heterogeneity[3,4]. To this end, risk stratification based on clinicopathological parameters, including T-stage, prostate-specific antigen (PSA) and Gleason score (GS)[2], has been developed to predict the prognosis of PCa after local therapy. With this stratification, patients

[1]Department of Urology, The First Affiliated Hospital, Sun Yat-sen University, Guangzhou 510080, China. [2]Clinical Trials Unit, The First Affiliated Hospital, Sun Yat-sen University, Guangzhou 510080, China. [3]Department of Pathology, The First Affiliated Hospital, Sun Yat-sen University, Guangzhou 510080, China. [4]Shanghai Applied Protein Technology Co., Ltd, Shanghai 201100, China. [5]Institute of Precision Medicine, the First Affiliated Hospital, Sun Yat-sen University, Guangzhou, Guangdong 510275, China. [6]These authors contributed equally: Wei Ou, Xin-Xin Zhang, Bin Li, Ying Tuo, Ren-Xuan Lin. ✉e-mail: chenlwu@mail.sysu.edu.cn; longll@mail.sysu.edu.cn; wangzr27@mail.sysu.edu.cn

are divided into low/intermediate-risk groups, which generally exhibited favorable outcomes, and high-risk group (aggressive PCa), 40–65% of which will develop BCR after local treatment. The achievement of these findings have shown notable advantages of molecular biomarkers, such as the Decipher and Prolaris test, in risk stratification for PCa over clinicopathologic factors[5–7]. Therefore, in-depth characterization of the molecular phenotypes to better unravel the heterogeneous nature of PCa is of great significance for the accurate identification of aggressive PCa.

Recently, with the development of genomic analyses, diverse genomic alterations, including mutations of *SPOP*, *FOXA1* or *ETS* gene fusions have been identified to contribute PCa[8]. More interestingly, these alterations are mainly linked to the androgen signaling pathway[8,9]. Due to the pivotal role of androgen signaling in boosting prostate carcinogenesis and progression[2], targeting androgen signaling via androgen deprivation therapy (ADT) is employed as the first-line systemic treatment of aggressive PCa[10]. However, according to the clinical trial (GETUG-AFU15), despite initially high response rate to ADT, the majority of patients will develop castration resistance inevitably over time, mostly within 2 years after ADT[11,12]. This possibly due to the acquired resistance to anti-androgen therapies, including bicalutamide, goserelin. To this end, multiple combination therapies, including with androgen receptor pathway inhibitors (ARPIs), such as enzalutamide, have been explored in preclinical. Of note, although ARPIs could prolong the progression-free survival of PCa patients, the PSA response rate associated with ARPIs is below 80%, with about 20% of patients exhibiting no response to ARPIs[13]. Thus, there is an urgent need to explore the molecular characteristics of aggressive PCa and develop novel therapeutic targets.

Due to the executor roles of proteins in biological processes, molecular subtyping and therapeutic target screening at the protein level are paramount for the precision cancer treatment. Recently, integrated proteogenomic exploration has played a pivotal role in molecular subtyping and drug development of various tumors[14–17]. To this end, based on the western intermediate-risk patient cohort, proteogenomic studies of PCa have been carried out to categorize PCa into five proteomic subtypes without significant prognostic differences. Compared with PCa patients from western countries, Chinese PCa displayed a higher proportion of high-risk patients, bearing different clinical and genomic characteristics[18–20], but lacking an in-depth illustration for molecular characteristics[21,22], and hindering the identification of novel interventional targets.

Here, to fill this gap, we delivered the multi-omic study in 145 Chinese PCa patients by integrating proteomic data with genomic, transcriptomic and phosphoproteomic data, and successfully divided PCa into immune subtype, arachidonic acid metabolic subtype, and the sialic acid metabolic subtype. We also defined the sialic acid metabolic subtype PCa is tightly associated with the highest BCR rates, with elevated NANS expression and sialic acid accumulation to shape an immunosuppressive microenvironment. Together, this study provides valuable insights into the molecular subtypes and precise target therapy for aggressive PCa.

## Results

### The proteomic subgroups and molecular characteristics of PCa
To illustrate the comprehensive molecular features of PCa, paired tumor and non-tumor prostate tissues sourced from 145 Chinese PCa patients were employed and subjected for multi-omic analysis (Fig. 1A and Fig. S1A). Among them, 69.6% patients were stratified as high-risk according to contemporary clinical risk assessments[2] (Table S1). With the high-quality results from Data Independent Acquisition (DIA)-based LC-MS/MS global proteomics (Fig. S1B), we totally identified 8694 proteins from 145 tumor and 141 adjacent normal samples (Fig. S1C), accompanied with more than 26,000 phosphorylation sites from these tissues (Fig. S1D).

Via an unsupervised clustering analysis based on Non-Negative Matrix Factorization (NMF) method, 145 PCa patients in the discovery cohort were classified into three distinct subgroups at the proteomic levels (Fig. 1B). By contrast, these patients were divided into three distinct subgroups at the transcriptomic level using the NMF method and PAM50 signatures respectively[23], with mild overlap (15–18%) between the transcriptomic and proteomic subtypes of PCa, which was consistent with previous report[22] (Fig. 1C). Of note, two groups were subtyped as metabolism pathways enrichment (termed as metabolism subgroup 1 and 2) and another group displayed immune-related pathways by Gene Sets Enrichment Analysis (GSEA) (Fig. 1D). Interestingly, the global transcriptomic molecular features of three proteomic subgroups were similar to their proteomic molecular features (Fig. S1E). Among them, the metabolism subgroup 2 exhibited the poorest prognosis (Fig. 1E), accompanied with clinical features related to poorer prognosis, such as higher preoperative PSA levels ($P = 0.002$), higher postoperative pathological Gleason score (GS) ($P = 0.028$) and higher proportion of lymphatic metastasis ($P = 0.03$) (Fig. 1F, Fig. S1F). Even taking these clinical factors into account, the proteomic subgroup 2 remained an independent prognostic predictor of PCa (Table S2).

Interestingly, given that AR activation is closely related to the aggravation of PCa, our identified subgroup 2 PCa displayed high correlation with AR activation (Fig. 1G), suggesting activation of AR signaling pathway as a fundamental feature of this subgroup. Simultaneously, metabolism subgroup 1 also displayed features of metabolic enrichment, however, patients in this group exhibited a favorable prognosis and no statistically significant association with AR activation (Fig. S1G). The detailed molecular differences of subgroup 1 and 2 will be further dissected as below.

### Multi-omics profile of the three PCa proteomic subgroups
To gain more insights into the three proteomic subgroups of PCa, we further performed genomic, transcriptomic and phosphoproteomic analysis as we described above (Fig. 1A). Interestingly, metabolism-associated subgroup 1 and 2 exhibited significantly higher tumor mutational burden (TMB) and somatic copy number aberration (SCNA) burden (Fig. 2A) compared to subgroup 3, suggesting an association between increased genomic instability and worse prognosis. Further gene mutation analysis showed that, *FOXA1*, an established AR pioneer transcription factor to enhance AR transcriptional activation[24], displayed a significantly higher prevalence in metabolism subgroup 2 ($P = 0.015$) (Fig. 2B). Moreover, patients in metabolism subgroup 2 exhibited a higher prevalence of tumor suppressor gene deletions (such as *RB1*, *CHD1* and *ZNF292*) and AR activation-related amplification (*GATA2*) and deletions (*MAP3K7*, *SPOPL*) (Fig. 2C). While, gene fusion analysis revealed that patients harboring the *TMPRSS2-ERG* (T-E) gene fusion, involved in lipid metabolism functions in PCa[22], belonging to the metabolism subgroup 1 (Fig. 2C). Kinase-Substrate Enrichment Analysis (KSEA) illustrated that cell cycle regulation-related kinases (UHMK1, CHEK1) and the AR pathway's target kinase CAMKK2[25] were significantly enriched in the metabolism subgroup 2 (Fig. 2D). Meanwhile, the kinase PRKD3 associated with PCa progression was significantly enriched in subgroup 1[26] (Fig. S2A), in consistent with the finding that expression of PRKD3 contributes to mast cell infiltration and angiogenesis in the tumor microenvironment of PCa[26]. By contrast, the atypical inflammatory kinase IKBKE was significantly enriched in the immune subgroup 3[27,28] (Fig. S2B), which has been reported to promote PCa tumor growth through modulating the Hippo pathway[29]. Therefore, the molecular features in the phosphoproteomic level are potentially associated with its proteomic subgroups.

For the immune subgroup, we described the immune-related characteristics of subgroup 3 by performing the single-sample GSEA (ssGSEA) analysis of typical immune gene sets. Compared with

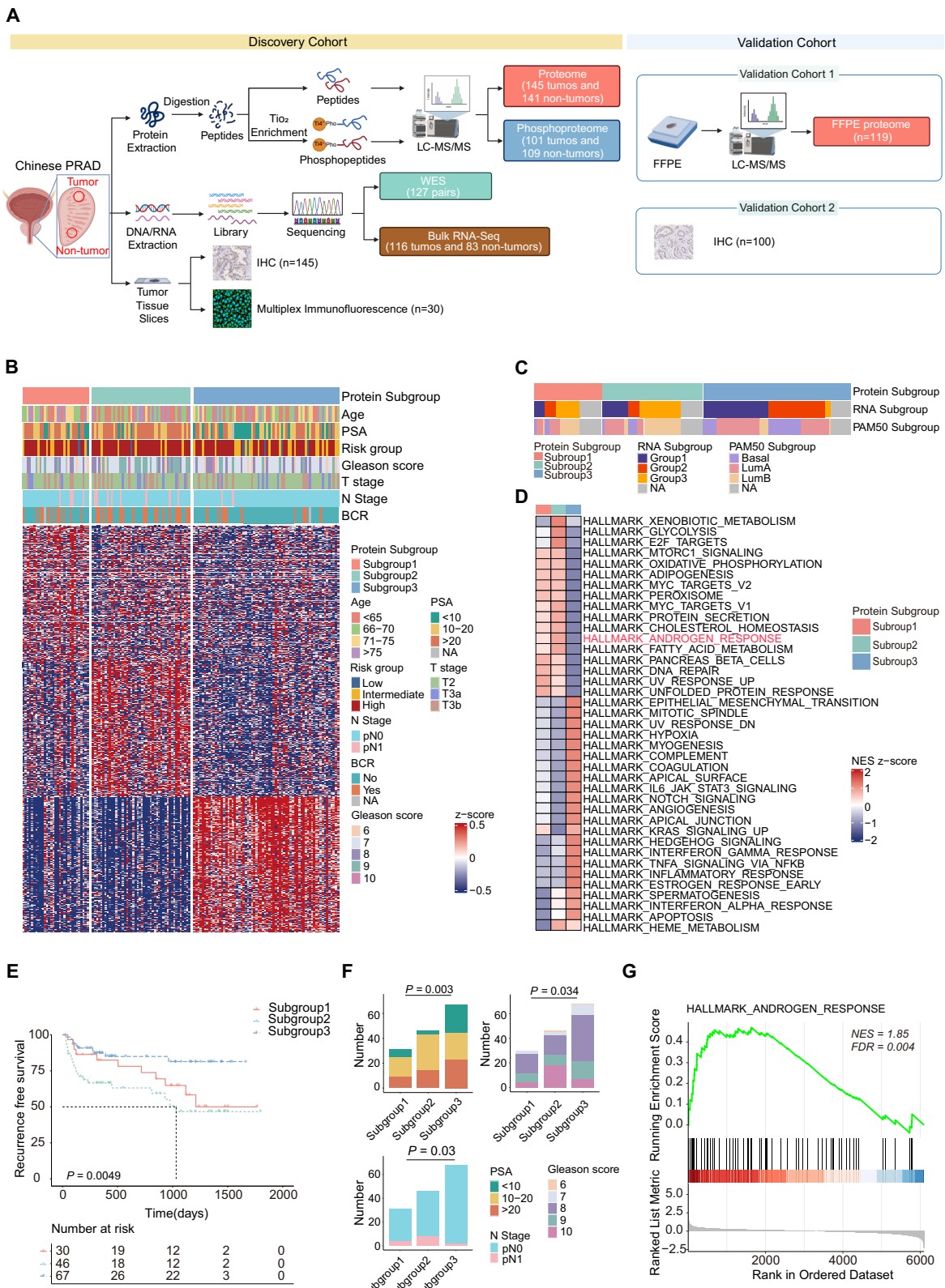

metabolism subgroup 1 and 2, the immune subgroup exhibited a more 'hot' overall immune profile, with significant upregulation of immune activation pathways such as pro-inflammation and lymphocyte infiltration (Fig. S2C). MCP-counter analysis further indicated higher effector immune cell (CD8⁺ T, CTLs, NK cells) infiltration in the immune subgroup than other two subgroups (Fig. S2D).

For the metabolism subgroup 1, GSEA analysis was used to show the details of metabolic features, and lipid metabolism pathways were significantly enriched in subgroup 1 (Fig. S2E). Notably, pathways associated with polyunsaturated fatty acid metabolism, including linoleic acid and arachidonic acid, were significantly enriched in this subgroup (Fig. S2E). The expression of key enzymes for the linoleic acid and arachidonic acid pathways were analyzed among subgroups.

**Fig. 1 | Proteomic subgroups and molecular characteristics of PCa. A** Study schematic. The left panel shows the multi-omic experimental design of the discovery cohort. The right panel shows the type of validation experiments, including formalin fixed paraffin-embedded (FFPE) proteome and immunohistochemistry (IHC) in two validation cohorts. LC-MS/MS represents liquid chromatography tandem mass spectrometry. WES represents whole exome sequencing, IF represents immunofluorescence. Created with BioRender.com, released under a Creative Commons Attribution-NonCommercial-NoDerivs 4.0 International license. **B** Heatmap visualizes the three proteomic subgroups of PCa based on the non-negative matrix factorization (NMF) unsupervised clustering algorithm and their corresponding clinical features in the discovery cohort. PSA represents prostate-specific antigen. BCR represents biochemical recurrence. NA represents not available. **C** Comparison of the proteomic subtyping (NMF-based) and two kinds of transcriptomic subtyping (NMF-based and PAM50-based) in the discovery cohort. NA represents not available. **D** Gene sets enrichment analysis (GSEA) showing distinct molecular characteristics among three proteomic subgroups of the discovery cohort (one subgroup vs the union of other two subgroups). **E** Kaplan-Meier curves showing the biochemical recurrence-free survival (bRFS) of patients in three proteomic subgroups of the discovery cohort. *P* value is calculated by Log-rank test. **F** Bar plots comparing the clinicopathological features among three proteomic subgroups of the discovery cohort, including prostate-specific antigen (PSA), Gleason score (GS) and lymph node metastasis (N stage). *P* values are determined using one-way ANOVA test. **G** GSEA showing the activation of the androgen response pathway in subgroup 2 compared to the union of subgroup 1 and 3 of the discovery cohort. Source data are provided as a Source Data file.

The results showed that several enzymes involved in the arachidonic acid metabolism, such as ALOX15, a key enzyme in catabolism of arachidonic acid[30], were significantly elevated in subgroup 1 (Fig. S2 F-G).

In summary, multi-omics analysis explained that the AR pathway was activated in subgroup 2 patients, which has the worst prognosis comparing with other subgroups. While, subgroup 1 displayed higher lipid metabolism, and subgroup 3 exhibited an immune hot feature. Thus, these findings further confirmed the accuracy of the protein-level subtyping in the discovery cohort.

## Molecular features of subgroup 2 associated with the worst prognosis

To further investigate the molecular features of subgroup 2, GSEA analysis highlighted that multiple sugar and glycogen metabolism-related pathways, especially the amino sugar and nucleotide sugar metabolism pathway, were noteworthy enriched in this subgroup (Fig. 3A). Furthermore, key enzymes' expression analysis of this pathway was performed. The result showed that a conspicuous upregulation of several sialic acid synthesis-related enzymes, especially UAP1, NPL and NANS, in metabolism subgroup 2 (Fig. 3B). Furthermore, we performed the metabolomic analysis, and observed the significant upregulation of N-Acetylneuraminic acid, the end product of the sialic acid metabolism pathway, in subgroup 2 (Fig. 3C). To verify our findings, we firstly constructed a 39-protein subtyping panel, which was filtered by the differential expression analysis and the Random Forest algorithm in the discovery cohort (Fig. S3A-C). Then, we additionally employed 119 PCa patients from other centers (details provided in the Methods section) as the validation cohort 1. Based on the proteomic profiling data and the 39-protein subtyping panel, the validation cohort 1 could be successfully categorized into three proteomic subgroups consistent with those identified in the discovery cohort (Fig. 3D, S3A). Importantly, proteomic subgroup 2 was also validated as higher sialic acid synthesis metabolism with the poorest prognosis in the validation cohort 1 (Fig. 3D-E). Together, these findings suggest that metabolic subgroup 2 exhibits elevated sialic acid synthesis.

Due to the pivotal role of NANS in the sialic acid synthesis pathway[31–34], and its expression is relatively high in subgroup 2 PCa, we thus mainly focused on the study of NANS in this process. To this end, our proteomic data firstly showed that NANS expression was significantly elevated in tumor tissues compared to paired adjacent normal tissues in the discovery cohort, especially in subgroup 2 (Fig. 3F, Fig. S3D). Besides, immunohistochemistry (IHC) staining in the discovery cohort corroborated the higher NANS expression in tumor tissues of subgroup 2 compared to subgroup 1 and 3, which has been further validated by proteomic data of the validation cohort 1 (Fig. 3G-H). Importantly, survival analysis established a noteworthy correlation between higher NANS expression and a reduced duration of biochemical recurrence-free survival (P = 0.0054) (Fig. 3I, Table S3). To further verify the function of NANS as a prognostic indicator, an additional 100 PCa patients from another center were recruited as the

validation cohort 2 (details provided in the Methods section) and conducted the IHC staining of NANS. Survival analysis in two validation cohorts both confirmed that higher NANS expression was associated with poor prognosis of PCa (Fig. 3J-K).

In summary, our findings highlight that metabolism subgroup 2 is characterized by a substantial enhancement in sialic acid synthesis. Notably, NANS is elevated in tumor tissues of this subgroup and significantly associated with patients' poor prognosis.

## NANS induces sialylation-driven immune suppression in subgroup 2

Sialic acid-modified (sialylation) residues play a widespread role in the formation of various glycoprotein chains on cell surface, thereby readily influencing intercellular communications[32,35]. To understand whether abnormalities in sialic acid synthesis contribute to the changes in tumor sialylation in subgroup 2, IHC staining of α2,3 and α2,6-sialylation in tumor tissues was performed in the discovery cohort. The results showed markedly increased sialylations in tumors of subgroup 2 compared to those in other two subgroups (Fig. 4A). To further explore the potential role of NANS in the regulation of sialic acid anabolism and sialylation of PCa, we employed a panel of 8 PCa cell lines, which have been initially stratified into three proteomic subtypes by using the nearest template prediction (NTP) algorithm as we have done in human PCa tissues (Fig. 4B). As a result, the human-derived 22Rv1 and murine Myc-CaP PCa cells were both classified into the aggressive subgroup 2. Besides, the western blot results confirmed the higher expression of NANS in subgroup 2 cell lines of PCa (Fig. S4A). Furthermore, we employed 22Rv1 and Myc-CaP cells for in vitro experiments to exploit the role of NANS-sialic acid in PCa. *NANS*-knockout PCa cells were constructed based on the corresponding single guide RNA (sgRNA) (Fig. 4C-D). The results showed that *NANS* depletion largely diminished sialic acid synthesis measured by targeted metabolomics (Fig. 4E), accompanied by decreased sialylation level of tumor cells (Fig. 4F).

Next, we evaluated whether NANS promotes tumor development through sialylation, and observed that knockout of *NANS* only mildly influenced the proliferation of PCa cells in vitro measured with CCK8 and colony-forming assays (Fig. S4B-C), indicating the essential involvement of other components in tumor microenvironment. Recent research has suggested the sialoglycan-sialic acid-binding immunoglobulin-type lectins (Siglecs) axis as a immune checkpoint which can be targeted to augment the anti-tumor immune response[32,36]. To dissect the immune microenvironment of PCa, the EcoTyper analysis was performed in the discovery cohort and revealed that increased infiltration of macrophages in subgroup 2 (Fig. 4G). In line with this finding, multiplex immunofluorescence (mIF) staining in the discovery cohort further confirmed the immunosuppressive microenvironment with increased infiltration of M2 macrophages and decreased infiltration of CD8+ T cells in the subgroup 2 (Fig. 4H). These results suggest that PCa patients in subgroup 2 exhibit a sialylation-associated immunosuppressive microenvironment. Besides, IHC

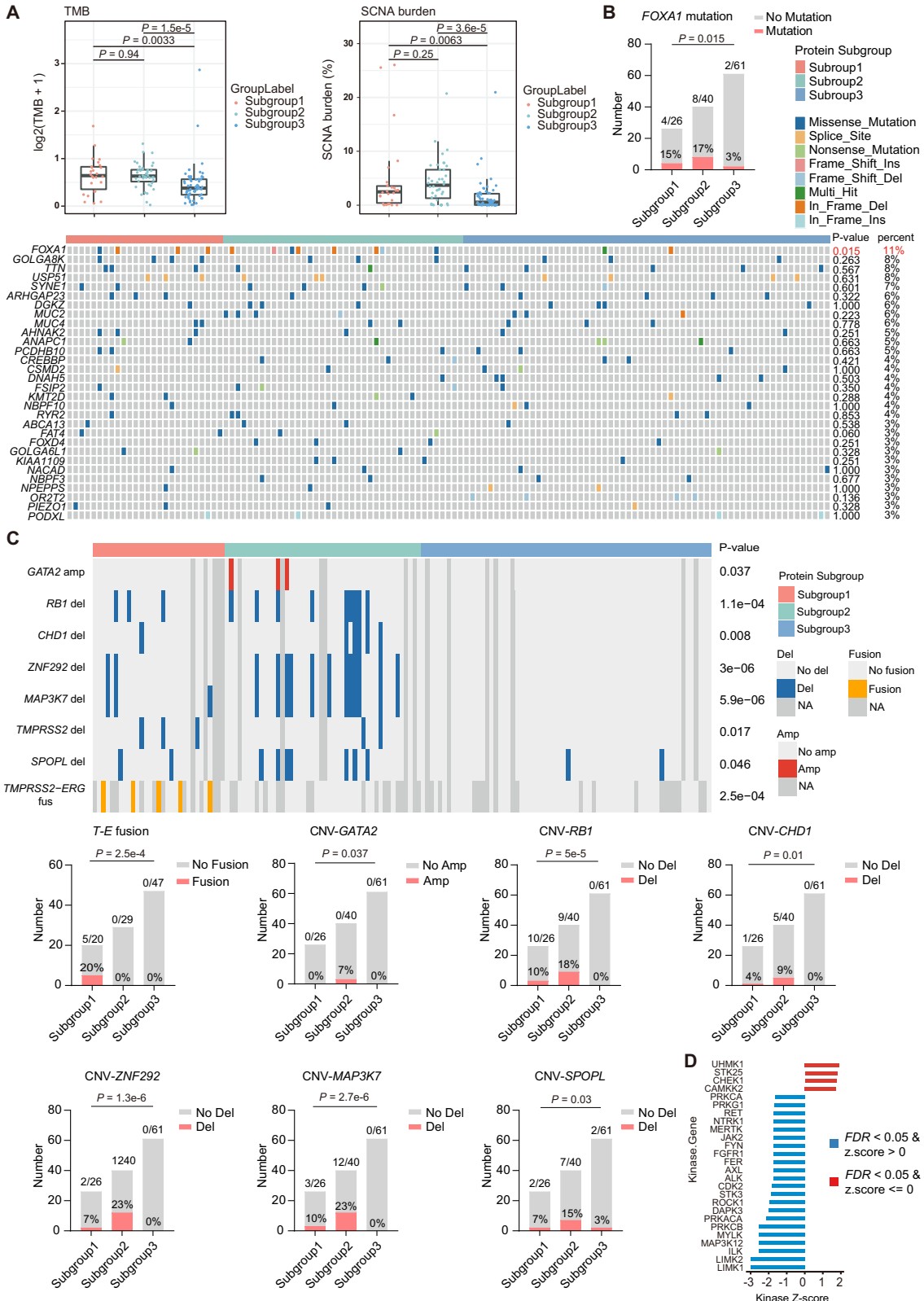

staining also confirmed the enhanced immune activitiy of tumors in the immune subgroup 3 compared to subgroup 1 (Fig. S4D).

### NANS depletion reverses PCa immunosuppressive micro-environment and results in tumor growth suppression

In comparison with an immune hot status of subgroup 3, both sub-group 1 and 2 displayed a relative cold microenvironment. Due to the

potential role of sialylation in immune modulation, we hypothesized that NANS-sialylatic acid would modulate immune response. To this end, we constructed a syngeneic PCa orthotopic transplanted mouse model with Myc-CaP cells in absent/present of *Nans* (Fig. S5A). Four weeks after tumor cell inoculation, the tumor tissues from two groups were collected and subjected to the proteomic profiling and the single-cell RNA sequencing (scRNA-seq). The scRNA-seq analysis unveiled 10

**Fig. 2 | Multi-omics profile of three PCa subgroups. A** Comparison of tumor mutational burden (TMB) and somatic copy number aberration (SCNA) burden among three proteomic subgroups of the discovery cohort ($n = 26/40/61$ for subgroup 1/2/3). The middle lines represent the median. Lower/upper hinges denote 25–75% IQR with whiskers extending to 1.5 IQR. P-values are determined using two-tailed Student's t test. **B** Bar plot comparing the *FOXA1* mutation frequency among three proteomic subgroups of the discovery cohort (top panel). Numbers at the top of the bar represent counts of each subgroup, with corresponding proportions at the bottom of the bar. Prevalence of top 30 recurrent mutation genes in three proteomic subgroups of the discovery cohort (bottom panel). *P* values are determined using one-way ANOVA test. **C** Prevalence of specific SCNAs and gene fusions in three proteomic subgroups of the discovery cohort (top panel). NA represents not available. Bar plots comparing frequency of specific SCNAs and gene fusions among three proteomic subgroups of the discovery cohort (bottom panel). Numbers at the top of the bar represent counts of each subgroup, with corresponding proportions at the bottom of the bar. *P* values are determined using one-way ANOVA test. **D** Kinase-substrate enrichment analysis (KSEA) showing significantly up-regulated and down-regulated kinases in subgroup 2 patients of the discovery cohort (subgroup 2 vs subgroup 1 and subgroup 3 combined). Red bars refer to FDR < 0.05 and z score > 0, blue bars refer to FDR < 0.05 and z score ≤ 0. Source data are provided as a Source Data file.

distinct cell types within the mouse tumor tissues (Fig. 5A, Fig. S5B-C). We identified 4 major subclusters of epithelial cells, including the Luminal C1-3 subclusters with high signal for luminal signature genes and the Cell Cycle subcluster with high signal for cell cycle-related genes (Fig. 5B, Fig. S5D-E). After *Nans* depletion, we observed a significant reduction of Luminal C1 epithelial cells, which displayed the activation of both malignant progression-related pathways and sialic acid-related pathways[37] ($p < 0.0001$, Fig. 5C-D). Meanwhile, a notable reduction of macrophages was also observed in the *Nans*-knockout group (p < 0.0001, Fig. 5E). We further identified four subclusters of macrophages, only the proportion of Lyve1[+] macrophages underwent a conspicuous decrease after *Nans* depletion ($p < 0.0001$, Fig. 5F-G, Fig. S5F-G). Notably, Lyve1[+] macrophages exhibited the highest expression of sialic acid-binding immunoglobulin-like lectin 1 (Siglec1) among all cell types (Fig. 5H). Therefore, we hypothesize that Lyve1[+] macrophages might be the major cell type affected by alterations of sialylation. Functional enrichment analysis indicated that Lyve1[+] macrophages predominantly manifested a M2-like phenotype and immunosuppression status (Fig. 5I). Consequently, we propose that *Nans* depletion, decreasing the sialylation level of tumor cells, play a contributory role in reversing the immunosuppressive microenvironment by impeding the infiltration of Lyve1[+] macrophages. The further IF results confirmed that depletion of *Nans* significantly alleviated the sialylation level of tumors, decreased M2 macrophages infiltration and increased CD8[+] T cells infiltration in tumor tissues (Fig. 5J). In line with these findings, *Nans* deficiency resulted in delayed tumor growth (Fig. 5K), decreased tumor volume and weights (Fig. 5L) in orthotopic transplanted mouse model. To further validate our findings, we next constructed the *Pten/Trp53* double knockout ($Pten^{PC-/-}$; $Trp53^{PC-/-}$) mouse model, which partially recapitulates features of subgroup 2 PCa confirmed by the proteomic subtyping results (Fig. S6A). After administration of adeno-associated virus (AAV) based *Nans* depletion, we also observed the reduction of both sialylation level and M2 macrophages infiltration in mouse tumor tissues (Fig. S6B-C). In addition, tumor volume and weights were significantly decreased in $Pten^{PC-/-}$; $Trp53^{PC-/-}$ mouse model after *Nans* depletion (Fig. S6D). These results in two diverse PCa mouse models together indicate that elevated NANS in aggressive PCa could promote sialylatic acid synthesis and further sialylation to remodel tumor immunoenvironment, resulting in PCa progression.

## Discussion

Localized PCa is highly variable in their response to therapies, but only a fraction of this heterogeneity can be explained with the clinical available prognostic factors[2]. Therefore, a rigorous understanding of the molecular factors that drive aggressivity and heterogeneity of PCa is essential for its precisive interventions. Thus, the genomic and transcriptomic subtypes of localized PCa have been established. For example, seven localized PCa subgroups have been genomically defined by fusions of ETS family genes such as *ERG, ETV, ETV4* or *FLI1*, or mutations in *SPOP, FOXA1* or *IDH1* from the Cancer Genome Atlas (TCGA) database. While, 26% of PCa still could not be classified with these genomic alterations[8]. Meanwhile, the Prolaris test (with 31 RNA

biomarkers) and the Decipher classifier (with 22 RNA biomarkers) have both developed to benefit PCa risk stratification over clinicopathologic factors[5,6,38]. Moreover, recent work based on the western population has divided intermediate-risk PCa into five subtypes, the prognostic differences among them were not significant[22]. To overcome these challenges, here we performed proteogenomic approach (including WES, bulk-RNAseq, proteomics and phosphoproteomics analyses) characterization of 145 Chinese PCa with a relative higher portion of aggressive types. In the current study, we classified PCa into three distinct subgroups associated with clinical and molecular characteristics based on the proteomic profiling, including two distinct metabolic-associated subgroups and an immune-hot feature subgroup. Of note, the sialic acid-enriched metabolic subgroup represents the poorest prognosis and the most invasive clinical features, accompanied by AR signaling activity and NANS-sialic acid-induced immunosuppression, thereby providing a valuable resource for further discovery and precise oncological practice in aggressive PCa.

Among our defined subgroups, subgroup 3 is characterized as the immune subgroup with the best prognosis. This subgroup is determined as 'immune hot' with higher infiltration of immune effector cells, such as CD8[+] T and NK cells. Since increased tumor-infiltrating lymphocytes (TILs) have been reported as a positive prognostic indicator in PCa[39], patients in subgroup 3 display a better prognosis. While subgroup 1 and 2 both display a relative 'immune cold' status, and have been defined as metabolic features, as arachidonic acid metabolic and sialic acid metabolic subgroup, respectively, with relatively poor prognosis compared to subgroup 3. In subgroup 1, the patients exhibit accumulated arachidonic acid, accompanied with elevated expression of lipoxygenase 15 (ALOX15) rather than cyclooxygenase 2 (COX-2), two enzymes catabolizing arachidonic acid[30,40]. Consistent with this finding, clinical trials of the selective COX-2 inhibitor celecoxib failed to improve outcomes in high-risk prostate cancer[40–42]. Instead, we speculate that ALOX15 may be the potential target of PCa in subgroup 1.

Protein kinases and phosphatases play crucial roles in the reversible phosphorylation-dephosphorylation cycle, regulating tumor development. The progression of PCa may be attributed to abnormal activation or overexpression of kinases. For instance, excessive activation of pathways like the mitogen-activated protein kinase (MAPK) and the nuclear factor-κB (NF-κB) pathway promotes the development and progression of PCa[43,44]. Recently, PIM kinases have also been identified as key oncogenic drivers in PCa[45]. Increasingly, inhibitors targeting kinases are being applied in the treatment of advanced PCa patients[46]. In the present study, the enrichment of AR-related kinases and immune-related kinases in different subgroups respectively correlates with their proteomic features, potentially adding in the risk stratification of PCa patients.

While, patients in subgroup 2 are significantly associated with the poorest prognosis with sialic acid accumulation and relatively higher activity of AR signaling pathway. Sialic acids could attach to terminals of glycoproteins and glycolipids on the cell surface[33], and up-regulation of sialic acid-containing glycans, termed hypersialylation, is a common feature of human malignancies possibly by driving tumor

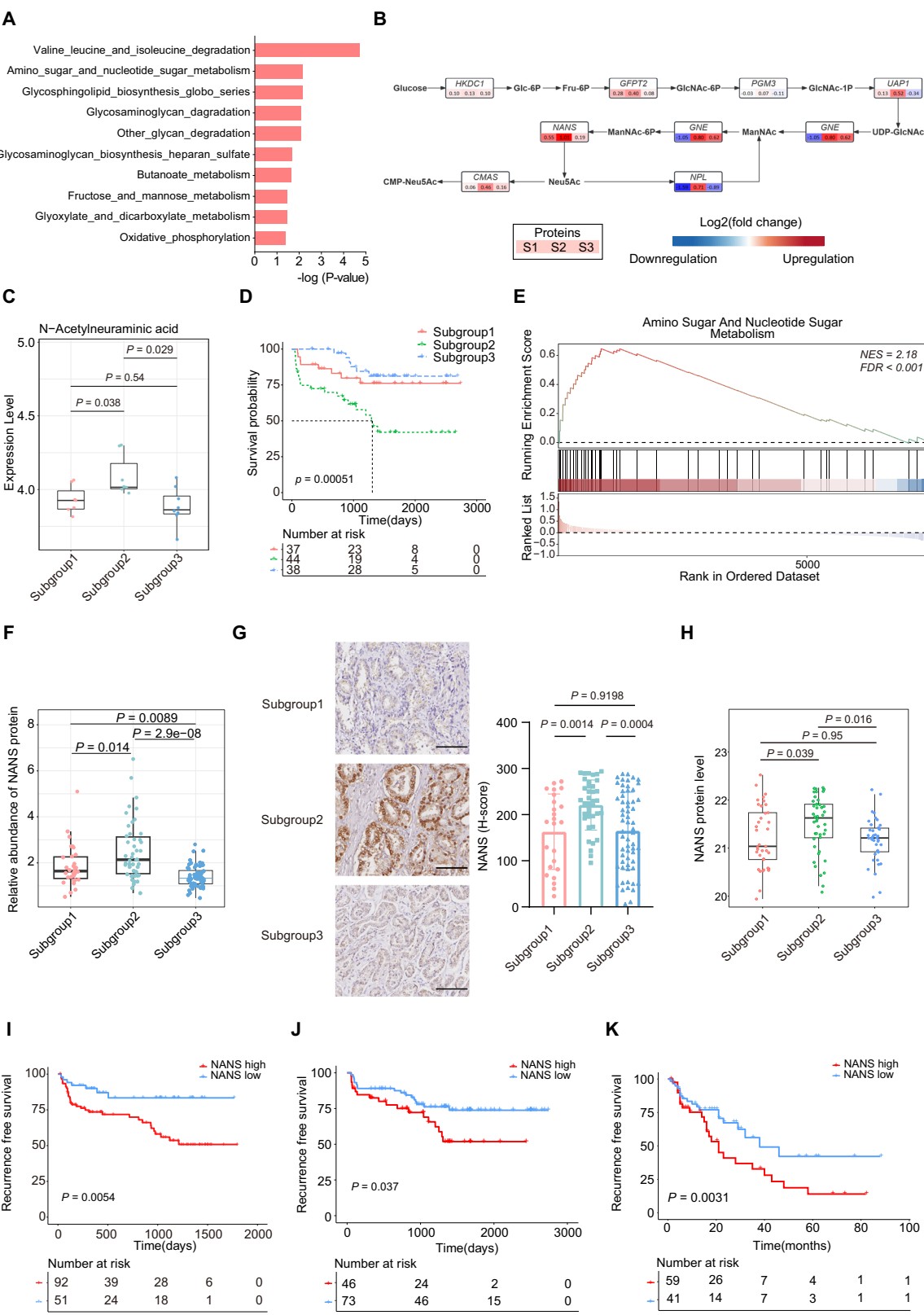

immune evasion through interactions with Siglecs on tumor-infiltrating immune cells[47–49]. In echoing this finding, we found that this type of PCa displays a 'cold immune' environment, coupled with increased hypersialylation and decreased tumor-infiltrating immune cells. Thus, conversion of the immunosuppressive environment by modulating sialic acid accumulation, would benefit the immunotherapies.

By integrated proteomic and single-cell RNA sequencing approach, we observed that the sialic acid synthesis pathways, including related enzymes, such as UAP1, NPL and NANS all showed a relatively high expression in the aggressive subgroup, coupled with the accumulation of metabolic intermediators, with the most high levels of sialic acids. NANS is the key enzyme of the last step for sialic acid synthesis[50]. Targeting NANS can directly block the sialic acid synthesis

**Fig. 3 | Higher sialic acid metabolism and NANS expression in subgroup 2 tumors associated with poor prognosis. A** GSEA showing top 10 up-regulated metabolic pathways in subgroup 2 of the discovery cohort (subgroup 2 vs subgroup 1 and 3). *P* values are determined using two-sided Fisher's exact test. **B** Differential expression of key enzymes in the sialic acid synthesis pathway among three subgroups of the discovery cohort. **C** Comparison of N-Acetylneuraminic acid abundance among three subgroups of the discovery cohort based on metabolomics (n = 7/7/8 for subgroup 1/2/3). The middle lines represent the median. Lower/upper hinges denote 25-75% IQR with whiskers extending to 1.5 IQR. *P* values are determined using two-sided Wilcoxon rank-sum test. **D** Kaplan-Meier curves showing bRFS of patients in three subgroups of the validation cohort 1. *P* value is calculated by Log-rank test. **E** GSEA showing the up-regulation of the amino sugar and nucleotide sugar metabolism pathway in subgroup 2 of the validation cohort 1 (subgroup 2 vs subgroup 1 and 3). **F** Comparison of NANS protein abundance among tumors of three subgroups in the discovery cohort (*n* = 31/46/68 for

subgroup 1/2/3). The middle lines represent the median. Lower/upper hinges denote 25-75% IQR with whiskers extending to 1.5 IQR. P-value is determined using two-sided Wilcoxon rank-sum test. **G** Immunohistochemistry staining for NANS in tumors of three subgroups in the discovery cohort (n = 37/44/38 for subgroup 1/2/3). Scale bar represents 100um. Data are presented as mean ± standard deviation (SD). *P* values are determined using two-tailed Student's t test. **H** Comparison of NANS proteomic abundance among tumors of three subgroups in the validation cohort 1 (*n* = 37/44/38 for subgroup 1/2/3). The middle lines represent the median. Lower/upper hinges denote 25-75% IQR with whiskers extending to 1.5 IQR. *P* values are determined using two-sided Wilcoxon rank-sum test. **I** Kaplan-Meier curves showing bRFS of patients in the discovery cohort. *P* value is calculated by Log-rank test. (**J**)Kaplan-Meier curves showing bRFS of patients in the validation cohort 1. P-value is calculated by Log-rank test. **K** Kaplan–Meier curves showing bRFS of patients in the validation cohort 2 grouped by immunostaining intensity of NANS. *P* value is calculated by Log-rank test. Source data are provided as a Source Data file.

with fewer side effects. Therefore, due to the pivot role of NANS in sialic acid synthesis, we chose to mainly focus on the function of NANS in mediating PCa aggressive phenotypes. However, whether other enzymes and accumulation of according metabolic products in this pathway also play important roles in PCa aggressive phenomenon and immune response deserves further investigation. Another interesting finding is that aggressive subgroup 2 also exhibits a higher activity of AR signaling compared with other groups. Therefore, whether the upregulation of sialic acid synthesis-related enzymes is regulated by AR, or accumulated sialic acid could directly or indirectly activate AR to promote the aggressive phenotype of PCa are worth to be elucidated. That could possibly provide the underlying mechanism and alteration intervention for the resistance of castration treatment.

Targeting NANS in vivo can reverse the immunosuppressive microenvironment of PCa and halt tumor growth. Similar findings on sialoglycan-Siglec immune axis between tumor cells and TAMs have also been reported in other malignancies[37,48]. As a result, desialylation represents an effective therapeutic approach to reshape tumor immune microenvironment and augment the antitumor immunity[51]. For example, in pre-clinical models of melanoma, intratumoral injection with a sialic acid mimetic blocks tumor sialic acid synthesis and suppress tumor growth[52]. Similarly, in the current study, depletion of *NANS* can markedly decrease sialylation level and inhibit tumor growth of PCa. Therefore, sialic acid-targeting avenue presents a promising anti-tumor therapy in PCa. However, whether accumulated sialic acid would play other functions beyond its role in mediating sialylation and repressing immunosuppressive environment is worth for exploration.

In conclusion, our study provides a comprehensive analysis of Chinese PCa by using multi-omic platforms. We identify three distinct proteomic subtypes of PCa with a subgroup of aggressive PCa characterized by higher sialic acid metabolism and poorest clinical outcomes. These findings highlight the strategy of targeting NANS-sialic acid signaling to modulate antitumor immunity through sialoglycan-Siglec axis for the precision therapy of aggressive PCa.

## Methods
### Study approval
The use of pathological specimens, as well as the review of all pertinent patient records, was approved by the Research Ethics Committee of the First Affiliated Hospital of Sun Yat-sen University ([2022] 101) and to the Helsinki Declaration of the World Medical Association. Informed consent was obtained from patients. The research adhered to established ethical guidelines and included exclusively male patients. All mice procedures were conducted in accordance with the Institutional Animal Ethics Committee of the First Affiliated Hospital of Sun Yat-sen University ([2023] 238).

### Patients and Clinical Samples
For the discovery cohort, paired tumor, adjacent non-tumor prostate tissues from 145 Chinese PCa patients were initially enrolled. All the treatment-naïve patients underwent radical prostatectomy (RadP) surgeries from May 2017 to October 2021 at the First Affiliated Hospital of Sun Yat-sen University. Tissue samples were collected within 30 minutes after operation and snap-frozen in liquid nitrogen.

For the validation cohort 1, a total of 119 Chinese men with localized PCa were enrolled. All the treatment-naïve patients underwent RadP surgeries from May 2017 to October 2021 at the Third and Seventh Affiliated Hospital of Sun Yat-sen University, and Guangzhou First People's Hospital. The validation cohort 1 is used for validation of proteomic subtyping and immunohistochemical (IHC) stainings of NANS.

For the validation cohort 2, a total of 100 Chinese men with localized PCa were enrolled. All the treatment-naïve patients underwent RadP surgeries from January 2015 to October 2018 at the Daping Hospital, Army Medical University. The validation cohort 2 is used for IHC stainings of NANS.

Adjacent 10 serial sections (5 mm thickness of each section) from each patient tumor were used for pathological diagnosis and confirmation. Gleason scores and tumor cellularity were evaluated by two genitourinary pathologists (T.v.d.K., and B.T.) on scanned haematoxylin- and eosin-stained (H&E) slides. All tumor tissues were macrodissected to reach 70% cellularity for the multi-omic experiments. The RFS was calculated with a biochemical recurrence (BCR) defined as two consecutive measurements of PSA > 0.2 ng/mL after RadP.

### Antibodies and Cell Lines
The details of antibodies and cell lines used in the present study are shown in the Supplementary data (Table S4). All prostate cancer cell lines were cultured in recommended media (Table S5). All cells were cultured at 37 °C in 5% CO2 cell culture incubator (Thermo Scientific, USA).

### Multiomic Workflow
The multiomic analysis in the discovery cohort was performed according to the following procedures (Fig. 1A). The prostate tissues were pulverized using the CryoPrepTM CP02 (Covaris) and then divided into three parts. For each case, 30 mg tissue sample was used for DNA extraction and whole exome sequencing (WES); 200 mg tissue sample was immediately transferred into a 1.5 mL EP tube and then added 1 mL RNA later reagent (Invitrogen) for RNA sequencing (RNA-seq); 50 mg tissue sample was lysed with SDS lysis buffer (4% SDS, 100 mM Tris-HCl, 0.1 M DTT, pH 7.6) and kept in -80°C for the following proteomic and phosphoproteomic analyses. And the remaining tumor tissues of each case were processed to the metabolomics analysis.

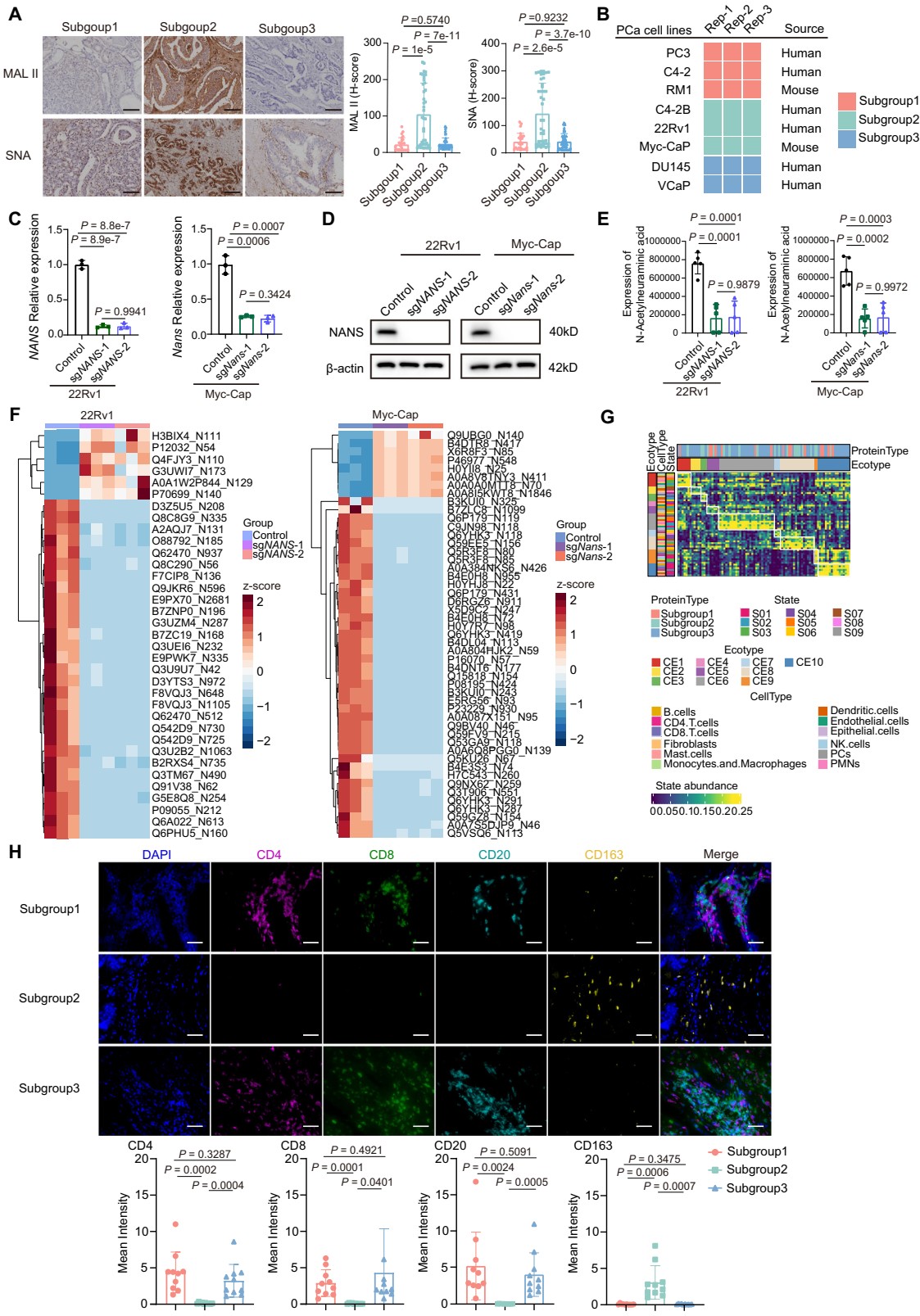

## DNA Extraction and Whole Exome Sequencing (WES)

Genomic DNA from prostate tissues were extracted using TIANamp Genomic DNA Kit (TIANGEN) according to manufacturer's protocol. DNA concentration was measured by Qubit® DNA Assay Kit in Qubit® 3.0 Flurometer (Invitrogen, USA). The exome sequences were efficiently enriched from genomic DNA using Agilent liquid capture system (Agilent SureSelect Human All Exon V6). DNA library were sequenced on Illumina NovaSeq6000 System for paired-end 150 bp reads to achieve a mean coverage depths of 300x on target coverage for all samples.

## WES Data Analysis

**Somatic mutation calling and filtering.** The original fluorescence image files obtained from Illumina platform are transformed to

**Fig. 4 | NANS increases the sialic acid anabolism and sialylation level accounting for immunosuppressive microenvironment of PCa. A** IHC staining and quantification for α2,3 and α2,6-sialylation modifications in tumor regions of three proteomic subgroups in the discovery cohort (n = 37/44/38 for subgroup 1/2/3). The scale bar represents 100 um. Data are presented as mean ± SD. *P* values are determined using two-tailed Student's t test. **B** Heatmap visualizes the proteomic subtypes of 8 PCa cell lines based on the nearest template prediction (NTP) algorithm. Three replicates for each cell line. **C** Knockout (KO) of *NANS* in 22Rv1 and Myc-CaP cells via single guide RNA (sgRNA) as determined by qRT-PCR. Empty vector is used as control. n = 3 biological replicates for each group. Data are presented as mean ± SD. *P* values are determined using two-tailed Student's t test. **D** KO of *NANS* in 22Rv1 and Myc-CaP cells via sgRNA as determined by Western blot. Empty vector is used as control. *P* values are determined using two-tailed Student's

t test. **E** Comparison of the relative abundance of N-Acetylneuraminic acid in 22Rv1 and Myc-CaP cells between sg*NANS* and control group based on the targeted metabolomics analysis. The middle lines represent the median. Lower/upper hinges denote 25–75% IQR with whiskers extending to 1.5 IQR. n = 5 biological replicates for each group. *P* values are determined using two-sided Wilcoxon rank-sum test. **F** Heatmap showing differentially expressed sialylated sites in 22Rv1 and Myc-CaP cells between sg*NANS* and control group. **G** EcoTyper analysis dissecting the tumor immune microenvironment of patients in three proteomic subgroups of the discovery cohort. **H** Multiplex immunofluorescence (mIF) staining and quantification of immune cells in three proteomic subgroups of the discovery cohort. n = 10 different samples for each group. The scale bar represents 50um. Data are presented as mean ± SD. *P* values are determined using two-tailed Student's t test. Source data are provided as a Source Data file.

short reads (raw data) by base calling and these short reads are recorded in FASTQ format. Clean data was mapped to the GRCh38/hg38 human reference genome by Burrows Wheeler Aligner (BWA) software and Samblaster to generate BAM file. Subsequently, Sambamba (v0.6.8, http://github.com/biod/sambamba/releases) was used to sort BAM files and mark duplicate reads according to chromosome position, then the validated BAMs were used for downstream analysis and variant calling. Possible somatic variants and indels (insertion/deletion) on exome data of tumor and matched non-tumor pairs were detected using MuTect (v2.2-25, https://github.com/broadinstitute/mutect) and Strelka (v1.0.13, https://github.com/Illumina/strelka/releases) with default parameters, respectively. ANNOVAR (v2017 June8, https://annovar.openbioinformatics.org/en/latest/user-guide/download/) was used to perform variant annotation.

**Exome-based somatic copy number alteration (SCNA) analysis.** Somatic Copy number alteration was called by CNVkit (v0.9.9, https://github.com/etal/cnvkit) and GISTIC 2.0 (https://github.com/broadinstitute/gistic2). Nomal-panel of all tumor samples was constructed and then each tumor compared to normal-panel, CNA were inferred by CNVkit using Circular Binary Segmentation algorithm with default parameters. Segment-level ratios were calculated and log2 transformed and copy number segmentations derived from CNVkit were processed with to retrieve gene-level copy number values and focal peaks. GISTIC2 was used to generate arm level and focal level SCNAs for the cohort with G-Score and FDR Q value indicating the significance and strength of the identified SCNAs.

**TMB analysis.** TMB was defined as the number of somatic mutations (including base substitutions and indels) in the coding region. To reduce sampling noise, synonymous alterations were also counted. In order to calculate the TMB, the total number of mutations counted was divided by the size of the coding sequence region of the Agilent SureSelect Human All Exon V6.

**RNA Extraction and Sequencing.** RNA was extracted from prostate tissues by using TRIzol reagent kit (Ambion, Invitrogen, USA) according to the reagent protocols. RNA integrity was assessed using the RNA Nano 6000 Assay Kit of the Bioanalyzer2100 system (Agilent Technologies, CA, USA). RNA samples exhibiting an RNA integrity number (RIN) greater than 4.0 were subjected to RNA sequencing. RNA library for RNA-seq was prepared as rRNA depletion and stranded method. And the concentration of library was measured by the Qubit® fluorometer and adjusted to 1 ng/uL. Agilent 2100 Bioanalyzer was deployed to examine the insert size of the acquired library. After library preparation and pooling of different samples, the samples were subjected for Illumina Novaseq 6000 sequencing and the RNA-seq use PE150 (paired-end 150nt) sequencing.

## RNA-seq Data Analysis

**RNA-seq data analysis with HISAT2.** Sequenced reads were trimmed with fastp V.0.23.1. In this step, clean data (clean reads) were obtained by trimming reads containing adapter or with low quality from raw data. Then clean reads were mapped to the reference genome (GRCh38) using HISAT2 V.2.1.0 and with default parameters. RSEM (v1.2.29) was used to quantify gene abundance as read counts. The R package "DESeq2" was applied to screen differentially expressed messenger RNAs (mRNAs) between different groups. Next, the *P*-value was calculated by the false discovery rate (FDR)-corrected method. The mRNAs with | log2 fold-change | >1 and P-adjusted <0.05 were filtrated as differentially expressed genes.

**Fusion events detection.** For searching gene fusion in the transcriptome, we applied STAR-Fusion (version 1.10.1) to chimeric-junction files generated in the previous RNA-seq variants calling procedure. The detected fusions were filtered if supporting reads were less than 20.

**ssGSEA analysis.** ssGSEA, a method for quantifying gene set enrichment in individual samples, was employed using the Gene Set Variation Analysis (GSVA) package in R. This approach allows for the assessment of variations in pathways and biological processes across a sample population, thus providing insights into the heterogeneity of immune microenvironment in patients of diverse proteomic subgroups. The ssGSEA analysis was based on 20 immune gene sets, including genes related to different immune cell types, functions, pathways, and checkpoints.

**MCP-counter analysis.** The normalized log2-transformed FPKM expression matrix was loaded into the MCP-counter package (V.1.1.0),29 which produces the absolute abundance scores for eight major immune cell types (CD3+ T cells, CD8+ T cells, cytotoxic lymphocytes, natural killer (NK) cells, B lymphocytes, monocytic lineage cells, myeloid DCs, and neutrophils), endothelial cells (ECs), and fibroblasts.

**EcoTyper analysis.** EcoTyper constructed a global atlas of transcriptionally distinct cell states from 16 types of human carcinoma and enabled the large-scale profiling of cell states and multicellular ecosystems[53]. To identify multicellular communities, the CIBERSORTx software (https://cibersortx.stanford.edu/) was applied to assess the accuracy of cell state recovery using transcriptomic data.

## Peptides Preparation, LC-MS/MS and Database Searching

**Protein extraction and digestion.** For tissue samples frozen and kept in liquid nitrogen, approximately fifty milligrams tissue samples were homogenized by MP FastPrep-24 homogenizer (24×2, 6.0 M/S, 60 s, twice) and then added in the SDT buffer lysis buffer (4%SDS, 100 mM Tris-HCl, pH7.6). After centrifuged at 14,000 g for 40 min, the supernatant was quantified with the BCA Protein Assay Kit (Bio-Rad, USA).

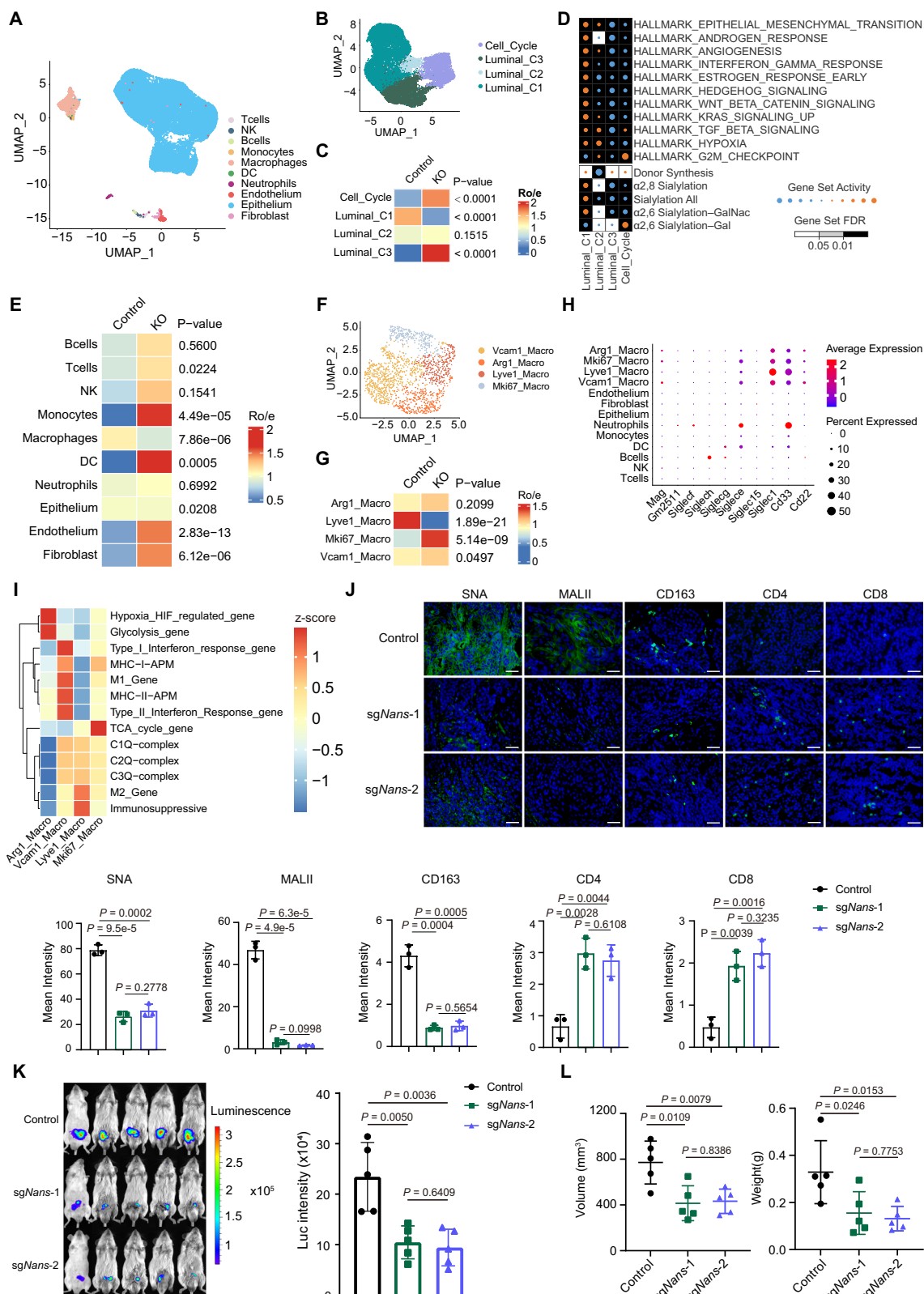

Protein sample was digested by filter-aided sample preparation protocol (FASP) (Wisniewski et al., 2009) using 10 kDa centrifugal filter tubes (Millipore). Trypsin was added to the samples(the trypsin: protein (wt/wt) ratio was 1:50) and incubated at 37 °C for 15–18 h (overnight). The peptides of each sample were desalted on C18 Cartridges (Empore™ SPE Cartridges C18 (standard density), bed I.D. 7 mm, volume 3 ml, Sigma), concentrated by vacuum centrifugation and

reconstituted in 40 µl of 0.1% (v/v) formic acid. The peptide content was estimated by UV light spectral density at 280 nm.

For formalin-fixed paraffin-embedded (FFPE) tissue samples, add 12.5 ul pH=10 100 mM tris-HCl(Tris hydrochloride) to the PCT tubes containing the samples, and place them on a horizontal shaker at 600 rpm for 30 min at 95 °C. 30 µL 6 M Urea/2 M thiourea/100 mM triethylammonium bicarbonate (TEAB), 5 µL 200 mM Tris (2-

**Fig. 5 | Inhibition of NANS reverses the immunosuppressive microenvironment and suppresses the tumor growth of PCa in orthotopic transplanted mouse model. A** Uniform manifold approximation and projection (UMAP) visualization of 29515 single cells from 6 mice tumor tissue samples, colored by cell type annotations. **B** UMAP visualization of epithelial cells, colored by cell type annotations. **C** Comparison of the Ro/e (the ratio of observed to expected cell numbers) in 4 epithelial cell subtypes before and after *Nans* knockout. P-values are determined using two-sided Chi-square test. **D** Bubble plot showing differentially activated hallmark pathways and sialic acid-related pathway among 4 epithelial cell subtypes. **E** Comparison of the Ro/e in all cell types before and after *Nans* knockout. *P* values are determined using two-sided Chi-square test. **F** UMAP visualization of macrophages, colored by cell type annotations. **G** Comparison of the Ro/e in 4 macrophage subtypes before and after *Nans* knockout. *P* values are determined using two-sided Chi-square test. **H** Dot plot depicting the expression of detected Siglec genes in each identified cell subtype, where dot size and color represent percentage of marker gene expression (pct. exp) and the averaged scaled expression (avg. exp. scale) value, respectively. **I** Heatmap depicting the enrichment of classical immune-related pathways in each identified macrophage subtype. **J** IF staining and quantification of sialylation level (SNA/MALII) and immune cells in murine tumor tissues of the sg*Nans* and control group. *n* = 3 for each group. The scale bar represents 50um. Data are presented as mean ± SD. *P* values are determined using two-tailed Student's t test. **K** Ex vivo imaging of murine tumor tissues of the sg*Nans* and control group. n = 5 for each group. Data are presented as mean ± SD. *P* values are determined using two-tailed Student's t test. **L** Comparion of tumor volume and weights between sg*Nans* and control group. *n* = 5 for each group. Data are presented as mean ± SD. *P* values are determined using two-tailed Student's t test. Source data are provided as a Source Data file.

carboxyethyl) phosphine (TCEP) and 2.5 μL 800 mM iodoacetamide (IAA) was added for reduction and alkylation. 85 μL TEAB was added into PCT tube to decrease the concentration of Urea to lower than 1.5 M. Then 10 ul Trypsin (0.5 ug/ul), 5 ul Lys-C (0.25 ug/ul) were added for protein digestion. Tryptic peptides were transferred into 1.5 mL tubes and digestion was then terminated by 15 μL 10% TFA. Then confirm that pH of the samples was between 2 and 3. SOLAμ (Thermo Fisher Scientific™, San Jose, USA) was applied for desalting and then desalted peptide samples were sent for further analysis.

**Phosphopeptides enrichment.** The phosphopeptides enrichment was performed using High-Select Fe-NTA kit (Thermo Scientific) according to the kit manual. First, 800 ug peptides reconstituted in 80% ACN/0.1% trifluoroacetic acid were incubated with Fe3 + -IMAC beads for 30 mins. The beads were then brought up in 80% ACN, 0.1% trifluoroacetic acid and loaded onto equilibrated C-18 Stage Tips, and washed by 80% ACN, 0.1% trifluoroacetic acid. Tips were rinsed twice with 1% formic acid, followed by sample elution off the Fe3 + -IMAC beads and onto the C-18 State Tips with 50% ACN, 0.1% formic acid twice. After lyophilized, the phosphopeptides were resuspended in 0.1% formic acid prior to ESI-LC-MS/MS analysis.

**Sialylated peptides enrichment.** For sialylated peptide enrichment, High-Select™ TiO2 Phosphopeptide Enrichment kit from Thermo Scientific were used for digested peptide mixtures from each sample. First, the desalted peptides were suspended in 150 μl binding/equilibration buffer and applied to the TiO2 spin tip. Afterwards, the TiO2 spin tips were washed with Binding/Equilibration buffer and Elution Buffer. The elutes of peptides were dried immediately and suspended with 40 μl 40 mM NH4HCO3 for deglycosylation by adding 2 μl PNGase F and 2U Sialidase A and keeping 3 hr incubation at 37 °C. The deglycosylated peptides were dried and resuspended with Binding/Equilibration buffer for further LC-MS analysis.

**Peptide fractionation.** we used hybrid spectral library generation in order to increase the depth and precision of protein or phosphopeptide identification. An equal aliquot from each sample in this cohort study was pooled into one sample as pool peptides. Digested pool peptides were then fractionated to 10 fraction using Thermo Scientific™ Pierce™ High pH Reversed-Phase Peptide Fractionation Kit. Each fraction was desalted on C18 Cartridges (Empore™ SPE Cartridges C18 (standard density), bed I.D. 7 mm, volume 3 ml, Sigma) and reconstituted in 40 μl of 0.1% (v/v) formic acid. The iRT calibration peptides were spiked before data-dependent acquisition (DDA) analysis.

**Quality control for LC-MS/MS.** We use spectronaut software as a larger set of calibration peptides to perform an extensive calibration and improve iRT (indexed retention time) precision and accuracy called iRT peptides. For both DDA and DIA experiments, iRT calibration peptides (Biognosys|iRT Kit |) were spiked into every sample to check chromatographic stability conveniently. The pooled peptides lysate was measured every ten samples runs as the quality-control standard. The quality-control standard was digested and analyzed using the same method and conditions as the tumor samples. A pairwise Spearman's correlation coefficient was calculated for all quality-control runs in the statistical analysis.

**LC-MS/MS.** All fractions for DDA library generation were injected on a Thermo Scientific Thermo Orbitrap Exploris 480 mass spectrometer connected to an Easy-nLC 1200 chromatography system (Thermo Scientific). The peptide was separated on a C18 Analytical Column (Thermo Scientific, ES802, 1.9 μm, 75 μm*20 cm) with a linear gradient of buffer B (84% acetonitrile in 0.1% formic acid) at a flow rate of 300 nl/min. A 90-min gradient was set as follows, 5%-5% B in 1min38s; 5%-6% B in 1min38s; 6%-8% B in 4min32s; 8%-10% B in 5min53s; 10%-12% B in 7min12s; 12%-16% B in 14min54s; 16%-23% B in 24min50s; 23%-26% B in 7min12s; 26%-28% B in 3min19s; 28%-30% B in 2min46s; 30%-32% B in 2min12s; 32%-34% B in 50s; 34%-36% B in 1min6s; 36%-39% B in 1min6s; 39%-50% B in 30 s; 50%-100% B in 27 s; 100%-100% B in 9min49s. MS detection method was positive ion, the scan range was 350-1500 m/z, resolution for MS1 scan was 60000 at 200 m/z, target of AGC (Automatic gain control) was 1e6, maximum IT was 50 ms, dynamic exclusion was 10.0 s. After each full MS—SIM scan, 20 ddMS2 scans followed according to inclusion list. Isolation window was 1.5 m/z, resolution for MS2 scan was 30000 (@m/z 200), AGC target was 1e5, maximum IT was 50 ms and normalized collision energy was 30 eV.

The peptides of each sample were analysed by Thermo Orbitrap Exploris 480 mass spectrometer connected to an Easy-nLC 1200 chromatography system in the data-independent acquisition (DIA) mode. Each DIA cycle contained one full MS—SIM scan, and 44 DIA scans covered a mass range of 350–1500 m/z with the following settings: SIM full scan resolution was 120,000 at 200 m/z; AGC 3e6; maximum IT 30 ms; profile mode; DIA scans were set at a resolution of 30,000; AGC target 3e6; Max IT auto; MS2 Activation Type HCD; normalized collision energy was 30 eV.

**Database searching of MS data.** For DDA library data, the FASTA sequence database was searched with Spectronaut (Biognosys) software. The database was downloaded at website:http://www.uniprot.org. iRT peptides sequence was added(Biognosys|iRT Kit |).The parameters were set as follows: enzyme is trypsin, max missed cleavages is 1, fixed modification is carbamidomethyl(C), dynamic modification is oxidation(M) and acetyl(Protein N-term). All reported data were based on 99% confidence for protein identification as determined by false discovery rate (FDR) ≤ 1%.

DIA data was analyzed with Spectronaut searching the above constructed spectral library. Calibration was set to non-linear iRT calibration with precision iRT enabled. Quantity was determined on MS/MS level using area of XIC peaks with enabled cross-run normalization.

Main software parameters were set as follows: retention time prediction type is dynamic iRT, interference on MS2 level correction is enabled, and cross-run normalization is enabled. All results were filtered based on Q value cutoff 0.01 (equivalent to FDR < 1%).

## Proteome and Phosphoproteome Data Analysis

**Data filtering and normalization.** For proteomics and phosphoproteomics, the data matrix was filtered to contain a maximum of 50% missing values across all samples, respectively. The remaining missing values were imputed using K-nearest neighbor (KNN) imputation implemented in the R-package (impute, v1.68.0). Finally, ComBat adjusted for batch effects in protein and phosphorylation data using the R-package (svr, version: 3.42.0). The batch effect was assessed by performing unsupervised PCA on the proteomic data. The leading PCs of the global proteomic data clearly separated the tumor from normal samples, and no obvious batch effect was observed.

**Unsupervised clustering analyses using NMF.** Non-negative matrix factorization (NMF)[54], an unsupervised clustering algorithm, and the R-package (NMF, version: 0.24.0) were used. Prior to the clustering analysis, transcriptomic and proteomic data preprocessing was performed. Standard deviations were calculated, and the top 25% standard variations were subjected to the clustering analysis (number of runs = 100, seed=6). The number of clusters considered ranged from 2 to 5. We compared the average pairwise consensus matrix for consensus clusters, the delta plot of the relative change in the area under the cumulative distribution function (CDF) curve, and the average silhouette distance within consensus clusters to determine the optimal factorization rank. The optimal k was defined as the maximum of the product of both metrics for cluster numbers between k = 2 and 5. The Fisher test was used to calculate the significance of clinical characteristics in different clusters.

**Pathway enrichment analysis.** The Wilcoxon test was used to test whether the differences between the subgroups were statistically significant. Differential expression analysis between different clusters with FC > 1.5 and $P$ value < 0.05 or FC < 1/1.5 and $P$ value < 0.05. Significantly differentially expressed proteins was performed KEGG pathway enrichment analysis using the R/Bioconductor clusterProfiler package (v4.2.2) and an FDR value < 0.05 was considered as the cut-off for significantly regulated pathways. Gene Set Enrichment Analysis (GSEA) was performed using the R/Bioconductor package clusterProfiler (v4.2.2). The Molecular Signatures Database (MSigDB) of hallmark gene sets and KEGG gene sets were used for enrichment analysis. An FDR value of 0.05 was used as a cut-off. The enrichment score (ES) in GSEA was calculated by first ranking the fold change of proteins from the most to least significant with respect to the two phenotypes, then using the entire ranked list to assess how each gene set was distributed across the ranked list.

**Kinase-substrate enrichment analysis.** To estimate changes in a kinase activity, kinase-substrate enrichment analysis (KSEA) (Wiredja et al., 2017) was performed on significantly differentiated phosphosites in two phenotypes. Known kinase-substrate site relationships from putative downstream targets (PSP) (Hijazi et al., 2020) were used as the K-S sources. A kinase score was given for each kinase based exclusively on the collective phosphorylation status of its substrates and transformed into z-score. For the kinase enrichment analysis, the threshold for significantly enriched kinases was p-value < 0.05.

**Machine learning for biomarkers screening.** Prior to select protein biomarkers, differential analysis (Wilcoxon test) was performed between one subtype and other two subtypes (considered as one group). FC ≥ 1.4 & $P$-adjusted<0.05 was considered as significant difference. Next, the top 50 differentially expressed proteins were subjected to the following analysis. Random forest algorithm was used to find potential biomarkers. Proteomics data was divided into training and test datasets with 7:3 ratio and cumulative AUC was applied to select minimum number of biomarkers with the training dataset (10-fold cross-validation). Finally, the model performance was measured using the test dataset.

**Unsupervised clustering using NTP.** To validate our proteomic subtyping results in the validation cohort, the nearest template prediction (NTP)[55], an unsupervised clustering algorithm, was adopted using the 39 proteins panel filtered by machine learning. The NTP algorithm provides a convenient method to make category predictions, using only a gene list and a test dataset to evaluate the prediction confidence calculated in each patient's gene expression data. To further analyze the biological characteristics of these subgroups, we performed Gene Set Enrichment Analysis (GSEA) to identify the pathway alterations that underlying each subgroup.

## Metabolites Extraction and LC−MS/MS Analysis

**Metabolites extraction.** The remaining tumor tissues of each case (21 tumor samples in total) after DNA/RNA/protein extraction were processed to the metabolomics experiments. 1000 µL of cold methanol/acetonitrile/water (2:2:1, v/v) extraction solvent was added to 80 mg sample with stock solutions of stable-isotope internal standards. The samples were homogenized and centrifuged at 14,000 g for 20 minutes at 4 °C and the supernatant was dried in a vacuum centrifuge at 4 °C.

**LC−MS/MS analysis for metabolites.** For LC-MS analysis, the samples were re-dissolved in 100 µL acetonitrile/water (1:1, v/v) solvent and centrifuged at 14,000 g at 4 °C for 15 min, then the supernatant was injected. LC−MS/MS analyses were performed using a UHPLC (1290 Infinity LC, Agilent Technologies) coupled to a QTRAP MS (6500 + , Sciex) in Shanghai Applied Protein Technology Co., Ltd. The analytes were separated on HILIC (Waters UPLC BEH Amide column, 2.1 mm × 100 mm, 1.7 µm) and C18 columns (Waters UPLC BEH C18-2.1×100 mm, 1.7 µm). The sample was placed at 4 °C during the whole analysis process. 6500 + QTRAP (AB SCIEX) was performed in positive and negative switch mode. MRM method was used for mass spectrometry quantitative data acquisition. The MRM ion pairs are showed in the attached file. A polled quality control (QC) samples were set in the sample queue to evaluate the stability and repeatability of the system. The Multiquant software was used to extract chromatographic peak area and retention time for quantitative data processing. Use the standards correct retention time to identify the metabolites. The QC samples were processed together with the biological samples and used for testing and evaluation the stability and repeatability of this system. Metabolites in QCs with the coefficient of variation (CV) less than 30 % were denoted as reproducible measurements.

## Single-cell Preparation and RNA-Sequencing

**Single-cell dissociation.** The mice tumor tissues were surgically removed and kept in MACS Tissue Storage Solution (Miltenyi Biotec) until processing. First, samples were enzymatically digested with 10 U/mL collagenase I (Worthington), 400 U/mL collagenase I (Worthington) 1 mg/ml Dispase (Worthington) and 30 U/mL DNase I (Worthington) for 45 min at 37 °C, with agitation. After digestion, samples were sieved through a 70 µm cell strainer, and re-suspended in PBS containing 0.04% BSA and re-filtered through a 35 µm cell strainer. Dissociated single cells were then stained with AO/PI for viability assessment using Countstar Fluorescence Cell Analyzer.

**Single-cell RNA-sequencing (scRNA-seq).** The scRNA-Seq libraries were generated using the 10X Genomics Chromium Controller

Instrument and Chromium Single Cell 5′ library & gel bead kit. Briefly, cells were concentrated to 1000 cells/uL and approximately 10,000 cells were loaded into each channel to generate single-cell Gel Bead-In-Emulsions (GEMs), which results into expected mRNA barcoding of 5,000-8000 single-cells for each sample. After the RT step, GEMs were broken and barcoded-cDNA was purified and amplified. The amplified barcoded cDNA was used to construct 5′ gene expression libraries. For 5′ library construction, the amplified barcoded cDNA was fragmented, A-tailed, ligated with adaptors and index PCR amplified. The final libraries were quantified using the Qubit High Sensitivity DNA assay (Thermo Fisher Scientific) and the size distribution of the libraries were determined using a High Sensitivity DNA chip on a Bioanalyzer 2200 (Agilent). All libraries were sequenced by illumina sequencer (Illumina, San Diego, CA) on a 150 bp paired-end run.

### Single-cell RNA Data Analysis

**Data filtering and statistical analysis.** Fastp with default parameter was applied for filtering the adaptor sequence and removed the low-quality reads to achieve the clean data. Then the feature-barcode matrices and aggregated matrix were obtained by aligning reads (include intron = true) to the mouse genome (mm10 Ensemble: version 100) using CellRanger-7.1.0. Seurat package (version: 4.1.1, https://satijalab.org/seurat/) was used for cell normalization and regression based on the expression table according to the UMI counts of each sample and percent of mitochondria rate to obtain the scaled data. Doublets and multiplets was filtered by Doubletfinder with parameter as follow: pN = 0.25; resolution = 0.8; DoubletRate = 0.1. In order to remove the batch effect, we used the fastMNN function (k = 5, d = 50, approximate = TRUE, auto.order = TRUE) from R package scran (v1.10.2) and applied the mutual nearest neighbor method based on the scale data of top 2000 high variable genes and sample batch info. Utilizing the graph-based cluster method (resolution was optimized in different sub-clustering result of different cell type), we acquired the unsupervised cell cluster result based the MNN top 10 principal and we calculated the significant marker genes by FindAllMarkers function with the Wilcox rank sum test algorithm under following criteria:1. lnFC > 0.25; 2. $p$ value < 0.05; 3. min.pct>0.1 and only markers with p.adj<0.05 was used for cell identification. In order to identify the sub-cell type detailed, the clusters of same cell type were selected for sub-clustering analysis, graph-based clustering and marker analysis.

**Differential gene expression analysis.** To identify differentially expressed genes among samples, the function FindMarkers with wilcox rank sum test algorithm was used under following criteria:1. logFC > 0.25; 2. pvalue < 0.05; 3. min.pct>0.1.

**QuSAGE analysis (gene enrichment analysis).** To characterize the relative activation of a given gene set, such as pathway activation, we performed QuSAGE (2.16.1) analysis based on and immune response gene sets from articles.

Cell Gene Enrichment Analysis: We applied single-cell gene set enrichment analysis analysis based on the domestic gene set and normalized gene expression matrix by ssGSEA function in GSVA package to achieve the gene enrichment score of each cell.

**Immunohistochemistry and Immunofluorescence Staining.** For immunohistochemistry staining, the tissue slides were fixed in 4% paraformaldehyde (PFA) for 12 hours, embedded by paraffin and sectioned (3 μm). Antigen retrieval was performed by boiling slides in tris-EDTA. Subsequently, the slides were blocked for peroxidases and nonspecific binding of antibodies using 3% H2O2. Then, slides were blocked with 10% goat serum, and incubated with primary antibody at 4 °C overnight. After washing, secondary antibodies were incubated for 30 min, followed by DAB (Aglient, K5007) and hematoxylin staining. Finally, slides were scanned using KF-PRO-020 Digital Slide Scanner (KFBio, China).

For immunofluorescence staining, after incubation with peroxidase-conjugated secondary antibodies, slides were incubated with dyes-labeled tyramide (Invitrogen). For quantification, positive cells were counted from at least 3 random slides of each mouse for a total of 4–8 pairs.

For immunofluorescence staining, the 3-μm-thick slides cut from the formalin-fixed paraffin-embedded blocks were dewaxed, rehydrated through, and fixed in neutral buffered formalin (10% NBF). Slides were incubated with primary antibodies overnight at 4 °C. The next day, slides were incubated with fluorescently labeled secondary antibodies and DAPI for 2 h at room temperature.

After washing with PBST for five times, the slides were sealed with an anti-quench sealing tablet. We used the TissueFAXS platform (TissueGnostics) at 20× magnification to scan the stained slides, then analysed the images with the ImageJ software.

**Multiplex immunohistochemistry.** Multiplex immunohistochemical staining was carried out using the PANO 7-plex IHC kit (Panovue, 10217100100), following the manufacturer's instructions. Briefly, tissue slides were first baked, dewaxed, rehydrated, and fixed in 10% neutral formalin. Antigen retrieval was achieved by heating in Tris-EDTA buffer using a microwave. After cooling, the sections were blocked with 10% goat serum to prevent nonspecific binding, followed by incubation with the primary antibody at 37 °C for 30 minutes. Next, HRP-conjugated secondary antibodies were applied for 10 minutes at room temperature, and Tyramide Signal Amplification (TSA) with fluorophores Opal was performed for 15 minutes. The antigen retrieval process was repeated to remove the antibody-TSA complex, allowing for the start of a new staining cycle. This process was repeated for four or five planned antibodies. Finally, the TSA-labeled slides were counterstained with DAPI and mounted using a mounting medium (Panovue, 0022001010). The stained slides were then scanned using the TissueFAXS Spectra imaging system (TissueGnostics, Austria).

### Functional Experiments

**NANS knockout.** CRISPR-Cas9-mediated *NANS* knockout is achieved by lentiCRISPRv2 system as previously reported[56]. *NANS* sgRNA sequences were cloned into the lentiCRISPR v2 vector. 22Rv1 and Myc-CaP cells were infected with empty or NANS-sgRNA lentivirus. Virus was incubated with target cells for 24 hours with 10 mg/mL polybrene, and cells were allowed to recover for 24 hours before selection. Infected cells were selected in 2 mg/mL puromycin until uninfected control cells were dead. Knockout efficiency was validated by Western blotting. The sgRNA sequences for 22Rv1 cells are listed as follows: sgNANS-1 (5′-TATGTGACGTTCCAACACCT-3′), sgNANS-2 (5′- TCAT GCCCAGAATACCCTAT-3′). The sgRNA sequences for Myc-CaP cells are listed as follows: sgNans-1 (5′-TCGTGCCCGGAATACCCGAT-3′), sgNans-2 (5′- GGGCTGTAGTGGGTACGCGC-3′).

**RNA isolation and quantitative qPCR.** Total RNA was extracted from $1.0 \times 10^5$ cells, and cDNA was synthesized with AMV-reverse transcriptase and random hexamer primers (Roche Diagnostic, Indianapolis, Indiana, USA). Quantitative PCR (qPCR) was performed using StepOneplus (Applied Biosystems, Carlsbad, California, USA). The relative amount of mRNA was calculated using the $2 - \Delta\Delta Ct$ method.

**Western blot.** 22Rv1 and Myc-CaP cells were lysed with RIPA buffer (Cat # PC101, EpiZyme Biotechnology, China) supplemented with protease inhibitor and phosphatase inhibitor cocktail (Roche). Protein extracts (20 μg) were separated on 10% PAGE Tris-Acetate gel (Cat # PG112, EpiZyme Biotechnology, China) by electrophoresis and subsequently transferred onto Immobilon-P PVDF membranes of 0.45 μm pore size (Cat # IPVH00010, Millipore, China). Membranes, which blocked with TBST containing 5% non-fat milk for 1 h, were incubated with the primary antibodies in 5% BSA in TBST overnight at 4 °C,

followed by incubating with the anti-rabbit secondary antibodies for 1 hour. Finally, western blot results were visualised by Amersham™ ImageQuant™ 800 (Cytiva).

**Cell proliferation assay.** For the CCK-8 assay, cells were seeded at a density of 2000 cells per well in a 96-well plate. After incubation, CCK-8 solution (C0039, Beyotime Biotechnology) was added to each well and incubated for 2 hours. Absorbance was then measured at 450 nm to determine cell viability.

For the colony formation assay, cells were plated in 6-well plates at a density of 500 cells per well. After approximately 7 days of incubation, cell colonies were washed with 1x PBS and stained with crystal violet for 20 min. The colonies were subsequently imaged and quantified to assess colony formation ability.

**Animal models.** All mice were maintained in a specific-pathogen-free facility and all related protocols were performed in compliance with the Guide for the Care and Use of Laboratory Animals and were approved by the Institutional Animal Ethics Committee of the First Affiliated Hospital of Sun Yat-sen University. The maximal tumor size/burden permitted ($2 \, cm^3$) by the ethics committee was not exceeded in the study. Additionally, all mice used in the experiments were male, as prostate tumors are specific to males. To establish the orthotopic transplanted mouse models of PCa, Myc-CaP cells stably expressing luciferase ($2 \times 10^6$ cells per mouse) were orthotopically injected into the dorsolateral prostate of FVB mice (eight-week-old, GemPharmatech, China) using a Hamilton syringe, $n = 5$ for each group. The tumor volume was measured by BLI using a NightOWL II LB 983 Imaging System (Berthold) every 10 days for 1 month. To establish the Pten/*Trp53* double knockout ($Pten^{PC-/-}$; $Trp53^{PC-/-}$) mouse models of PCa, *Pbsn-Cre4* transgenic mice were obtained from F. Wang[57]. *Pten*-floxed mice were generated by H. Wu[58]. *Trp53*-floxed mice were from the Jackson Laboratory. Mice were maintained on a C57BL/6 background in all genetically engineered mouse models (GEMMs). For in vivo studies, 2.5-month-old *Pten/Trp53* double knockout mice were administered orthotopic injection of adeno-associated virus (AAV) containing *Nans* shRNA ("aav-shNans-Sq1") or scramble control shRNA, $n = 5$ for each group. Each mouse received $1 \times 10^{11}$ genome copies (GC) in PBS/0.001% Pluronics F68. All mice were euthanized and tumors were harvested 1 month after inoculation, followed by photography and pathological analysis. The tumor size was measured using calipers in two dimensions to generate a tumor volume using the formula: $0.5 \times$ (length $\times$ width$^2$).

**Statistics.** Standard statistical tests were used to analyze the clinical data, including but not limited to Student's t test, Chi-square test, Fisher's exact test, Kruskal-Wallis test, Log-rank test. For categorical variables, Fisher's exact test or Chi-square test was used; and for continuous variables, Spearman correlation was used. All statistical tests were two-sided, and statistical significance was considered when $P$ value < 0.05. To account for multiple-testing, the $P$ values were adjusted using the Benjamini-Hochberg FDR correction. Kaplan–Meier plots (Log-rank test) were used to describe biochemical recurrence-free survival. The optimal cut-off point determined by the the 'maxstat' R package (maximally selected rank statistics) was used to explore the prognostic efficacy of a single protein. All the analyses of clinical data were performed in R (version 3.4.3). For functional experiments, each was repeated at least three times independently, and results were expressed as mean ± standard deviation (SD). Statistical analysis was performed using GraphPad Prism (version 5.01).

**Reporting summary**
Further information on research design is available in the Nature Portfolio Reporting Summary linked to this article.

## Data availability

The raw proteomics and phosphoproteomics data generated during this study have been archived in the iProX database (https://www.iprox.cn) and are available via the following accession IDs: PXD056748 (also available as IPX0009562000) for PCa patients, PXD058635 (also available as IPX0010395000) for PCa cell lines, and PXD058636 (also available as IPX0010400000) for mice models. The raw whole exome sequencing (WES) and RNA-seq data of PCa patients have been deposited in the Genome Sequence Archive (GSA-human, https://ngdc.cncb.ac.cn/gsa-human) under accession ID HRA008293; also available under accession number PRJCA029068. The raw single cell RNA sequencing data of mice models are available at the Genome Sequence Archive (GSA, https://ngdc.cncb.ac.cn/gsa) under accession ID CRA021269; also available under accession number PRJCA033071. The raw metabolomics data used in this study have been deposited in the OMIX database (https://ngdc.cncb.ac.cn/omix) and are available via the following accession IDs: OMIX008183 for PCa patients, and OMIX008185 for PCa cell lines. The processed matrix of patients' RNA-seq was also deposited on OMIX (accession number: OMIX008531). Dataset HRA008293 and OMIX008531 are available under restricted access because of data privacy and supervision. For research purpose, access can be obtained by the DAC (Data Access Committees) of the GSA-human database and OMIX database. The approximate response time for accession requests is about one month. Once access has been approved, the data will be available for two months. The remaining data are available within the Article, Supplementary Information or Source Data file. Source data are provided with this paper.

## Code availability

The codes generated in this study are publicly available in GitHub at https://github.com/Diluczhang/ProstateCancerProteins.git (DOI: 10.5281/zenodo.14878222). The remaining data are available within the Article or Supplementary Information.

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

## Acknowledgements
This work was funded by grants from the National Natural Science Foundation of China (NSFC-82372056, Wang). the National Natural Science Foundation of China (NSFC-82203677, Long). the National Natural Science Foundation of China (NSFC-82273299, Chen). Guangzhou Municipal Science and Technology Project (NO. 2023A04J2217, Wang). Sun Yat-sen University First Affiliated Hospital Kelin Emerging Talent Program (R07019, Wang). We wish to thank the Third and Seventh Affiliated Hospital of Sun Yat-sen University, Guangzhou First People's Hospital and the Daping Hospital, Army Medical University for providing us the slides for IHC staining and clinical data of PCa patients in the validation cohorts.

## Author contributions
Z.R.W. and W.O. conceptualized the project. H.C.U., M.H.L., J.H.L. and X.J.G. coordinated clinical sample collection. W.O., X.X.Z., B.L. and Y.T. curated the data. W.O., X.X.Z. and P.F.L. performed the formal analysis. Z.R.W., L.L.L., W.O. and X.X.Z. performed experiments. B.L., Y.T., P.F.L. and F.F.Z. provided technical support. W.O., X.X.Z., Y.T., R.X.L. and H.C.U. designed and supervised the experiments. Z.R.W., L.L.L. and L.W.C. acquired funding for the project. W.O., X.X.Z. and R.X.L. wrote the original draft of the manuscript. Z.R.W., L.L.L., L.W.C., W.O., X.X.Z., and J.P.G. reviewed and edited the manuscript. The order of the co-first authors was decided based on scientific contribution to the paper. All authors read and approved the final version of the manuscript.

## Competing interests
The authors declare no competing interests.
