## [Transparent Peer Review file · Nature Communications]

Integrated Proteogenomic Characterization of Localized Prostate Cancer Identifies Biological Insights and Subtype-Specific Therapeutic Strategies

Corresponding Author: Professor Zongren Wang

Version 0:

Reviewer comments:

Reviewer #1

(Remarks to the Author)

In the current study, Ou and colleagues performed multi-omic profiling of 145 cases of localized prostate cancer. Proteome-based stratification revealed three distinct subtypes: immune subgroup; arachidonic acid metabolic subgroup; and sialic acid metabolic subgroup which had highest rates of biochemical recurrence (BCR) compared to the other two subtypes. N-acetylneuraminic acid synthase (NANs) was identified to be a prognostic biomarker for aggressive prostate cancer. Using a combination of cell line-derived orthotopic mouse models, scRNA sequencing and immunofluorescence, authors found that targeting of NANs can reverse the immunosuppressive microenvironment through restricting the sialoglycan-sialic acid recognizing immunoglobulin superfamily lectin (Siglec) axis, thereby preventing tumor growth.

Overall, the reported findings are of interest and the multi-omic approach a major strength. Nonetheless, the manuscript is a bit hard to follow given the largely descriptive nature of the three clusters and there are some fundamental concerns regarding logic and reported data.

1. Sialic acid related enzymes- were there any other carbohydrate antigens measured beyond NANA that were differential between the groups? Additionally, were levels of NANA (or corresponding proteins) elevated in tumor compared to adjacent control tissues? It also appears that a subset of samples were used for these analyses. What is the rationale for sample selection? It would be more beneficial to evaluate all tumor tissues.
2. For the validation, a 39-protein subtyping panel was developed using Random Forest. It is unclear if this algorithm was applied to the original dataset and then validated in the independent 119 PCa samples or whether this was developed in the independent samples. It is also a bit odd that NANS was not in the 39-protein panel, despite being a primary focal point of the study.
3. Subgroup II was associated with poor prognosis, accompanied by higher preoperative PSA levels, higher postoperative Gleason Score as well as higher tumor mutational burden and somatic copy number aberrations compared to subgroup III. Authors state that elevated tumor NANs was associated with reduced bRFS; however, it should be shown that NANs is an independent predictor of bRFS when considering the above-mentioned variables. This can be evaluated using multivariate cox proportional hazard models.
4. NANs high vs low classifier. What was the criteria for NANs High vs Low described in Figure 3? Additionally, was this based on the entire specimen set or restricted to the subgroup II samples?
5. It is a bit unclear why Myc-CaP cells were used based on proteomic subtyping if the intention is to evaluate the impact of NANs on the tumor immunophenotype. It would be of considerable interest to evaluate the impact of NANs regardless of the subgroup.
6. Figure 4E- for sgNANs- was this based on one of the two clones or both? Additionally, both parental control as well as sgControl should be used when comparing against sgNANS cells. Moreover, it is recommended that authors replicate key experiments using at least one additional PCa model system.

7. The sample size for Figure 4I should be increased. This is based on an N = 3 and it is unclear how these samples from the discovery cohort were chosen.
8. Figure 5C- the reported p-values do not reflect the bar plots which show considerable variability and are based on an N= 3.
9. Figure 5I- please show all mice rather than a single representative picture. For statistical analyses, what type of test was used? This information should be provided. This comments holds true for all relevant figures.

Minor

Descriptive statistics (e.g. mean, stdev, fold-change, pairwise analyses, etc) for the 9,252 proteins from the 145 tumors and 141 adjacent normal samples should be included as a supplemental datafile.

Reviewer #2

(Remarks to the Author)

The identification of indolent and aggressive disease is of significant clinical importance. In the current manuscript, the authors conducted proteogenomic analyses on a cohort of Chinese prostate cancer patients and identified a subset with poorer outcomes, and NANS as a possible transducer of modulation of immune microenvironment and potential therapeutic target.

Regarding the in vivo experiment:

The authors utilized MycCaP prostate tumor cells classified as subgroup II tumors, exhibiting more aggressive behavior. This choice of mouse tumor cells serves as an appropriate model for assessing the involvement of the immune system in tumor growth. The in vivo experiment involved orthotopic injection of parental and NANS knockdown cells, revealing that NANS knockdown tumors exhibited attenuated growth compared to parental cells. Additionally, the results indicated altered immune infiltration in the NANS knockdown tumors, supporting the authors' hypothesis.

However, several points require clarification and further analysis:

The manuscript mentions the use of six tumors for scRNA-Seq analysis in Figure 5A, but it's unclear whether these tumors represent control and NANS knockdown groups. Furthermore, the method by which alterations in the immune microenvironment were determined needs more details, especially since only a single UMAP was included. Moreover, it would be beneficial to include tumor cells in the single-cell RNA sequencing to demonstrate any altered tumor cells phenotype. Additionally, the statistical significance in Figure 5C, where only two or three data points are shown for each condition, needs clarification, especially considering the reported p-values reaching levels of e^{-9} .

Further analyses of the generated tumors are also needed. For instance, assessing the expression of NANS in vivo and determining whether alterations in NANS result in changes to tumor subgroup assignments after knockdown.

Unfortunately, the effects of NANS knockdown were demonstrated in a single model. Additional transgenic models should be evaluated to strengthen the evidence supporting the role of NANS in altering immune cell infiltration and promoting tumor progression in vivo.

Overall, while the in vivo experiment provides valuable insights, addressing these points would enhance the clarity and robustness of the findings.

Additional Comments

The authors identified three different subgroups of patients based on their proteomic analysis, with subgroup II having more aggressive disease and worse survival. Comparison with the existing parameters or combination of the new subgrouping with clinical parameters of existing methods to determine the aggressiveness of the disease and survival benefit differences to demonstrate potential clinical use for all three cohorts used should be included.

While the authors evaluated the Chinese patient cohort it would be of interest to include an analysis of TCGA, to determine whether these specific characteristics are unique for Chinese patients only.

The authors cite a paper on proteomic analysis of Caucasian patients that resulted in the identification of four protein groups and five patients' groups, with protein groups having increased immune signature and two of the patient's subgroups having worse outcome. Cross-analysis of these existing data with the herein reported data would be of interest.

The authors performed an analysis of the phosphoproteome of the tumor samples but this analysis and interpretation of its results are not incorporated well and sufficiently discussed in the manuscript.

The proteomic/ transcriptomic analyses of patients' tumor samples suggested alteration of immune infiltration and analysis of in vivo MycCaP tumors indicated NANS involvement, specifically M2 macrophages, IHC for M2 presence and NANS expression in patients' tumor samples should be performed and quantified to provide additional support for the conclusions.

Minor Comments

Clinical characteristics of the patients in the second and third clinical cohorts should be included.

I suggest an inclusion of the number of patients and percentage in Figures (e.g., 1C).

The coloring in Figure 2B is not clear.

Statistics for the results summarized in the following statement should be included. "subgroup II exhibited a higher prevalence of tumor suppressor gene deletions (such as RB1, 122 CHD1 and ZNF292) and AR activation-related amplification (GATA2) and deletions (MAP3K7, 123 SPOPL) (Figure 2C). While, gene fusion analysis revealed that patients harboring the TMPRSS2-124 ERG (T-E) gene fusion, involved in lipid metabolism functions in PCa22, belonging to the metabolism subgroup I (Figure 2C).

The gene list in the 39-gene panel used should be provided.

All data points should be plotted in all bar graphs and P value reported in a consistent fashion (as of now “*” and number are used).

The authors used low NANS and High NANS to evaluate the potential survival effects of NANS expression. No information was included how was the threshold for “low “ and ‘high” levels determined?

Reviewer #3

(Remarks to the Author)

Ou et al profiled the genome, transcriptome, proteome and phosphoproteome of 145 cases of localized prostate cancer. Their proteomic revealed 3 subtypes with distinct molecular features: immune subgroup, arachidonic acid and sialic acid metabolic subgroups with highest biochemical recurrence rates. They identified a key enzyme in sialic acid pathway called NANS as prognostic biomarker and a therapeutic target for aggressive prostate cancer disease. However targeting NANS shows modest effect in tumor growth and was not compared to standard of care. The manuscript is rich of omic data however RNA-seq received very little attention in the manuscript. This is a unfortunate, as various transcriptomic subtypes of prostate cancer have been defined and well characterized, including luminal and basal subtypes. It would be interesting to see how the proteomic subgroups defined in this manuscript relate to transcriptomic subtypes, and not doing so is a major weakness of this manuscript. The proteomic heatmap figure even includes an “RNA subgroup” annotation, but this is not mentioned anywhere in the manuscript. All figure captions need much more detail. The Reporting Summary says that no software was used for data analysis, but this is clearly not the case.

Specific comments:

- In Introduction, the authors mention ADT and the almost inevitable development of castration resistance. However, androgen receptor pathway inhibitors (ARPIs) are now increasingly being used in addition to ADT in the first-line treatment setting, even for localized disease (especially high-risk localized PCa), yet the authors fail to mention ARPIs in the Introduction.
 - Figure 1 feels odd, and even slightly confusing, to see all of the stacked bar plots before the heatmap with the actual clusters. The figure panels should be reorganized so that readers see the heatmap before any other subpanels of Fig 1B. In fact, why are the bar plots included in Fig 1B when they are not referred to in the text until after Fig 1D?
 - Some of the figure legend colors are difficult to distinguish from each other when the heatmap columns are so narrow: PSA “>20” and “10-20” are difficult to distinguish, T stages T3a and T3b are difficult to distinguish, and it’s impossible (to my eye) to see the intermediate risk group in the “Risk group” annotation at the top of the heatmap.
 - The heatmap includes “RNA Subgroup” along the top of the heatmap, but it has not yet been mentioned in the text and is also not mentioned in the caption. RNA subgroups are not defined anywhere in the manuscript, so why are they included in the heatmap in Fig 1B?
 - The GSEA methods for Fig 1C,E needs more detail. GSEA can only compare one group versus another, but there are 3 groups to compare in Fig 1, so what do the GSEA results for each subgroup actually represent? Did one group serve as a common reference for two separate GSEA comparisons, or was each group compared to the union of the other two groups, or something else?
 - Suppl Fig 1 C,D: Do the figures show the number of proteins detected per sample (this is what the axes labels and figure caption imply), as opposed to the cumulative number of proteins detected across samples? If so, it appears that every tumor sample had more proteins detected than every normal sample, which is very surprising – why is this? Is there some sort of systematic tumor vs normal bias in the MS analysis? This would make tumor vs normal biological comparisons challenging as they would be confounded by any systematic bias. Or do panels C and D show cumulative numbers of proteins identified with samples deliberately ordered with all Normal first then all Tumor? If so, the axes labels and the figure caption should be changed to make this clear.
 - Fig 2B: What do the light and dark shades in the bar plot mean? Does the height of the bar represent the total number of patients in a subgroup and the light shaded area represent the patients with FOXA1 mutations? This is what I assume based the oncoprint and the main text, but it is not at all clear. A much more informative legend and more detailed caption are needed.
 - Fig 2C: What does dark grey in the oncoprint indicate? Missing data? There is no legend for the dark grey color. What do the numbers at the right side of the oncoprint? Are they p-values? This needs to be explained in the figure panel and/or caption. If they are p values, what do they represent? Are they from ANOVA tests of associations of the genomic alterations with the proteomic subgroups? Caption needs much more information.
 - o Rows labelled as “del” (“RB1 del”, “CHD1 del”, etc.) show both deletions and amplifications, and there are separate legends for “Amp” and “Del”, yet “Amp” is also included in the “Del” legend. The legend for the mutations is incorrectly title as “CNV”. Is there a reason for including FOXA1 mutations in Fig 2C when it was already shown in Fig 2B? What about FOXA1 CNAs? FOXA1 amplification is common in prostate cancer, so this would surely be relevant to include in Fig 2C, especially if the authors are showing FOXA1 mutations and GATA2 amplifications.
 - Fig 2D: it is unclear what it is the meaning of Enrichment analysis is for Subgroup II versus what? Versus Subgroups I and III combined? Or versus Subgroup I. Caption needs more detail and/or the figure panel needs more annotation.
 - Supplementary Fig 2B:
 - o What statistical tests were performed? Are p values from ANOVAs? Or are they from a comparison of two particular subgroups, and if so what test was used? Or something else?
 - Supplementary Figure 2C: The GSEA compares Subgroup I to what? To Subgroups II and III combined? Or just Subgroup I? Or...?
 - Line 159: What is the validation cohort? No reference or details are provided in the main text.
- Lines 159—161: The authors developed a 39-protein subtyping panel and classifier? Considering that the proteomic subgroups with different survival probabilities were validated in a different cohort, this is very interesting and deserves more

details in the text!

- Supplementary Fig 2F: Does this show the 39 signature proteins in the discovery cohort or the validation cohort? This isn't clear.
- Fig 4B: The grouping of cell lines into proteomic subgroups is very surprising. DU145 and PC3 are generally considered to be aggressive-phenotype AR- NE- cells lines that cluster together in most analyses (e.g. based on transcriptomic or epigenomic data), while VCaP is essentially an AR+ NE+ cell line and C4-2 is a very AR-driven cell line. It is therefore very unexpected to see PC3 and C4-2 group together on one hand and DU145 and VCaP to group together on another hand. Can the authors comment on why proteome-based classification groups together cell lines that we know to be phenotypically very different, and separates cell lines that are known to be phenotypically similar? Does this represent a major disconnect between the transcriptome and proteome, and what are the clinical implications of this? Or is it possible that cell lines do not adequately reflect the proteome of primary prostate cancer since they were generated from mets?
 - o Is the proteomic classification of cell lines based on the 39-protein signature panel?
 - o Is the proteomic classification of cells based on proteomic data, or on application of the 39-protein signature to RNA (gene expression) data?
 - o Where did the data for these cell lines come from? Were they new data generated by this study, or were publicly available data used?
- Page 9 / Fig 4: Why did the authors choose to use a murine cell line when they showed that two human cell lines were also in the same subgroup. Given the availability of two human cell lines in the subgroup of interest, it seems like an odd choice to use a murine cell line for in vitro analyses. It even gives the impression that the authors tried human cell lines and the experiments didn't support the hypothesis, so they resorted to a murine cell line.

Comments on Methods

- The methods mention two validation cohorts, but this was not at all obvious in the main text. It is only explained once you dive in the Methods.
- Please provide the rationale of using hg19 used as the reference genome for mapping sequencing reads from human specimens? hg38 has been around for a long time and should be preferred.
- The methods describe V(D)J / TCR sequencing, but this is not mentioned anywhere else in the manuscript...
- Does the definition of TMB include subclonal mutations?
- Despite it's widespread use, FPKM is not a suitable normalization unit for RNA-seq data. TMM, Upper Quartile, or TPM normalization would be better.
- The best way to include a logFC threshold into differential expression analysis is to build it into the statistical testing procedure (as enabled by modern differential expression analysis methods for RNA-seq), rather than to test for differential expression different from 0 and subsequently apply a logFC threshold post hoc.
- Line 642: why was limma needed for ssGSEA?
- Please provide references for the gene sets used for ssGSEA.
- What signature matrix was used for CIBERSORTx?
- Line 669: When the authors say "data were selected after pretreatment", do they mean "... after preprocessing"?
- The authors mention multiple statistical tests that were used throughout the manuscript, but the main text and figure captions never specify the test that was used in a particular situation. The authors should be clearer about which statistical test was used for each analysis.

Reviewer #4

(Remarks to the Author)

In this paper, Ou et al. performed genomic, transcriptomic, proteomic, and phosphoproteomic profiling for a cohort of samples obtained from 145 individuals with prostate cancer (PC). Based on protein expression profiles, tumor samples were stratified into three subtypes with distinct clinical and molecular features. One of them, referred to as "Subgroup II" and characterized by an increased expression of enzymes involved in sialic acid metabolism, activated AR signaling, and reduced immune cell infiltration, demonstrated the worst survival. The authors performed multiple experiments with cell lines and mouse models to explore the connection between sialic acid metabolism and immunosuppression, and prioritized NANS protein, as a biomarker candidate and a potential therapeutic target. The association of NANS protein expression with recurrence-free survival has been in two independent datasets collected at different hospitals in China.

This work is potentially of great importance for the research of PC and for patients of Chinese origin. However, because the manuscript lacks some key details essential for a full understanding of this work and for the replication of the results and figure captions are insufficient, I cannot recommend this work for publication in its current form.

Major comments:

- 1) Although one of the main outcomes of this work is the discovery of three subtypes of prostate cancer, the method for stratifying patients, as well as the creation and validation of a panel of 39 proteins, lack sufficient detail (see comments on lines 668-707 below). Also, I did not locate any links to code and raw data used in this article.
- 2) What is the relationship between mRNA-based subtypes (Fig. 1B, RNA Subgroup) and proteome-based subgroups 1-3 discovered and investigated in this work? Is the expression of NANS gene/transcript(s) associated with RFS?
- 3) The authors show that three subgroups have distinct distributions of mutations, T or N stages, GS, etc., but do not evaluate associations of these tumor and patient features with survival. Is it possible that the associations of subgroups with RFS are explained by the associations between clinical features and RFS?

4) The section Statistics (page 36, lines 803-813) lists all statistical tests used in this work, but it remains difficult for the reader to understand which statistical test was used in each specific analysis. Also, the authors frequently report p-values without mentioning any multiple testing adjustment procedure. I would recommend specifying the statistical test(s) used along with the result description and/or in figure captions.

Minor comments:

Fig. 1 B: I would recommend using the same color scales for bar plots comparing the clinicopathological features and annotation of the heatmap and separating panels with the barplots and the heatmap. Why does the upper panel show the barplots for only four out of eight features (RNA subgroup, PSA, T Stage and BCR are not shown)?

Fig. 1E. This figure shows GSEA results for HALLMARK_ANDROGEN_RESPONSE gene set. However, this gene set is not displayed in Fig. 1C (the closest match I see in Fig. 1C is the "Androgen_response" gene set). Please explain this or highlight HALLMARK_ANDROGEN_RESPONSE gene set in Fig. 1C.

page 5, lines 109-111: "Simultaneously, although metabolism subgroup I also displayed features of metabolic enrichment, however, this group patients exhibited a favorable prognosis and lower association with AR activation (Figure S1E)." - Please rephrase, because according to Figure S1E, no significant enrichment of the HALLMARK_ANDROGEN_RESPONSE gene set in the comparison Subgroup 1 vs Subgroups 2/3 is observed. For example, "lower association" -> "no statistically significant association"

Fig. 2B: the relationship between the upper and lower panels is not explained.

Fig. 2 B and C: the transparent color in bar plots is not explained.

Fig. 2 D and page 6, lines 126-128: Why does the unadjusted p-value cutoff of 0.05 was applied here? Same as in GSEA, correction for multiple testing is necessary here. What are the results of KSEA for two other subgroups?

Fig 2SA caption: shouldn't it be ssGSEA?

Fig. S2B,C,E, Fig.3A: Was correction for multiple testing performed here?

Fig. S2F: What are "SEG1-3" in color legend?

page 9, lines 159-161: "This cohort could be successfully subtyped into three proteomic subgroups by a 39-protein subtyping panel, which filtered by the Random Forest algorithm and validated in our cohort (Figure S2F)." - Please provide more details on how the panel was constructed and validated, and explain how the values shown in the heatmap in Fig. S2F were obtained.

Fig. 3C caption: "expression of N-Acetylneuraminic acid" - the term "expression" is usually applied to genes, transcripts, and proteins.

Fig. 3E: The caption does not describe the figure.

Fig. 4B: Please provide more details on how this figure was obtained (specify algorithm, its input and parameters) and what the cell color corresponds to.

lines 183-194 and Fig. 4 C-G: please introduce sgNANS in the main text and in the figure caption.

Fig. 4E: The color scale is not labeled. Not fully clear what the rows of the heatmap correspond to.

page 9, lines : "Recent research has suggested the sialoglycan-Siglec axis as a novel immune checkpoint which can be targeted to augment the anti-tumor immune response." - please add the reference.

page 9, lines 200-202: To dissect the immune microenvironment of PCa, the EcoTyper analysis was performed and revealed increased infiltration of M2 macrophages in subgroup II (Fig.4H) - please specify which cohort was used in this analysis. The same concerns Fig. 4I.

Figs. 5F,G: Please explain how exactly these figures were obtained and label the color scale.

lines 257-258: "26% of PCa cannot be classified³¹". - according to reference 31, a driver abnormality was not identified in 26% of tumors, but these tumors can still be classified based on mRNA expression.

31: "Despite this detailed molecular taxonomy of primary prostate cancers, 26% of all tumors studied appeared to be driven by still-occult molecular abnormalities or by one or more frequent alterations that co-occur with the genomically defined classes."

page 26, line 587: Why TCR-enriched libraries were used in single-cell sequencing protocol?

Typos/mistakes:

line 209: "In compared ..."

lines 212-213: "cells in the absent/present of Nans"

line 232: "to remoulade"

line 265: "In the current study, we defined PCa into three distinct subgroups"

lines 376,530: "metabonomic"

line 651: "EcoTyper were constructed"

line 605: "in default parameters"

line 610: "Nomal-panal"

line 634: "significantly different expressed genes."

line 663: "PCa", should be "PCs"

Version 1:

Reviewer comments:

Reviewer #1

(Remarks to the Author)

This Reviewer appreciates changes made by authors, including addition of substantive data, to address prior concerns. These additions have strengthened the manuscript.

Reviewer #2

(Remarks to the Author)

The authors addressed majority of the reviewers' comments; however, some issues still remain.

The analysis of proteomics data revealed three subgroups of tumors/patients, with metabolism pathways enrichment subgroup 1 and subgroup 2 and immune-related pathways subgroup 3 that was stated to have exhibited a more 'hot' overall immune profile, with significant upregulation of immune activation pathways such as pro-inflammation and lymphocyte infiltration. It was also shown that these subgroups have different survivals with group 2 being the most aggressive. However, it is hard to make a complete picture of how the proteomic based subgroups relate to immune cells infiltration and levels of NANS. The new IF data shown in figure 4H indicate immune infiltration in subgroup 1 being rather similar to subgroup 3 and low immune cell presence only in subgroup 2 which is somewhat in contrast to proteomics assignment of the groups with only subgroup 3 exhibits higher activities of immune -related pathways. These results need to be discussed in relation to groups assignment and potential biological differences in alteration of immune infiltration. It is also somewhat difficult to make association between the proteomics subgroups with NANS levels as the newly presented data (levels of NANS in adjacent normal and tumor samples in the whole cohort and in the subgroups Figure 3 and Sfigure3), show that levels of NANS are rather similar in group 1 and 2 tumors. Last but not least, the authors state that subgroup 2 exhibits higher activity of androgen receptor signaling when compared to combined subgroups 1 and 3. But is the AR signaling different between subgroups 1 and 2?

The authors addressed the potential concordance/discordance between the proteomics and transcriptomics subgroups. However; what would be of interest is to evaluate transcriptome of the proteomics subgroups.

Based on additional analysis using data from TCGA, the authors concluded that results from Chinese patients are not recapitulated in Caucasian cohort data available. However, the manuscript includes evaluation of a panel of 8 human PCa cell lines, and their stratification into three proteomic subtypes using the nearest template prediction (NTP) algorithm similarly to human PCa tissues (Fig 4B). These assignments of the subgroups do not correspond to the aggressiveness assignment to the clinical samples, as PC-3 and DU 145, aggressive prostate cancer cell lines do not fall into the aggressive category (PC-3 subgroup 1 and DU 145 subgroup 3). Similarly, it does not appear that cell lines assigned to subgroup 2 have higher AR signaling, and levels of NANS for these cell lines were not provided. In light of these results, it is not clear how to incorporate the results using 22Rv1, where NANS knockdown in 22Rv1 similarly to Myc-CaP alters sialylation. Also, the sialylated sites altered appear to be different between these two cell lines. These issues need to be addressed. In the revised manuscript, the authors introduced data with Pten/Trp53 double knockout (PtenPC^{-/-}; Trp53PC^{-/-}) mouse model, and they state that this model partially recapitulates the features of aggressive PCa. It is this reviewer's opinion that characteristics of this model, relevant to the topic of this manuscript, should be provided in more details than the statement "partially recapitulates" characteristics of aggressive cancer. For example, does this model exhibit subgroup 2 characteristics and what are the levels of NANS in these tumors?

While the point how was the cut off for assignment to NANS low and NANS high groups selected was addressed, it is not clear whether the cut off from the discovery cohort was used in the validation cohorts or whether the cut offs were calculated was each cohort independently, as the rebuttal states "the comparison of NANS expression was performed based on the entire discovery cohort or corresponding validation cohorts". If different cut off was calculated for each cohort, that would significantly decrease translational relevance of the results.

One of the previous comments was why Myc-CaP cells were used based on proteomic subtyping if the intention was to evaluate the impact of NANS on the tumor immunophenotype, and that it would be of considerable interest to evaluate the impact of NANS regardless of the subgroup. This comment was not addressed. The response was justification of why Myc-CaP was used but no additional data were provided about impact of NANS on tumor immunophenotype regardless of the subgroups.

The newly provided single cell analysis, showing three control and three Nans knockdown tumors, shows that three control tumors are rather separated into different clusters (suppl figure 5C) any explanation of this separation of control tumors?

Reviewer #3

(Remarks to the Author)

the authors responded adequately to the reviewer comments.

Reviewer #4

(Remarks to the Author)

I would like to thank the authors for revising the manuscript and for addressing most of my comments. While the manuscript has shown improvement after the revision, I still have some questions regarding the data availability and code provided to replicate the computational analysis. Since this is essential for the reproducibility and transparency of the research, I cannot recommend this manuscript for publication until these questions are resolved.

1) The authors state that:

"The raw proteomics and phosphoproteomics data generated

in this study is publicly available in the iProX database under the accession IDIPX0009562000"

"The whole exome sequencing (WES) and RNA-seq raw data that support the findings of this study have been deposited in the Genome Sequence Archive (GSA) under accession ID HRA008293".

However, I could not find either IPX0009562000 on <https://www.iprox.cn/>, or HRA008293 on <https://ngdc.cnca.ac.cn/gsa/>.

Also, I could not find any link to the raw metabolomics data used in this study. To ensure data availability, I recommend the authors provide direct links for each dataset used in this study.

"Data access can be obtained by contacting Dr. Wang [wangzr27@mail.sysu.edu.cn]"

To enable anonymous and quick peer review, I kindly ask the authors to include a password for reviewer access in the future.

2) The authors write that "the codes generated in this study are publicly available in Github at <https://github.com/Diluczhang/Proteome-of-PCA.git>."

Currently, the repository provides only the code (with minimal documentation) for Random Forest, differential protein expression analysis, NTP and for some plots. I could not find the code for other analyzes, incl. the analysis of variants, differential expression with DESeq2, metabolomics, or single-cell data.

Moreover, the code for RF model learning and feature selection found in RadomForeast/radomforest.py appears inconsistent with the analysis description in the manuscript. For example, I could not locate 10-fold CV and it seems that only a simple train-test split is implemented. Since input data used by the authors are not available, I cannot replicate this analysis and test the code directly.

I recommend the authors ensure that the code they used and provided in the repository aligns with the analysis described in the manuscript. Moreover, it is recommended to provide the readers with the code and inputs necessary to replicate all essential analysis steps.

Minor questions and corrections:

Fig. 1A: labels are difficult to read, at least 200% zoom is required.

Fig. 2D: lacks x-axis label.

Fig. S3B: the color bar lacks a label. Does it present averaged protein expressions or z-scores? In which dataset?

Typos:

"metabonomics"

"Uniformmanifold"

"Github"

Version 2:

Reviewer comments:

Reviewer #2

(Remarks to the Author)

This reviewer feels that the authors have addressed the comments provided by the reviewers in the rebuttal. However, it is also this reviewer's opinion that more of the responses from the rebuttal should have been integrated into the manuscript.

Reviewer #4

(Remarks to the Author)

I sincerely appreciate the authors' efforts in revising the manuscript and for providing links to the data and up-to-date code used in their study.

In response to Q3, the authors state: "We have provided direct links for each dataset, allowing reviewers to access them without the need for additional passwords. These datasets will be made publicly accessible to all researchers upon the publication of the article. By then, the proteomic, phosphoproteomic, and metabolomic data are publicly available for online viewing and download".

However, I was unable to find download links for the following datasets:

* PCa patients, raw metabolomics data: <https://ngdc.cnca.ac.cn/omix/preview/7s2OX3fe>

* RNA-seq matrix: <https://ngdc.cnca.ac.cn/omix/preview/xKwuJZE7>

* PCa cell lines, raw metabolomics data: <https://ngdc.cnca.ac.cn/omix/preview/J6fxWY6S>

I would appreciate it if the authors could confirm that all datasets will be publicly accessible upon the publication of the article.

Point-to-point response:

We thank all reviewers for their professional and insightful comments. We thank you for your time and effort to evaluate our prior submission. In this point-to-point response letter, *reviewer's comments were marked in dark blue italics*, followed by our point-to-point responses. All revised or supplementary new text is underlined in dark red. And the relevant Figures were marked in black **Bolds**.

Reviewer: 1

Q1. Sialic acid related enzymes- were there any other carbohydrate antigens measured beyond NANA that were differential between the groups? Additionally, were levels of NANA (or corresponding proteins) elevated in tumor compared to adjacent control tissues? It also appears that a subset of samples were used for these analyses. What is the rationale for sample selection? It would be more beneficial to evaluate all tumor tissues.

Response:

We appreciate the reviewer's kind evaluation of our work. Actually, besides NANS, there were another 2 sialic acid related enzymes (UAP1 and NPL) whose protein expression levels were significantly differential between subgroups (subgroup 2 versus both other groups). Among these three enzymes, NANS was the most highly differentially expressed protein with a fold-change of 2 (FC=2) between subgroups. While the expression differences of NPL (FC=1.63) and UAP1 (FC=1.43) were relatively lower (**Revised Figure 3B**). Meanwhile, UAP1, as the upstream regulator of sialic acid synthesis pathway, catalyzes the synthesis of UDP-N-acetylglucosamine (UDP-GlcNAc), which functions as an integration point of multiple metabolic pathways, such as the sialic acid synthesis pathway and the hexosamine biosynthetic pathway (*Oncogene*, 2015, PMID: 25241896). Therefore, targeting UAP1 may also disturb metabolites synthesis of other pathways, and possibly result in unexpected off-target in tumor therapies. By contrast, NPL, residing in the branch loop of the sialic acid synthesis pathway, catalyzes the process from the sialic acid back to N-acetylmannosamine (ManNAc) (*Sci Adv*, 2023, PMID: 37390204). Therefore, targeting NPL will cause uncertain effects on this pathway. While, NANS is defined as the key enzyme of the last step for sialic acid synthesis. Targeting NANS can directly block the sialic acid synthesis with fewer side effects. That is the reason NANS was selected as a potential therapeutic target and further focus of our study.

Besides, the comparison of NANS expression levels between tumor and adjacent control tissues was conducted only in subgroup 2 patients of the discovery cohort. With your kind suggestion, we further conducted the differential expression analysis of NANS protein between tumor and adjacent control tissues across all patients of the discovery cohort, as well as separately in subgroup 1 and in subgroup 3 patients. The results showed that expression levels of NANS were elevated in tumor compared to adjacent control tissues in the whole discovery cohort ($P=8.8e-12$, **Revised Figure S2F**). Importantly, compared to adjacent control tissues, the expression of NANS in

subgroup 2 exhibited the most significantly increase ($P=4.4e-09$ for subgroup 2, $P=0.0014$ for subgroup 1, $P=0.0041$ for subgroup 3, **Figure 3F, Revised Figure S3D-F**). And we have revised our statement as “To this end, our proteomic data firstly showed that NANS expression was significantly elevated in tumor tissues compared to paired adjacent normal tissues in the discovery cohort, especially in subgroup 2 (Figure 3F, Figure S3D-F)” in the revised manuscript. (page 9, line 186 to line 188).

B

Figure 3B. Differential expression analysis of key enzymes involved in the sialic acid synthesis pathway among three proteomic subgroups of the discovery cohort.

F

Figure 3F. Comparison of NANS proteomic abundance between tumor and paired non-tumor samples in subgroup 2 patients of the discovery cohort. The middle lines represent the median, and the lower and upper hinges denote the 25-75% IQR, with whiskers extending up to a maximum of 1.5 times IQR. P-value is determined using the two-sided Wilcoxon rank-sum test.

Revised Figure S3D. Comparison of NANS protein expression level between tumor and paired normal tissues in all patients of the discovery cohort. P-value was determined using the two-sided Wilcoxon rank-sum test.

Revised Figure S3E. Comparison of NANS protein expression level between tumor and paired normal tissues in subgroup 1 patients of the discovery cohort. P-value was determined using the two-sided Wilcoxon rank-sum test.

Revised Figure S3F. Comparison of NANS protein expression level between tumor and paired normal tissues in subgroup 3 patients of the discovery cohort. P-value was determined using the two-sided Wilcoxon rank-sum test.

Q2. For the validation, a 39-protein subtyping panel was developed using Random Forest. It is unclear if this algorithm was applied to the original dataset and then validated in the independent 119 PCa samples or whether this was developed in the independent samples. It is also a bit odd that NANS was not in the 39-protein panel, despite being a primary focal point of the study.

Response:

We thank the reviewer for raising this excellent issue. Indeed, the Random Forest algorithm was applied to the discovery cohort to develop the 39-protein subtyping panel, which was subsequently validated in independent 119 PCa samples (the validation cohort 1). And we have revised our statement as “To verify our findings, we firstly constructed a 39-protein subtyping panel, which was filtered by the differential expression analysis and the Random Forest algorithm in the discovery cohort. Then, we additionally employed 119 PCa patients from other centers (details provided in the Methods section) as the validation cohort 1. Based on the proteomic profiling data and the 39-protein subtyping panel, the validation cohort 1 could be successfully categorized into three proteomic subgroups consistent with those identified in the discovery cohort.” in the revised manuscript. (page 8, line 174 to line 180)

Further, as for the reviewer’s concern about the 39-protein subtyping panel without the target protein NANS, we attempted to incorporate NANS into the subtyping panel. After recalculating the classification performance of the 40-protein panel including NANS, we regretted to report that NANS did not significantly improve the panel's classification performance. The detailed ROC curves are showed as below. (**Response Figure 1A:** the 39-protein model, AUC=97.16%; **Response Figure 1B:** the NANS + 39-protein model, AUC=97.49%, P=0.55).

Based on the aforementioned attempts, we propose that this result may be related to the fundamental principles of the Random Forest algorithm, a machine learning technique that primarily selects biomarkers based on differences in protein expression.

NANS, however, was identified as an important regulator not only due to its significant higher expression in subgroup 2 tumor tissues, but also based on the functional enrichment analysis of molecular pathways. Its regulatory function in sialic acid metabolism and tumor progression was further validated in detailed *in vitro* and *in vivo* experiments. Collectively, although NANS was not involved in the 39-protein subtyping panel, it holds significant biological relevance in regulating tumor progressions. Similar cases of target molecules not included in the subtyping biomarker panel have also been reported in some studies (*Molecular Cell*, 2021, PMID: 34375582; *NPJ Precision Oncology*, 2024, PMID: 38245587), suggesting NANS as a potential driver instead of subtype-divided marker for PCa.

Response Figure 1A. The Receiver Operating Characteristic Curves (ROC) of the 39-protein subtyping model in the validation dataset.

Response Figure 1B. The Receiver Operating Characteristic Curves (ROC) of the 40-protein subtyping model (39-protein subtyping model plus NANS) in the validation dataset.

Q3. Subgroup II was associated with poor prognosis, accompanied by higher preoperative PSA levels, higher postoperative Gleason Score as well as higher tumor mutational burden and somatic copy number aberrations compared to subgroup III. Authors state that elevated tumor NANS was associated with reduced bRFS; however, it should be shown that NANS is an independent predictor of bRFS when considering the above-mentioned variables. This can be evaluated using multivariate cox proportional hazard models.

Response:

We appreciate the reviewer's kind suggestion and totally agree that multivariate cox proportional hazard models should be supplemented for independent predictor evaluation. To this end, the univariate and multivariate cox proportional hazard model was employed, the results showed that the protein level of NANS was an independent prognostic predictor of PCa (HR=2.51, P=0.035). And this result was added in **Revised**

Table S3 and described as “Importantly, survival analysis established a noteworthy correlation between higher NANS expression and a reduced duration of biochemical recurrence-free survival (P=0.0054) (Figure 3I, Revised Table S3).” in the revised manuscript. (page 9, line 191 to line 193).

Revised Table S3			
Variables	HR	95%CI	P-values
TMB	1.12	(0.60,2.09)	0.715
SCNA burden	0.99	(0.93,1.05)	0.687
Age (years)	0.98	(0.93,1.03)	0.354
PSA(10ng/ml)	1.03	(1.00,1.05)	0.042
Gleason score			
<=7	1.00		
>=8	2.49	(1.17,5.30)	0.018
T-stage			
pT2	1.00		
pT3	1.64	(0.79,3.42)	0.184
N-stage			
pN0	1.00		
pN1	1.38	(0.56,3.41)	0.487
NANS group			
NANS low	1.00		
NANS high	2.51	(1.06,5.93)	0.035

Revised Table S3. The multivariable cox regression for the biochemical recurrence-free survival (including NANS expression level) in the discovery cohort.

Q4. NANS high vs low classifier. What was the criteria for NANS High vs Low described in Figure 3? Additionally, was this based on the entire specimen set or restricted to the subgroup II samples?

Response:

We thank the reviewer’s professional comment and regret to make you confused. For **Figure 3I-K**, we determined the optimal cut-off point for NANS expression with the maximally selected rank statistics from the 'maxstat' R package. With that, we stratified patients into NANS-high and NANS-low groups. Besides, the comparison of NANS expression was performed based on the entire discovery cohort or corresponding validation cohorts. To clarify, we have added the description in the Methods part as “The optimal cut-off point determined by the the 'maxstat' R package (maximally selected rank statistics) was used to explore the prognostic efficacy of a single protein.” in the revised manuscript. (page 36, line 809 to line 811). And we have also adjusted the figure legend of **Figure 3I-K** as follows:

Figure 3I. Kaplan-Meier curves showing the bRFS of patients in the discovery cohort grouped by the proteomic abundance of NANS. P-value is calculated by the Log-rank test.

Figure 3J. Kaplan-Meier curves showing the bRFS of patients in the validation cohort 1 grouped by the proteomic abundance of NANS. P-value is calculated by the Log-rank test.

Figure 3K. Kaplan-Meier curves showing the bRFS of patients in the validation cohort 2 grouped by the immunostaining intensity of NANS in the tumor region. P-value is calculated by the Log-rank test.

Q5. It is a bit unclear why Myc-CaP cells were used based on proteomic subtyping if the intention is to evaluate the impact of NANS on the tumor immunophenotype. It would be of considerable interest to evaluate the impact of NANS regardless of the subgroup.

Response:

Thank you for your valuable suggestion. Based on the following considerations, we chose Myc-Cap cells to assess the impact of NANS on the immunophenotype: 1) due to study the impact of NANS on tumor immunophenotype *in vivo*, we thus chose a mouse-derived PCa cell line, such as Myc-Cap; 2) According to the 39-protein subtyping panel, 8 PCa cell lines were classified into three proteomic subgroups based on the NTP algorithm. Among them, the Myc-Cap cell line belonged to subgroup 2 which was defined as the aggressive subgroup with the worst prognosis; 3) To further verify our results, we also included the *in vitro* experiments with another human-derived 22Rv1 cells which was also defined as the aggressive subgroup 2 in the revised manuscript. Similar trends were observed in these two diverse PCa cell lines (**Revised Figure 4 C-F, Revised Figure S4 A-B**).

Revised Figure 4C. Knockout (KO) of *NANS* in 22Rv1 and Myc-CaP cells via single guide RNA (sgRNA) as determined by qRT-PCR. Empty vector is used as control. P-values are determined using two-tailed Student's t test.

Revised Figure 4D. KO of *NANS* in 22Rv1 and Myc-CaP cells via sgRNA as determined by Western blot. Empty vector is used as control. P-values are determined using two-tailed Student's t test.

Revised Figure 4E. Comparison of the relative abundance of N-Acetylneuraminic acid in 22Rv1 and Myc-CaP cells between sg*NANS* and control group based on the targeted metabolomics analysis. The middle lines represent the median, and the lower and upper hinges denote the 25-75% IQR, with whiskers extending up to a maximum of 1.5 times IQR. $n=5$ for each group. P-values are determined using the two-sided Wilcoxon rank-sum test.

Revised Figure 4F. The heatmap showing differentially expressed sialylated sites in 22Rv1 and Myc-CaP cells between sg*NANS* and control group.

Revised Figure S4A. CCK8 assays for 22Rv1 and Myc-CaP cells of sgNANS and control group. P-values are determined using two-way ANOVA test.

Revised Figure S4B. Colony-forming assays for 22Rv1 and Myc-CaP cells of sgNANS and control group. Data are presented as mean \pm Standard deviation (SD). P-values are determined using two-tailed Student's t test.

Q6. Figure 4E- for sgNANS- was this based on one of the two clones or both? Additionally, both parental control as well as sgControl should be used when comparing against sgNANS cells. Moreover, it is recommended that authors replicate key experiments using at least one additional PCa model system.

Response:

Thank you for your kind suggestion. We fully agree with your suggestion that both sgNans group should be included in the experiments of **Figure 4E**. Therefore, we added the detection results of sialylation sites in Myc-CaP cells of another sgNans group and updated the new version of heatmap comparing the sialylation level among the control group and two sgNans group (**Revised Figure 4F**). Besides, based on your suggestion, we also replicated a series of key experiments in another human-derived PCa cell line, 22Rv1 cells, which also belongs to the aggressive subgroup 2 (**Revised Figure 4C-F**, **Revised Figure S4A-B**). Similar trends were observed in two different PCa cell lines, which together demonstrated the role of NANS in regulating the synthesis of sialic acid of PCa.

Meanwhile, to further verify the function of NANS as a therapeutic target of aggressive PCa, we additionally constructed the *Pten/Trp53* double knockout (*Pten*^{PC-/-}; *Trp53*^{PC-/-}) mouse model, which partially recapitulates the features of aggressive PCa. After administration of adeno-associated virus (AAV) based *Nans*

depletion, we observed the reduction of both sialylation level and M2 macrophages infiltration in mouse tumor tissues (**Fig. S6A-B**). In addition, tumor volume and weights were significantly decreased in *Pten*^{PC-/-}; *Trp53*^{PC-/-} mouse model after Nans depletion (**Fig. S6C**). These results in two diverse PCa mouse models together indicate that elevated NANS in aggressive PCa could promote sialylatic acid synthesis and further sialylation to remodel tumor immunoenvironment, resulting in PCa progression.

Revised Figure 4C. Knockout (KO) of *NANS* in 22Rv1 and Myc-CaP cells via single guide RNA (sgRNA) as determined by qRT-PCR. Empty vector is used as control. P-values are determined using two-tailed Student's t test.

Revised Figure 4D. KO of *NANS* in 22Rv1 and Myc-CaP cells via sgRNA as determined by Western blot. Empty vector is used as control. P-values are determined using two-tailed Student's t test.

Revised Figure 4E. Comparison of the relative abundance of N-Acetylneuraminic acid in 22Rv1 and Myc-CaP cells between sg*NANS* and control group based on the targeted metabolomics analysis. The middle lines represent the median, and the lower and upper hinges denote the 25-75% IQR, with whiskers extending up to a maximum of 1.5 times IQR. $n=5$ for each group. P-values are determined using the two-sided Wilcoxon rank-sum test.

Revised Figure 4F. The heatmap showing differentially expressed sialylated sites in 22Rv1 and Myc-CaP cells between sg*NANS* and control group.

Revised Figure S4A. CCK8 assays for 22Rv1 and Myc-CaP cells of *sgNANS* and control group. P-values are determined using two-way ANOVA test.

Revised Figure S4B. Colony-forming assays for 22Rv1 and Myc-CaP cells of *sgNANS* and control group. Data are presented as mean \pm Standard deviation (SD). P-values are determined using two-tailed Student's t test.

Revised Figure S6A. IHC staining and quantification for Nans in murine tumor tissues of the control and *Nans* knockout (KO) group in *Pten*^{PC-/-}; *Trp53*^{PC-/-} mouse model. n=5 for each group. The scale bar represents 100um. Data are presented as mean ± SD. P-values are determined using two-tailed Student's t test.

Revised Figure S6B. IF staining and quantification of sialylation level (SNA/MALII) and immune cells in murine tumor tissues of the control and *Nans* KO group in *Pten*^{PC-/-}; *Trp53*^{PC-/-} mouse model. n=3 for each group. The scale bar represents 50um. Data are presented as mean ± SD. P-values are determined using two-tailed Student's t test.

Revised Figure S6C. Comparison of tumor volume and weights between the control and *Nans* KO group. n=5 for each group. P-values are determined using two-tailed Student's t test.

Q7. The sample size for Figure 4I should be increased. This is based on an N = 3 and it is unclear how these samples from the discovery cohort were chosen.

Response:

Thank you for your kind suggestion. In order to enhance the reliability of our study results, we have increased the sample size of immunofluorescence (IF) staining to 30 in total (10 cases per subgroup). In that case, we found that our previous results still remained reliable. We updated **Revised Figure 4I** with the new quantification results of 10 cases per subgroup. Besides, we also uploaded all the IF staining figures of 30 patients to the Source Data.

Revised Figure 4H. Multiplex immunofluorescence (IF) staining and quantification of immune cells in three proteomic subgroups of the discovery cohort. n=10 for each group. The scale bar represents 50um. Data are presented as mean \pm SD. P-values are determined using two-tailed Student's t test.

Q8. Figure 5C- the reported p-values do not reflect the bar plots which show considerable variability and are based on an N= 3.

Response:

We thank the reviewer for bringing this issue to our attention, and we regret that these were not made clear in our prior submission. Given the limited sample size of scRNA-seq data, in the analysis of Figure 5C, we did not compare the observed cell numbers between two groups directly. According to the previous studies (*Cell*, 2021, PMID: 33545035; *Cell*, 2021, PMID: 33657410; *Nature*, 2018, PMID: 30479382), we actually compared the Ro/e (the ratio of observed to expected cell numbers) between these two groups and calculated the p-values of Chi-square test. We regret to make you confused about the original version of Figure 5C. To clarify, we adjusted the format of Figure 5C (**Revised Figure 5E**) as follows. Besides, we also adjusted the figure legends of Figure 5C (**Revised Figure 5E**) as "Comparison of the Ro/e (the ratio of observed to expected cell numbers) in major cell types before and after *Nans* knockout. P-values are determined using the Chi-square test."(page 53, line1210 to line1211)

Revised Figure 5E. Comparison of Ro/e (the ratio of observed to expected cell numbers) in all cell types before and after *Nans* knockout. P-values are determined using the Chi-square test.

Q9. Figure 5I- please show all mice rather than a single representative picture. For statistical analyses, what type of test was used? This information should be provided. This comments holds true for all relevant figures.

Response:

Thank you for your suggestion. Following your advice, we have updated the images of all mice for revised **Revised Figure 5K**. Besides, P-values in Figure 5I were determined using two-tailed Student's t test and we also updated the specific statistical test method in the figure legend of all relevant figures. We adjusted the figure legend of **Revised Figure 5K** and **Revised Figure 5L** as follows:

Revised Figure 5K. Ex vivo imaging of murine tumor tissues of the *sgNans* and control group. n=5 for each group. P-values were determined using two-tailed Student's t test.

Revised Figure 5L. Comparison of tumor volume and weights between *sgNans* and control group. n=5 for each group. P-values are determined using two-tailed Student's t test.

Minor

Descriptive statistics (e.g. mean, stdev, fold-change, pairwise analyses, etc) for the 9,252 proteins from the 145 tumors and 141 adjacent normal samples should be included as a supplemental datafile.

Response:

Thank you for your suggestion. Following your advice, we have included the detailed

descriptive statistics of all proteins from the 145 tumors and 141 adjacent normal samples (including mean, SD, fold-change, pairwise analyses) in the Source Data.

Reviewer #2

Q1. The manuscript mentions the use of six tumors for scRNA-Seq analysis in Figure 5A, but it's unclear whether these tumors represent control and NANS knockdown groups.

Response:

Thank you for your kind suggestion. We are sorry to make you confused. Following your advice, we have updated the UMAP visualization of 29515 single cells, (left panel) colored by sample origin, (right panel) colored by group origin (control group or *Nans* KO group) in **Revised Figure S5C**.

Revised Figure S5C. UMAP visualization of 29515 single cells, (left panel) colored by sample origin (right panel) colored by group origin (control group or *Nans* KO group).

Q2. Furthermore, the method by which alterations in the immune microenvironment were determined needs more details, especially since only a single UMAP was included. Moreover, it would be beneficial to include tumor cells in the single-cell RNA sequencing to demonstrate any altered tumor cells phenotype.

Response:

Thank you for your kind suggestion. Based on the following considerations, we chose to focus on alterations of macrophages in the immune microenvironment: 1) In **Figure 4G**, the EcoTyper analysis was used to dissect the tumor immune microenvironment of PCa, which indicating the increased infiltration of macrophages in tumor tissues of subgroup 2 patients with the poorest prognosis. The IF staining in **Figure 4H** further confirmed the increased infiltration of M2-macrophages in tumor tissues of subgroup 2 patients. 2) The scRNA-seq data of tumor tissues from the orthotopic transplanted mouse models revealed a notable reduction in the proportion of macrophages after *Nans* knockout. Moreover, among four subsets of macrophages, only the proportion of Lyve1+ macrophages which displaying M2-like immunosuppressive features underwent a conspicuous decrease after *Nans* knockout. 3) NANS is the key enzyme for sialic acid synthesis. Up-regulation of sialic acid-containing glycans, termed hypersialylation, is a common feature of human malignancies (*Glycobiology*, 2018,

PMID: 29309569; *Frontiers in immunology*, 2021, PMID: 34966391). Growing evidence on sialoglycan-Siglec axis between tumor cells and macrophages have been reported to drive tumor progression and immune evasion in malignancies, such as breast cancer and pancreatic cancer (*Nature communications*, 2021, PMID: 33627655; *Sci Transl Med*, 2022, PMID: 36322632). Therefore, based on these literature and experimental findings, we chose to focus on alterations of macrophages in the immune microenvironment.

Besides, based on your kind suggestion, we have included tumor cells in the scRNA-seq analysis. We identified 4 major subclusters of epithelial cells, including the Luminal C1-3 subclusters with high signal for luminal signature genes and the Cell Cycle subcluster with high signal for cell cycle-related genes (**Revised Figure 5B, Revised Figure S5E-F**). After *Nans* depletion, we observed a significant reduction of Luminal C1 epithelial cells, which displaying the activation of both malignant progression-related pathways and sialic acid-related pathways (**Revised Figure 5C-D**). These results based on scRNA-seq data indicated the altered phenotype of tumor cells after *Nans* depletion.

G

Figure 4G. EcoTyper analysis dissecting the tumor immune microenvironment of patients in three proteomic subgroups of the discovery cohort.

Figure 4H. Multiplex immunofluorescence (IF) staining and quantification of immune cells in three proteomic subgroups of the discovery cohort. $n=10$ for each group. The scale bar represents 50 μ m. Data are presented as mean \pm SD. P-values are determined using two-tailed Student's t test.

Revised Figure 5B. UMAP visualization of epithelial cells, colored by cell type annotations.

Revised Figure 5C. Comparison of the Ro/e (the ratio of observed to expected cell numbers) in 4 epithelial cell subtypes before and after Nans knockout. P-values are determined using the Chi-square test.

Revised Figure 5D. Bubble plot showing differentially activated hallmark pathways and sialic acid-related pathway among 4 epithelial cell subtypes.

Revised Figure S5E. UMAP visualization of epithelial cells, colored by group origin (control group or Nans KO group).

Revised Figure S5F. Marker gene expression for each epithelial cell type, where dot size and color represent percentage of marker gene expression (pct. exp) and the averaged scaled expression (avg. exp. scale) value, respectively.

Q3. Additionally, the statistical significance in Figure 5C, where only two or three data points are shown for each condition, needs clarification, especially considering the reported p-values reaching levels of e-9.

Response:

We thank the reviewer for bringing this issue to our attention, and we regret that these were not made clear in our prior submission. Given the limited sample size of scRNA-seq data, in the analysis of **Figure 5C**, we did not compare the observed cell numbers between two groups directly. According to the previous studies (*Cell*, 2021, PMID: 33545035; *Cell*, 2021, PMID: 33657410; *Nature*, 2018, PMID: 30479382), we actually compared the Ro/e (the ratio of observed to expected cell numbers) between these two groups and calculated the p-values of Chi-square test. We regret to make you confused about the original version of **Figure 5C**. To clarify, we adjusted the format of **Figure 5C** (**Revised Figure 5E**) as follows. Besides, we also adjusted the figure legends of

Revised Figure 5E as “Comparison of the Ro/e (the ratio of observed to expected cell numbers) in major cell subsets before and after *Nans* knockout. P-values are determined using the Chi-square test.” (page 53, line 1210 to line 1211)

Revised Figure 5E. Comparison of the Ro/e (the ratio of observed to expected cell numbers) in major cell subsets before and after *Nans* knockout. P-values are determined using the Chi-square test.

Q4. Further analyses of the generated tumors are also needed. For instance, assessing the expression of NANS in vivo and determining whether alterations in NANS result in changes to tumor subgroup assignments after knockdown.

Response:

We thank the reviewer’s professional comment and provide additional analyses of the generated mouse tumors based on the reviewer’s suggestion. First of all, we performed the IHC staining of NANS in mouse tumor tissue slices of the *Nans* knockout group and the control group. The staining results confirmed that the expression level of NANS was significantly lower in the tumors of the *Nans* knockout group than that of the control group (**Revised Figure S5A**). Secondly, we collected 6 tumor samples of orthotopic transplanted mouse model constructed with Myc-CaP cells, which belongs to the aggressive subgroup 2, for proteomic profiling (3 biological replicates for the control group and the *Nans* knockout group). Based on the 39-protein subtyping panel from our discovery cohort, 6 tumor samples were divided into diverse proteomic subgroups by the Nearest template prediction (NTP) algorithm. Among them, 3 tumor samples from the control group were divided into subgroup 2, while 2 tumor samples from the *Nans* knockout group were shifted into subgroup 3 and another one was shifted into subgroup 1 (**Revised Figure S5B**). Therefore, based on the proteomic data, *Nans* knockout indeed changed the tumor subgroup demonstrate the role of NANS in altering the tumor microenvironment in PCa.

Revised Figure S5A. IHC staining and quantification for Nans in tumor regions of orthotopic transplanted mouse models from the control group and *sgNans* group. n=5 for each group. The scale bar represents 100um. Data are presented as mean \pm SD. P-values are determined using two-tailed Student's t test.

Revised Figure S5B. Heatmap visualizes the proteomic subtype assignments of murine tumor tissues of the control and *Nans* knockout (KO) group in orthotopic transplanted mouse models based on the NTP algorithm. Three biological replicates for each group.

Q5. Unfortunately, the effects of NANS knockdown were demonstrated in a single model. Additional transgenic models should be evaluated to strengthen the evidence supporting the role of NANS in altering immune cell infiltration and promoting tumor progression in vivo.

Response:

Thank you for your insightful suggestion. In the original manuscript, we have constructed the orthotopic transplanted mouse models of PCa with Myc-CaP cells to assess the role of NANS in regulating the tumor immune microenvironment and PCa progression. We fully agree with your suggestion that one more mouse model is necessary to strengthen the evidence. Therefore, we conducted additional series of experiments using the *Pten*^{PC-/-} ; *Trp53*^{PC-/-} GEMM mouse model, which partially recapitulates the features of aggressive PCa. After administration of adeno-associated virus (AAV) based *Nans* depletion, we observed the reduction of both sialylation level and M2 macrophages infiltration in mouse tumor tissues (**Revised Figure S6A-B**). In addition, tumor volume and weights were significantly decreased in *Pten*^{PC-/-} ; *Trp53*^{PC-/-} mouse model after *Nans* depletion (**Revised Figure S6C**). These results in two diverse PCa mouse models together indicate that elevated NANS in aggressive PCa

could promote sialylatic acid synthesis and further sialylation to remodel tumor immunoenvironment, resulting in PCa progression.

Revised Figure S6A. IHC staining and quantification for Nans in murine tumor tissues of the control and *Nans* knockout (KO) group in *Pten*^{PC-/-}; *Trp53*^{PC-/-} mouse model. n=5 for each group. The scale bar represents 100um. Data are presented as mean ± SD. P-values are determined using two-tailed Student's t test.

Revised Figure S6B. IF staining and quantification of sialylation level (SNA/MALII) and immune cells in murine tumor tissues of the control and *Nans* KO group in *Pten*^{PC-/-}; *Trp53*^{PC-/-} mouse model. n=3 for each group. The scale bar represents 50um. Data are presented as mean ± SD. P-values are determined using two-tailed Student's t test.

Revised Figure S6C. Comparison of tumor volume and weights between the control and *Nans* KO group. n=5 for each group. P-values are determined using two-tailed Student's t test.

Additional Comments

Q6. The authors identified three different subgroups of patients based on their proteomic analysis, with subgroup II having more aggressive disease and worse survival. Comparison with the existing parameters or combination of the new subgrouping with clinical parameters of existing methods to determine the aggressiveness of the disease and survival benefit differences to demonstrate potential clinical use for all three cohorts used should be included.

Response:

We appreciate the reviewer’s professional advice and totally agree with that clinical parameters should be taken into account when assessing the prognostic role of proteomic subgroups. Based on your suggestion, we added the analysis of univariate and multivariate cox proportional hazard models for independent prognostic predictor evaluation. The results confirmed that the proteomic subgroup 2 was an independent prognostic predictor of PCa (HR=2.53, P=0.021). And this result was added in **Revised Table S2** and described as “Even taking these clinical factors into account, the proteomic subgroup 2 remained an independent prognostic predictor of PCa (Revised Table S2).” in the revised manuscript. (page 5, line 111 to line 113)

Revised Table S2

Variables	HR	95%CI	P-values
Age (years)	0.96	(0.92,1.01)	0.132
PSA (10ng/ml)	1.03	(1.00,1.05)	0.027
Gleason score			
<=7	1.00		
>=8	2.35	(1.11,5.01)	0.026
T-stage			
pT2	1.00		
pT3	1.63	(0.81,3.30)	0.172
N-stage			
pN0	1.00		
pN1	1.25	(0.52,3.03)	0.621
Proteomic subgroups			
Subgroup 3	1.00		
Subgroup 2	2.53	(1.15,5.58)	0.021
Subgroup 1	1.91	(0.75,4.88)	0.174

Revised Table S2. The multivariable cox regression for the biochemical recurrence-free survival (including proteomic subgroups) in the discovery cohort.

Q7. While the authors evaluated the Chinese patient cohort it would be of interest to include an analysis of TCGA, to determine whether these specific characteristics are unique for Chinese patients only.

Response:

Thanks for your constructive comments. Actually, we have tried to perform the analysis of TCGA database at the beginning. But in the TCGA database, the protein expression data of prostate cancer only include 192 proteins measured by reverse-phase protein array (RPPA). In our study, we have identified more than 8000 proteins using data-independent acquisition (DIA) based proteomic technology. Unfortunately, there is no overlap between our 39-protein subtyping panel and 192 TCGA proteins. Therefore, it is difficult to validate our results in TCGA database due to the gap of different proteomic technology. Nevertheless, we attempted to validate our protein subgroups in a reported Caucasian cohort (*Cancer cell*, 2019, PMID: 30889379), revealing that the

Caucasian PCa patients could also be categorized into three subgroups using our 39-protein subtyping panel. Notably, patients in subgroup 3 of the Caucasian cohort exhibited the poorest prognosis, which was different from findings in the Chinese cohort. However, the interpretation of different results in two cohorts from diverse races is limited by different proteomic techniques. Further real-world data with strict quality control is warranted to answer the question whether these molecular *characteristics are* exclusive to Chinese patients. And actually we are now prospectively collecting tumor samples in multiple centers from different countries for further investigation.

Q8. The authors cite a paper on proteomic analysis of Caucasian patients that resulted in the identification of four protein groups and five patients' groups, with protein groups having increased immune signature and two of the patient's subgroups having worse outcome. Cross-analysis of these existing data with the herein reported data would be of interest.

Response:

Thanks for your constructive comments. Following this reviewer's recommendation, we performed the following analyses to confirm the proteomic subgroups in the cohort of the Caucasian PCa patients (*Cancer cell*, 2019, PMID: 30889379). Based on the 39-protein subtyping panel from our discovery cohort, the Caucasian PCa cohort was divided into three proteomic subgroups by the Nearest template prediction (NTP) algorithm. However, in the Caucasian cohort, PCa patients in subgroup 3 had the worst prognosis, which was different from the Chinese cohort (**Response Figure 2**). We think there are three major reasons accounting for the prognostic difference of proteomic subgroups between the Chinese and Caucasian cohorts. Firstly, the protein expression profiles between two cohorts were different due to diverse proteomic techniques. In the Caucasian cohort, the proteomic data was acquired by the data dependent mode. While in our Chinese cohort, the proteomic data was acquired by the data independent mode. There was only 23/39 overlap of the 39-protein subtyping panel between two cohorts, which may lead to the instability of the subtyping results. Secondly, the Caucasian cohort only included PCa patients in the intermediate-risk group while our Chinese cohort included PCa patients in all risk groups with 70% in the high-risk group. Patients in the high-risk group have higher biochemical recurrence rates than patients in the intermediate-risk group. Thirdly, the racial difference may directly lead to the different protein expression profiles of two cohorts. Genomic analyses have shown significant different prevalence of key gene mutations in PCa among patients from different races. For example, 41% of Chinese men with localized PCa contained mutations in FOXA1, whereas only 3.5% of localized PCa from the US cohort (TCGA) (*Nature reviews Urology*, 2021, PMID: 33692499). Besides, the protein expression profiles of PCa are also diverse among different ethnicities. A study calculated the weighted ERG oncoprotein frequencies across men of different ethnicities with prostate cancer. This study reported that 13–22% of men in the East Asian population who were diagnosed with prostate cancer had ERG+ tumors, whereas 43–53% of tumors were ERG+ in patients of European descent from Europe and the USA (*Nature reviews Urology*, 2018, PMID: 28872154). Therefore, the diverse races, the different proteomic

techniques and population composition may all account for the prognostic difference of proteomic subgroups between two cohorts. Even so, our study remains important for revealing the proteomic features of PCa, as we show an intact proteomic landscape in a huge Chinese cohort. It is useful for researchers to integrate and cross analyze proteomic data all over the world. In the future, we intend to further interrogate powerful signature proteins based on proteomic data from diverse races.

Response Figure 2. Kaplan-Meier curves of three proteomic subgroups comparing the biochemical recurrence-free survival (bRFS) in the Caucasian cohort. P-value was calculated by Log-rank test.

Q9. The authors performed an analysis of the phosphoproteome of the tumor samples but this analysis and interpretation of its results are not incorporated well and sufficiently discussed in the manuscript.

Response:

Thank for your critical comments. In the original manuscript, we only presented the results of kinase-substrate enrichment analysis (KSEA) in subgroup 2 patients. Based on your suggestion, we provide supplementary analysis of the phosphoproteome (KSEA) in subgroup 1 and 3 patients (**Figure S2A-B**). The detailed description of the KSEA analysis are added and described as “Meanwhile, the kinase PRKD3 associated with PCa progression was significantly enriched in subgroup 1 (Figure S2A), in consistent with the finding that expression of PRKD3 contributes to mast cell infiltration and angiogenesis in the tumor microenvironment of PCa (Journal of molecular medicine, 2023, PMID: 36843036). By contrast, the atypical inflammatory kinase IKBKE was significantly enriched in the immune subgroup 3 (Nature immunology, 2003, PMID: 12692549; Science, 2003, PMID: 12702806) (Figure S2B), which has been reported to promote PCa tumor growth through modulating the Hippo pathway (Nucleic acids research, 2020, PMID: 32324216). Therefore, the molecular features in the phosphoproteomic level are potentially associated with its proteomic subgroups.” in the Result part (page 7, line 137 to line 144) and as “Protein kinases and phosphatases play crucial roles in the reversible phosphorylation-dephosphorylation cycle, regulating tumor development. The progression of PCa may be attributed to abnormal activation or overexpression of kinases. For instance, excessive activation of pathways like the

mitogen-activated protein kinase (MAPK) and the nuclear factor- κ B (NF- κ B) pathway promotes the progression of PCa (*Clinical cancer research*, 2009, PMID: 19638457; *Cancer research*, 2019, PMID: 30952632; *Cancer research*, 2014, PMID: 24686169). Recently, PIM kinases have also been identified as key oncogenic drivers in PCa (*Signal Transduct Target Ther*, 2020, PMID: 32296034). Increasingly, inhibitors targeting phosphorylated kinases are being applied in the treatment of advanced prostate cancer patients (*Trends Cancer*, 2024, PMID: 38341319). In the present study, the enrichment of AR-related kinases and immune-related kinases in different subgroups respectively correlates with their proteomic features, potentially aiding in the risk stratification of PCa patients” in the Discussion part of revised manuscript. (page 14, line 308 to line 316).

Figure S2A. Kinase-Substrate Enrichment Analysis (KSEA) showed significantly up-regulated and down-regulated kinases in subgroup 1. Red bars refer to FDR< 0.05 and z score > 0, blue bars refer to FDR< 0.05 and z score \leq 0.

Figure S2B. Kinase-Substrate Enrichment Analysis (KSEA) showed significantly up-regulated and down-regulated kinases in subgroup 3. Red bars refer to FDR< 0.05 and z score > 0, blue bars refer to FDR< 0.05 and z score \leq 0.

Q10. The proteomic/ transcriptomic analyses of patients' tumor samples suggested alteration of immune infiltration and analysis of in vivo MycCaP tumors indicated NANS involvement, specifically M2 macrophages, IHC for M2 presence and NANS expression in patients' tumor samples should be performed and quantified to provide additional support for the conclusions.

Response:

Thank you for your valuable suggestion. Actually, in the original manuscript, we presented the IHC staining results of NANS in the discovery cohort and found significantly higher expression of NANS in subgroup 2 (**Figure 3G**). Following your advice, we also supplemented the IHC staining of M2 macrophage (CD163) in all patients of the discovery cohort. Similarly, we observed that consistent with our *in vivo* experimental findings, there was a significantly increased infiltration of M2 macrophage in tumor tissues from subgroup 2 patients (**Response Figure 3**). This further enhances the reliability of our research results.

Figure 3G. Immunohistochemistry (IHC) staining and quantification for NANS in tumor regions of three proteomic subgroups in the discovery cohort (n=145). The scale bar represents 100µm. Data are presented as mean \pm Standard deviation (SD). P-values are determined using two-tailed Student's t test.

Response Figure 3. Immunohistochemistry (IHC) staining and quantification for CD163 among three proteomic subgroups in the discovery cohort (n=145). The scale bar represents 100µm. P-values are determined using two-tailed Student's t test.

Minor Comments

Q11. Clinical characteristics of the patients in the second and third clinical cohorts should be included.

Response:

Thank you for your valuable suggestion. Following the reviewer's recommendation, we have added the detailed clinical information of the second and third clinical cohorts in the Source Data.

Q12. I suggest an inclusion of the number of patients and percentage in Figures

(e.g., 1C).

The coloring in Figure 2B is not clear.

Response:

Thank you for your valuable suggestion. Following the reviewer's recommendation, we have adjusted the color and format of **Revised Figure 2B** and **Revised Figure 2C** as follows:

Revised Figure 2B. Bar plot comparing the *FOXA1* mutation frequency among three proteomic subgroups of the discovery cohort. Numbers at the top of the bar represent the counts of each subgroup, with the corresponding proportions at the bottom of the bar. P-values are determined using one-way ANOVA test.

Revised Figure 2C. Bar plots comparing frequency of specific SCNAs and gene fusions among three proteomic subgroups of the discovery cohort. Numbers at the top of the bar represent the counts of each subgroup, with the corresponding proportions at the bottom of the bar. P-values are determined using one-way ANOVA test.

Q13. Statistics for the results summarized in the following statement should be included. "subgroup II exhibited a higher prevalence of tumor suppressor gene deletions (such as RB1, 122 CHD1 and ZNF292) and AR activation-related amplification (GATA2) and deletions (MAP3K7, 123 SPOPL) (Figure 2C). While, gene fusion analysis revealed

that patients harboring the *TMPRSS2-124 ERG (T-E)* gene fusion, involved in lipid metabolism functions in PCa22, belonging to the metabolism subgroup I (Figure 2C).

Response:

Thank you for your valuable suggestion. Following the reviewer’s recommendation, we have adjusted the format of **Revised Figure 2C** and included the detailed statistics data of each bar plot as follows:

Revised Figure 2C. Prevalence of specific SCNAs and gene fusions in three proteomic subgroups of the discovery cohort (top panel). NA represents not available. Bar plots comparing frequency of specific SCNAs and gene fusions among three proteomic subgroups of the discovery cohort (bottom panel). Numbers at the top of the bar represent the counts of each subgroup, with the corresponding proportions at the bottom of the bar. P-values are determined using one-way ANOVA test.

Q14. The gene list in the 39-gene panel used should be provided.

Response:

Following this reviewer’s recommendation, we have added the detailed gene list of the 39-gene panel in the Source Data.

Q15. All data points should be plotted in all bar graphs and P value reported in a consistent fashion (as of now “” and number are used).*

Response:

Thank you for your valuable feedback and we regret these confusing presentations of

graphs and data. Based on your suggestions, we have adjusted all bar graphs to box plots with all data points, and the P values have been reported in the consistent fashion of specific values.

Q16. The authors used low NANS and High NANS to evaluate the potential survival effects of NANS expression. No information was included how was the threshold for “low” and ‘high” levels determined?

Response:

We thank the reviewer’s professional comment and regret to make you confused. For **Figure 3I-K**, we determined the optimal cut-off point for NANS expression with the maximally selected rank statistics from the 'maxstat' R package. With that, we stratified patients into NANS-high and NANS-low groups. Besides, the comparison of NANS expression was performed based on the entire discovery cohort or corresponding validation cohorts. To clarify, we have added the description in the Methods part as “The optimal cut-off point determined by the the 'maxstat' R package (maximally selected rank statistics) was used to explore the prognostic efficacy of a single protein.” in the revised manuscript. (page 36, line 809 to line 811). And we have also adjusted the figure legend of **Figure 3I-K** as follows:

Figure 3I. Kaplan-Meier curves showing the bRFS of patients in the discovery cohort grouped by the proteomic abundance of NANS. P-value is calculated by the Log-rank test.

Figure 3J. Kaplan-Meier curves showing the bRFS of patients in the validation cohort 1 grouped by the proteomic abundance of NANS. P-value is calculated by the Log-rank test.

Figure 3K. Kaplan-Meier curves showing the bRFS of patients in the validation cohort 2 grouped by the immunostaining intensity of NANS in the tumor region. P-value is calculated by the Log-rank test.

Reviewer #3

Q1. The manuscript is rich of omic data however RNA-seq received very little attention in the manuscript. This is an unfortunate, as various transcriptomic subtypes of prostate cancer have been defined and well characterized, including luminal and basal subtypes. It would be interesting to see how the proteomic subgroups defined in this manuscript relate to transcriptomic subtypes, and not doing so is a major weakness of this

manuscript. The proteomic heatmap figure even includes an “RNA subgroup” annotation, but this is not mentioned anywhere in the manuscript.

Response:

Thanks for your insightful comments. We fully agree with the reviewer that it is necessary to perform the transcriptomic subtyping analysis and compare different transcriptomic subgroups and proteomic subgroups defined in our manuscript. First of all, we performed the NMF unsupervised clustering using the transcriptomic data in the discovery cohort. The discovery cohort was divided into three transcriptomic subgroups. There was 15% overlap between the NMF transcriptomic and the proteomic subgroups (**Revised Figure 1C**). Based on the reviewer’s recommendation, we also performed the PAM50 transcriptomic subtyping, which divided the cohort into the basal, luminal A and luminal B subtypes (**Revised Figure 1C**). There was 18% overlap between the PAM50 transcriptomic and the proteomic subgroups, indicating the relatively low overlap rate between the transcriptomic and proteomic subtyping in our discovery cohort. In order to comprehensively illustrate this question, we also performed the literature search. We found that the overlap rate between the transcriptomic the proteomic subtyping ranged from 0.2-0.8 in different malignancies (*Cancer cell*, 2019, PMID: 30889379; *Cell*, 2019, PMID: 31585088; *J Hematol Oncol*, 2022, PMID: 35659036; *J Hematol Oncol*, 2022, PMID: 36434634). Besides, in another multiomic study of *Cancer Cell*, the correlation coefficient between transcriptome and proteome data in PCa is only 0.2, which is similar to our study (*Cancer cell*, 2019, PMID: 30889379). Therefore, there is limited correlation between transcriptome and proteome in PCa, and exploration of the protein levels of PCa is of great importance and may provide another layer clinical implications.

We also included the detailed transcriptomic subtyping analysis and described as “By contrast, these patients were divided into three distinct subgroups at the transcriptomic level using the NMF method and PAM50 signatures respectively, with mild overlap (15%-18%) between the transcriptomic and proteomic subtypes of PCa, which was consistent with previous report (*Cancer cell*, 2019, PMID: 30889379) (**Figure 1C**).” in the Result part of the revised manuscript. (page 5, line 102 to line 105).

Revised Figure 1C. Comparison of the proteomic subtyping (NMF-based) and two kinds of transcriptomic subtyping (NMF-based and PAM50-based) in the discovery cohort. NA represents not available.

Specific comments:

Q2. In Introduction, the authors mention ADT and the almost inevitable development of castration resistance. However, androgen receptor pathway inhibitors (ARPIs) are

now increasingly being used in addition to ADT in the first-line treatment setting, even for localized disease (especially high-risk localized PCa), yet the authors fail to mention ARPIs in the Introduction.

Response:

Thanks for your insightful comments. We agree with the reviewer that it is necessary to introduce androgen receptor pathway inhibitors (ARPIs). We have added the description of ARPIs as “To this end, multiple combination therapies including with androgen receptor pathway inhibitors (ARPIs), such as enzalutamide, have been explored in preclinical(*The New England journal of medicine*, 2014, PMID: 24881730). ARPIs can prolong the progression-free survival of PCa patients. But the PSA response rate of ARPIs is lower than 80%, with about 20% of patients show no response to ARPIs(*Journal of clinical oncology : official journal of the American Society of Clinical Oncology*, 2016, PMID: 26811535). Thus, there is an urgent need to explore the molecular characteristics of aggressive PCa and develop novel therapeutic targets” in the Introduction part. (page 3 to page 4, line 67 to line 71).

Q3. Figure 1 feels odd, and even slightly confusing, to see all of the stacked bar plots before the heatmap with the actual clusters. The figure panels should be reorganized so that readers see the heatmap before any other subpanels of Fig 1B. In fact, why are the bar plots included in Fig 1B when they are not referred to in the text until after Fig 1D?

Response:

Thanks for your professional comments and we are sorry to make you confused. Based on your suggestions, we have adjusted the layout of Figure 1 as follows. The bar plots were included in **Revised Figure 1F** and **Revised Figure S1E** as the order which they are referred to in the text.

Figure 1B. Heatmap visualizes the three proteomic subgroups of PCa based on the non-negative matrix factorization (NMF) unsupervised clustering algorithm and their corresponding clinical features in the discovery cohort. PSA represents prostate-specific antigen. BCR represents biochemical recurrence. NA represents not available. **Figure 1C.** Comparison of the proteomic subtyping (NMF-based) and two kinds of transcriptomic subtyping (NMF-based and PAM50-based) in the discovery cohort. NA represents not available.

Figure 1D. Gene sets enrichment analysis (GSEA) showing distinct molecular characteristics among three proteomic subgroups of the discovery cohort (one subgroup vs the union of other two subgroups).

Figure 1E. Kaplan-Meier curves showing the biochemical recurrence-free survival (bRFS) of patients in three proteomic subgroups of the discovery cohort. P-value is calculated by the Log-rank test.

Revised Figure 1F. Bar plots comparing the clinicopathological features among three proteomic subgroups of the discovery cohort, including prostate-specific antigen (PSA), Gleason score (GS) and lymph node metastasis (N stage). P-values are determined using one-way ANOVA test.

Figure 1G. GSEA showing the activation of the androgen response pathway in

subgroup 2 compared to the union of subgroup 1 and 3 of the discovery cohort.

Revised Figure S1E. Bar plots comparing the clinicopathological features among three proteomic subgroups of the discovery cohort, including age, tumor stage and risk group. P-values are determined using one-way ANOVA test.

Q4. Some of the figure legend colors are difficult to distinguish from each other when the heatmap columns are so narrow: PSA “>20” and “10-20” are difficult to distinguish, T stages T3a and T3b are difficult to distinguish, and it’s impossible (to my eye) to see the intermediate risk group in the “Risk group” annotation at the top of the heatmap.

Response:

Thanks for your professional comments. We regret that our original presentation of Figure 1B may confused the reviewers and readers. Based on your suggestions, we have adjusted the color and font size of **Revised Figure 1B** to make it easier to read.

Revised Figure 1B. Heatmap visualizes the three proteomic subgroups of PCa based on the non-negative matrix factorization (NMF) unsupervised clustering algorithm and their corresponding clinical features in the discovery cohort. PSA represents prostate-specific antigen. BCR represents biochemical recurrence. NA represents not available.

Q5. The heatmap includes “RNA Subgroup” along the top of the heatmap, but it has not yet been mentioned in the text and is also not mentioned in the caption. RNA subgroups are not defined anywhere in the manuscript, so why are they included in the heatmap in Fig 1B?

Thanks for your insightful comments. We regret to present the unmodified version of the heatmap which made you confused. Actually, we have attempted to perform the transcriptomic subtyping analysis in the original version of manuscript which was presented in the heatmap. However, we finally decided to focus on the result of proteomic subtyping. We apologized that we made a mistake on the heatmap updating. Based on your valuable comments above, we fully agree that it is necessary to perform the transcriptomic subtyping analysis and compare different transcriptomic subgroups and proteomic subgroups defined in our manuscript. Therefore, we have taken the RNA subgroups out of the heatmap and performed the comparison between transcriptomic subgroups and proteomic subgroups separately (**Revised Figure 1C**). We also included the detailed transcriptomic subtyping analysis and described as “By contrast, these patients were divided into three distinct subgroups at the transcriptomic level using the

NMF method and PAM50 signatures respectively, with mild overlap (15%-18%) between the transcriptomic and proteomic subtypes of PCa, which was consistent with previous report (*Cancer cell*, 2019, PMID: 30889379) (Figure 1C).” in the Result part of the revised manuscript. (page 5, line 102 to line 105).

Revised Figure 1C. Comparison of the proteomic subtyping (NMF-based) and two kinds of transcriptomic subtyping (NMF-based and PAM50-based) in the discovery cohort. NA represents not available.

Q6. The GSEA methods for Fig 1C,E needs more detail. GSEA can only compare one group versus another, but there are 3 groups to compare in Fig 1, so what do the GSEA results for each subgroup actually represent? Did one group serve as a common reference for two separate GSEA comparisons, or was each group compared to the union of the other two groups, or something else?

Response:

Thanks for your critical comments and we are sorry to make you confused. The GSEA in Figure 1C and Figure 1E were conducted between one group and the union of the other two groups, such as subgroup 1 versus subgroup 2 & 3, subgroup 2 versus subgroup 1 & 3, subgroup 3 versus subgroup 1 & 2. To clarify, the figure legend of **Revised Figure 1D** and **Revised Figure 1G** was adjusted as follows:

Revised Figure 1D. Gene sets enrichment analysis (GSEA) showing distinct molecular characteristics among three proteomic subgroups of the discovery cohort (one subgroup vs the union of other two subgroups).

G

Revised Figure 1G. GSEA showing the activation of the androgen response pathway in subgroup 2 compared to the union of subgroup 1 and 3 of the discovery cohort.

Q7. Suppl Fig 1 C,D: Do the figures show the number of proteins detected per sample (this is what the axes labels and figure caption imply), as opposed to the cumulative number of proteins detected across samples? If so, it appears that every tumor sample had more proteins detected than every normal sample, which is very surprising – why is this? Is there some sort of systematic tumor vs normal bias in the MS analysis? This would make tumor vs normal biological comparisons challenging as they would be confounded by any systematic bias. Or do panels C and D show cumulative numbers of proteins identified with samples deliberately ordered with all Normal first then all Tumor? If so, the axes labels and the figure caption should be changed to make this clear.

Response:

We appreciate the reviewer’s careful evaluation of our work. We regret that our original data presentation may confused the reviewers and readers. The **Figure S1 C-D** actually show the cumulative number of proteins and phosphorylation sites in tumor and normal samples. To clarify, we have adjusted the axes labels and the figure legends of **Figure S1 C-D** to make this clear. The vertical axes labels were adjusted as “Cumulative number of protein/phosphoprotein identification”. The figure legend of **Figure S1C-D** was adjusted as follows:

Figure S1C. Cumulative number of protein identifications in tumor and non-tumor samples, based on the Software: MaxQuant v1.5.30.

Figure S1D. Cumulative number of phosphorylation identifications in tumor and non-tumor samples, based on the Software: MaxQuant v1.5.30.

Q8. Fig 2B: What do the light and dark shades in the bar plot mean? Does the height of the bar represent the total number of patients in a subgroup and the light shaded area represent the patients with FOXA1 mutations? This is what I assume based the oncoprint and the main text, but it is not at all clear. A much more informative legend and more detailed caption are needed.

Response:

Thanks for your professional comments. We regret that our original presentation of **Figure 2B** may confused the reviewers and readers. Based on your suggestions, we have adjusted the format and legend of **Revised Figure 2B** to make it easier to read.

Revised Figure 2B. Bar plot comparing the FOXA1 mutation frequency among three proteomic subgroups of the discovery cohort. Numbers at the top of the bar represent the counts of each subgroup, with the corresponding proportions at the bottom of the bar. P-values are determined using one-way ANOVA test.

Q9. Fig 2C: What does dark grey in the oncoprint indicate? Missing data? There is no

legend for the dark grey color. What do the numbers at the right side of the oncoprint? Are they p-values? This needs to be explained in the figure panel and/or caption. If they are p values, what do they represent? Are they from ANOVA tests of associations of the genomic alterations with the proteomic subgroups? Caption needs much more information.

o Rows labelled as “del” (“RB1 del”, “CHD1 del”, etc.) show both deletions and amplifications, and there are separate legends for “Amp” and “Del”, yet “Amp” is also included in the “Del” legend. The legend for the mutations is incorrectly title as “CNV”. Is there a reason for including FOXA1 mutations in Fig 2C when it was already shown in Fig 2B? What about FOXA1 CNAs? FOXA1 amplification is common in prostate cancer, so this would surely be relevant to include in Fig 2C, especially if the authors are showing FOXA1 mutations and GATA2 amplifications.

Response:

Thanks for your constructive comments and we are sorry to make you confused. The dark grey color in the oncoplot indicates the missing data. The numbers at the right side of the oncoplot represents P-values of ANOVA tests of genomic alterations in different proteomic subgroups. There were some mistakes in row labels and captions for CNV. To clarify, we have adjusted the row labels and captions of this figure (**Revised Figure 2C**). Besides, we also included the analysis of FOXA1 amplification in this oncoplot based on the reviewer’s suggestion. The results showed that there was no significant difference of FOXA1 amplification among three proteomic subgroups (**Response Figure 4**). Therefore, we updated the revised version of Figure 2C without FOXA1 amplification in the revised manuscript (**Revised Figure 2C**).

Response Figure 4. Prevalence of specific gene fusions and SCNAs in three proteomic subgroups. NA represents missing data. Numbers on the bar plot represent the count of individuals, with figures in parentheses indicating the proportion of mutations within each group. P-values were determined using one-way ANOVA test.

Revised Figure 2C. Prevalence of specific SCNAs and gene fusions in three proteomic subgroups of the discovery cohort (top panel). NA represents not available. Bar plots comparing frequency of specific SCNAs and gene fusions among three proteomic subgroups of the discovery cohort (bottom panel). Numbers at the top of the bar represent the counts of each subgroup, with the corresponding proportions at the bottom of the bar. P-values are determined using one-way ANOVA test.

Q10. Fig 2D: it is unclear what it is the meaning of Enrichment analysis is for Subgroup II versus what? Versus Subgroups I and III combined? Or versus Subgroup I. Caption needs more detail and/or the figure panel needs more annotation.

Response:

Thanks for your critical comments and we are sorry to make you confused. The KSEA in **Figure 2D** was conducted between subgroup 2 and the union of subgroups 1 and subgroup 3. To clarify, the figure legend of **Figure 2D** was adjusted as “Kinase-substrate enrichment analysis (KSEA) showing significantly up-regulated and down-regulated kinases in subgroup 2 patients of the discovery cohort (subgroup 2 vs subgroup 1 and subgroup 3 combined). Red bars refer to $FDR < 0.05$ and $z \text{ score} > 0$, blue bars refer to $FDR < 0.05$ and $z \text{ score} \leq 0$.” (page 47, line 1127 to line 1130)

Figure 2D. Kinase-substrate enrichment analysis (KSEA) showing significantly up-regulated and down-regulated kinases in subgroup 2 patients of the discovery cohort (subgroup 2 vs subgroup 1 and subgroup 3 combined). Red bars refer to $FDR < 0.05$ and $z \text{ score} > 0$, blue bars refer to $FDR < 0.05$ and $z \text{ score} \leq 0$.

Q11. Supplementary Fig 2B: What statistical tests were performed? Are p values from ANOVAs? Or are they from a comparison of two particular subgroups, and if so what

test was used? Or something else?

Response:

Thanks for your critical comments and we are sorry to make you confused. FDR values from Kruskal-Wallis test and B-H correction among three subgroups were presents in **Figure S2B**. To clarify, we have adjusted the figure legend of **Figure S2B (Revised Figure S2D)** as “Comparison of immune cell infiltrations among three proteomic subgroups by MCP-counter analysis. P-values were determined using Kruskal-Wallis test and B-H correction (N.S. $p>0.05$; * $p<0.05$ and *** $p<0.001$)”. (page 9, line 89 to line 92 in Supplement)

Revised Figure S2D. MCP-counter analysis showing immune cell infiltrations among three proteomic subgroups in the discovery cohort. P-values are determined using Kruskal-Wallis test and B-H correction (N.S. $p>0.05$; * $p<0.05$ and *** $p<0.001$).

Q12. Supplementary Figure 2C: The GSEA compares Subgroup I to what? To Subgroups II and III combined? Or just Subgroup I? Or...?

Response:

Thanks for your professional comments and we are sorry to make you confused. The GSEA in **Figure S2C** compared subgroup 1 to subgroup 2 and subgroup 3 combined. To clarify, we have adjusted the figure legend of **Figure S2C (Revised Figure S2E)** as “GSEA showing the top 10 up-regulated metabolic pathways in subgroup 1 patients of the discovery cohort (subgroup 1 vs subgroup 2 and subgroup 3 combined)”.

E

Figure S2E

Revised Figure S2E. GSEA showing the top 10 up-regulated metabolic pathways in subgroup 1 patients of the discovery cohort (subgroup 1 vs subgroup 2 and subgroup 3 combined)

Q13. Line 159: What is the validation cohort? No reference or details are provided in the main text. Line 159: “To verify our findings, we employed an external validation cohort with 119 PCa patients.”

Response:

Thanks for your professional comments and we are sorry to make you confused about the description of the validation cohorts. In our manuscript, a total of 219 treatment-naïve Chinese patients with localized PCa were enrolled across two validation cohorts. For the validation cohort 1, 119 patients underwent radical prostatectomy from May 2017 to October 2021 at the Third and Seventh Affiliated Hospital of Sun Yat-sen University and Guangzhou First People’s Hospital were enrolled. Tumor tissue paraffin sections of the validation cohort 1 were used for validation of proteomic subtyping and immunohistochemical (IHC) staining of NANS.

For the validation cohort 2, 100 patients who underwent radical prostatectomy surgeries from January 2015 to October 2018 at the Daping Hospital, Army Medical University were enrolled. Tumor tissue paraffin sections of the validation cohort 2 were used for IHC staining of NANS. Detailed descriptions of two validation cohorts were presented in the Methods section.

For ease of reading and understanding, we also adjusted the description of two validation cohorts in the Result part as “To verify our findings, we firstly constructed a 39-protein subtyping panel, which was filtered by the differential expression analysis and the Random Forest algorithm in the discovery cohort. Then, we additionally employed 119 PCa patients from other centers (details provided in the Methods section) as the validation cohort 1. Based on the proteomic profiling data and the 39-protein subtyping panel, the validation cohort 1 could be successfully categorized into three proteomic subgroups consistent with those identified in the discovery cohort.” (page 8, line 174 to line 180) and “To further verify the function of NANS as prognostic indicator, additional 100 PCa patients from another center were recruited as the validation cohort 2 (details provided in the Methods section) and conducted the IHC

staining of NANS. Survival analysis in two validation cohorts both confirmed that higher NANS expression was associated with poor prognosis of PCa.” (page 9, line 193 to line 197)

Q14. Lines 159–161: The authors developed a 39-protein subtyping panel and classifier? Considering that the proteomic subgroups with different survival probabilities were validated in a different cohort, this is very interesting and deserves more details in the text!

Line 159-161: “To verify our findings, we employed an external validation cohort with 119 PCa patients. This cohort could be successfully subtyped into three proteomic subgroups by a 39-protein subtyping panel, which filtered by the Random Forest algorithm and validated in our cohort (Figure S2F).”

Response:

Thanks for your professional comments. Based on your suggestions, we have adjusted the detailed description of 39-protein subtyping panel in the manuscript as “To verify our findings, we firstly constructed a 39-protein subtyping panel, which was filtered by the differential expression analysis and the Random Forest algorithm in the discovery cohort (Figure S3A-C). Then, we additionally employed 119 PCa patients from other centers (details provided in the Methods section) as the validation cohort 1. Based on the proteomic profiling data and the 39-protein subtyping panel, the validation cohort 1 could be successfully categorized into three proteomic subgroups consistent with those identified in the discovery cohort (Figure 3D, Figure S3A)”. (page 8, line 174 to line 180) Besides, we added a flowchart to illustrate the pipeline for the construction and validation of the 39-protein subtyping panel in detail (**Revised Figure S3A**).

Revised Figure S3A. Design of the analytical pipeline for construction and validation of the 39-protein subtyping panel.

Revised Figure S3B. The 39-protein panel for proteomic subtyping of PCa filtered by the Random Forest algorithm, SEG represents subgroup enriched genes.

Revised Figure S3C. The Receiver Operating Characteristic Curves (ROC) of the 39-protein subtyping model in the validation dataset.

Revised Figure 3D. Kaplan-Meier curves showing the bRFS of patients in three proteomic subgroups of the validation cohort 1. P-value is calculated by the Log-rank test.

Q15. Supplementary Fig 2F: Does this show the 39 signature proteins in the discovery cohort or the validation cohort? This isn't clear.

Response:

Thanks for your professional comments and we are sorry to make you confused about the Figure legend of **Figure S2F**. The 39 signature proteins were selected from the discovery cohort. The heatmap of **Figure S2F** showed the protein expression levels of the 39 signature proteins in the discovery cohort. To clarify, we have adjusted the Figure legend of **Figure S2F (Revised Figure S3B)** as “The 39-protein panel for proteomic subtyping of PCa filtered by the Random Forest algorithm, SEG represents subgroup enriched genes.”. (page 10, line 106 to line 107 in Supplement)

Revised Figure S3B. The 39-protein panel for proteomic subtyping of PCa filtered by the Random Forest algorithm, SEG represents subgroup enriched genes.

Q16. Fig 4B: The grouping of cell lines into proteomic subgroups is very surprising. DU145 and PC3 are generally considered to be aggressive-phenotype AR- NE- cells lines that cluster together in most analyses (e.g. based on transcriptomic or epigenomic data), while VCaP is essentially an AR+ NE+ cell line and C4-2 is a very AR-driven cell line. It is therefore very unexpected to see PC3 and C4-2 group together on one hand and DU145 and VCaP to group together on another hand. Can the authors comment on why proteome-based classification groups together cell lines that we know to be phenotypically very different, and separates cell lines that are known to be phenotypically similar? Does this represent a major disconnect between the transcriptome and proteome, and what are the clinical implications of this? Or is it possible that cell lines do not adequately reflect the proteome of primary prostate cancer since they were generated from mets?

Response:

Thanks for your valuable comments. In our proteomic subtyping results of cell lines, PC3 and C4-2 cells were grouped together, while DU145 and Vcap cells were also grouped together. We consider this due to three aspects of interferences: Firstly, cell lines sometimes do not adequately reflect the proteome of primary prostate cancer as they were generated from metastatic tumors, which may affect their biological phenotypes. Secondly, there is significant discordance between the transcriptome and the proteome of PCa. In our discovery cohort, the correlation coefficient between transcriptome and proteome data of prostate cancer was only 0.123 (**Response Figure 5A**). As to other studies, the correlation coefficient between transcriptome and proteome data was around 0.2-0.8 in different malignancies (*Cancer cell*, 2019, PMID: 30889379; *Cell*, 2019, PMID: 31585088; *J Hematol Oncol*, 2022, PMID: 35659036; *J Hematol Oncol*, 2022, PMID: 36434634). Therefore, these cell lines possibly exhibit different phenotypes of AR dependence at protein level. Finally, despite subgroup 2 patients show the transcriptomic activation of the androgen signaling with higher AR score, there are still part of patients in subgroup 1 and 3 present high AR score and high AR protein expression (**Response Figure 5B**), which partially corresponded to the subtyping results of cell lines. This outcome indicated that tumors with distinct AR dependence could be clustered together by our 39-protein subtyping panel, which was not strictly AR-related. To comprehensively understand and address this question, we also performed the literature search. We find that there are actually some disconnections between the biological phenotypes and molecular subtypes based on omics data. For instance, PC3 and DU145 with similar biological phenotypes are grouped into different subgroups based on the transcriptomic data of *Cancer Research* (**Response Figure 5C**) (*Cancer research*, 2016, PMID: 27302169). Indeed, this scientific question is intriguing and warrants further exploration in the future.

Response Figure 5A. Correlation between mRNA and protein expression of all genes. Blue: positively correlated genes; yellow: negatively correlated genes.

Response Figure 5B. Comparison of AR score and AR protein expression in three proteomic subgroups and adjacent normal samples.

Response Figure 5C. “Clustering of different prostate cancer cell lines.”(*Cancer research*, 2016, PMID: 27302169)

Q17.

- o Is the proteomic classification of cell lines based on the 39-protein signature panel?*
- o Is the proteomic classification of cells based on proteomic data, or on application of the 39-protein signature to RNA (gene expression) data?*
- o Where did the data for these cell lines come from? Were they new data generated by this study, or were publicly available data used?*

Response:

Thanks for your critical comments. The proteomic classification of cell lines was performed by application of the 39-protein subtyping panel to the newly generated proteomic data of these cell lines. We used 8 PCa cell lines and 3 independent biological replicates of every cell line for proteomic profiling. The generated proteomic raw data of all cell lines were uploaded to the database according to the submission requirements with the accession ID IPX0009562000.

Q18. Page 9 / Fig 4: Why did the authors choose to use a murine cell line when they showed that two human cell lines were also in the same subgroup. Given the availability of two human cell lines in the subgroup of interest, it seems like an odd choice to use a murine cell line for in vitro analyses. It even gives the impression that the authors tried human cell lines and the experiments didn't support the hypothesis, so they resorted to a murine cell line.

Response:

Thank you for the valuable question. Orthotopic mouse model has been usually used to discuss the tumor immune microenvironment, as it still remains the host immune response (*Cancer research*, 2015, PMID: 26294211). Because of the immunological rejection, murine cell line is used to construct the orthotopic tumor mouse model. In our study, we aim to illustrate the effect of NANS deletion to the tumor immune microenvironment using the orthotopic mouse model. Therefore, we first chose the murine cell line Myc-Cap for *in vitro* experiments as same as *in vivo* experiments.

Additionally, to further validate the reliability of our study, we supplemented *in vitro* experiments with another human PCa cell line (22Rv1) which belongs to subgroup 2. We observed that *Nans* depletion in 22Rv1 cells largely reduced sialic acid synthesis measured by targeted metabolomics (**Revised Figure 4C-E**), accompanied with decreased sialylation level of tumor cells (**Revised Figure 4F**). Besides, we found that knockout of *NANS* barely influenced the proliferation of 22Rv1 cells *in vitro* measured with CCK8 and colony-forming assays (**Revised Figure S4A-B**), indicating the essential involvement of other components in tumor microenvironment.

Therefore, the results from human-derived 22Rv1 and murine Myc-CaP PCa cells both confirmed the regulatory function of NANS in the sialic acid synthesis and tumor sialylation of PCa.

Revised Figure 4C. Knockout (KO) of *NANS* in 22Rv1 and Myc-CaP cells via single guide RNA (sgRNA) as determined by qRT-PCR. Empty vector is used as control. P-values are determined using two-tailed Student's t test.

Revised Figure 4D. KO of *NANS* in 22Rv1 and Myc-CaP cells via sgRNA as determined by Western blot. Empty vector is used as control. P-values are determined using two-tailed Student's t test.

Revised Figure 4E. Comparison of the relative abundance of N-Acetylneuraminic acid in 22Rv1 and Myc-CaP cells between sg*NANS* and control group based on the targeted metabolomics analysis. The middle lines represent the median, and the lower and upper hinges denote the 25-75% IQR, with whiskers extending up to a maximum of 1.5 times IQR. $n=5$ for each group. P-values are determined using the two-sided Wilcoxon rank-sum test.

Revised Figure 4F. The heatmap showing differentially expressed sialylated sites in 22Rv1 and Myc-CaP cells between sg*NANS* and control group.

Revised Figure S4A. CCK8 assays for 22Rv1 and Myc-CaP cells of sgNANS and control group. P-value is determined using two-way ANOVA test.

Revised Figure S4B. Colony-forming assays for 22Rv1 and Myc-CaP cells of sgNANS and control group. Data are presented as mean \pm Standard deviation (SD). P-values are determined using two-tailed Student's t test.

Comments on Methods

Q19. The methods mention two validation cohorts, but this was not at all obvious in the main text. It is only explained once you dive in the Methods.

Response:

Thanks for your professional comments. We fully agree with the reviewer that detailed description of two validation cohorts should be added in the main text. Based on your suggestions, we have added the description of two validation cohorts in the manuscript as “To verify our findings, we firstly constructed a 39-protein subtyping panel, which was filtered by the differential expression analysis and the Random Forest algorithm in the discovery cohort. Then, we additionally employed 119 PCa patients from other centers (details provided in the Methods section) as the validation cohort 1. Based on the proteomic profiling data and the 39-protein subtyping panel, the validation cohort 1 could be successfully categorized into three proteomic subgroups consistent with those identified in the discovery cohort.” (page 8, line 174 to line 180) and “To further verify the function of NANS as prognostic indicator, additional 100 PCa patients from another center were recruited as the validation cohort 2 (details provided in the Methods section) and conducted the IHC staining of NANS. Survival analysis in two validation cohorts both confirmed that higher NANS expression was associated with poor prognosis of PCa.” (page 9, line 193 to line 197)

Q20. Please provide the rationale of using hg19 used as the reference genome for mapping sequencing reads from human specimens? hg38 has been around for a long time and should be preferred.

Response:

Thanks for your professional comments. To clarify this question, we checked the original codes for the reference genome which we used for mapping sequencing reads from human specimens. Actually we used hg38 as the reference genome in the original codes as we presents in the Response Figure 3. We are sorry to make this written mistake and we have adjusted it in the Method part as “Clean data was mapped to the GRCh38/hg38 human reference genome by Burrows Wheeler Aligner (BWA) software and Samblaster to generate BAM file.”.

```
10  UNCLUSTED
31  Homo sapiens GRCh38.dna.alt.fa.gz
68  Homo sapiens GRCh38.dna.chromosome.1.f
```

Response Figure 3. The original codes for the reference genome in WES data analysis.

Q21. The methods describe V(D)J / TCR sequencing, but this is not mentioned anywhere else in the manuscript...

Response:

Thank you for your precise evaluation of our work. We regret that our original presentation may confused the reviewers and readers. We have actually conducted the V(D)J/TCR sequencing when performing the single-cell sequencing. But in the present study, the V(D)J/TCR sequencing data was indeed not involved. It was a written mistake to involve the method of V(D)J/TCR sequencing in the original manuscript. To obviate misunderstanding, we have removed the content about V(D)J/TCR sequencing in Methods part of the revised manuscript.

Q22. Does the definition of TMB include subclonal mutations?

Response:

Thank you for your professional comments. The analysis of TMB in our manuscript includes subclonal mutations. To clarify this question, we have added the detailed definition of TMB in the Method part as “Tumor mutation burden (TMB) was defined as the number of somatic mutations (including base substitutions and indels) in the coding region. Besides, both clonal and subclonal mutations were included in the analysis. To reduce sampling noise, synonymous alterations were also counted. In order to calculate the TMB, the total number of mutations counted was divided by the size of the coding sequence region of the Agilent SureSelect Human All Exon V6.”

Q23. Despite it's widespread use, FPKM is not a suitable normalization unit for RNA-seq data. TMM, Upper Quartile, or TPM normalization would be better.

Response:

Thank you for your professional suggestions. Following your recommendation, we reviewed and summarized the published researches related to mRNA data normalization. Given our study cohort includes 145 patients, samples for next-

generation sequencing were obtained from paired tumor and adjacent non-tumor tissues, sequenced at the same depth on the same platform (Illumina NovaSeq 6000), indicating that the batch effect between samples is not significant. Therefore, we did not choose the Upper Quartile method, commonly used for microarray sequencing, or the TMM algorithm, often used for comparisons between different tissue sample groups, for RNA expression normalization (*BMC bioinformatics*, 2015, PMID: 26511205). And we agree that TPM is indeed more suitable than FPKM for differentiating samples within the same group, and it is algorithmically superior to FPKM to some extent. So we reconducted transcriptomic-related analyses based on TPM normalization, and the outcomes indicated that TPM-based analyses (EcoTyper, MCP-Counter and ssGSEA) showed consistent trends compared to the original FPKM-based analyses. These results indirectly supported the reliability of our transcriptomic data. Considering the reviewer's advice and the established superiority of TPM over FPKM (*Theory in biosciences = Theorie in den Biowissenschaften*, 2012, PMID: 22872506), we have replaced the FPKM-based analyses with TPM-based contents in the manuscript as follows:

Revised Figure 1C. Comparison of the proteomic subtyping (NMF-based) and two kinds of transcriptomic subtyping (NMF-based and PAM50-based) in the discovery cohort. NA represents not available.

Revised Figure 4G. EcoTyper analysis dissecting the tumor immune

microenvironment of patients in three proteomic subgroups of the discovery cohort.

Revised Figure S2C. Single-sample gene set enrichment analysis (ssGSEA) of immune-related pathways among three proteomic subgroups in the discovery cohort. FDR values are determined using Kruskal-Wallis test and Benjamini-Hochberg (B-H) correction.

Revised Figure S2D. MCP-counter analysis showing immune cell infiltrations among three proteomic subgroups in the discovery cohort. P-values are determined using Kruskal-Wallis test and B-H correction (N.S. $p > 0.05$; * $p < 0.05$ and *** $p < 0.001$).

Q24. The best way to include a logFC threshold into differential expression analysis is to build it into the statistical testing procedure (as enabled by modern differential expression analysis methods for RNA-seq), rather than to test for differential expression different from 0 and subsequently apply a logFC threshold post hoc.

Response:

Thanks for your critical comments. We regret that our original presentation may confused the reviewers and readers. Actually, we used the DESeq2 to perform the differential gene expression analysis. Genes with $|\log_2 \text{fold-change}| > 1$ and P-adjusted < 0.05 were filtrated as differentially expressed genes. In the whole process, we did not test for differential expression different from 0. To clarify, we adjusted the Method part for differential expression analysis as “Sequenced reads were trimmed with fastp V.0.23.1. In this step, clean data (clean reads) were obtained by trimming reads containing adapter or with low quality from raw data. Then clean reads were mapped to the reference genome (GRCh38) using HISAT2 V.2.1.0 and with default parameters. RSEM (v1.2.29) was used to quantify gene abundance as read counts. The R package “DESeq2” was applied to screen differentially expressed messenger RNAs (mRNAs)

between different groups. Next, the P-value was calculated by the false discovery rate (FDR)-corrected method. The mRNAs with $|\log_2 \text{ fold-change} | > 1$ and P-adjusted < 0.05 were filtrated as differentially expressed genes”.

Q25. Line 642: why was limma needed for ssGSEA?

Response:

Thank you for your precise evaluation of our work. We regret that we made a mistake in the Method part of ssGSEA in our original manuscript. To clarify, we have adjusted it in the Method part as “ssGSEA, a method for quantifying gene set enrichment in individual samples, was employed using the Gene Set Variation Analysis (GSVA) package in R. This approach allows for the assessment of variations in pathways and biological processes across a sample population, thus providing insights into the heterogeneity of immune microenvironment in patients of diverse proteomic subgroups. The ssGSEA analysis was based on 20 immune gene sets, including genes related to different immune cell types, functions, pathways, and checkpoints”.

Q26. Please provide references for the gene sets used for ssGSEA.

Response:

Thank you for your precise evaluation of our work. The detailed information of 20 immune-related gene sets used for ssGSEA in Supplementary Figure 2C were provided in the Source Data (including the specific genes and the corresponding references).

Q27. What signature matrix was used for CIBERSORTx?

Response:

Thank you for your professional comments. Based on the reference, FPKM, TPM and RPKM matrix are all allowed in the CIBERSORTx software(*Cell*, 2021, PMID: 34597583). In the revised manuscript, we used the TPM matrix of all tumor samples in the discovery cohort.

Q28. Line 669: When the authors say “data were selected after pretreatment”, do they mean “... after preprocessing”?

Response:

Thank you for your precise evaluation of our work. We regret that our original description may confused the reviewers and readers. Based on your suggestions, we have adjusted the description in the Method part as “ Prior to the clustering analysis, transcriptomic and proteomic data preprocessing was performed”.

Q29. The authors mention multiple statistical tests that were used throughout the manuscript, but the main text and figure captions never specify the test that was used in a particular situation. The authors should be clearer about which statistical test was used for each analysis.

Response:

Thank you for your precise evaluation of our work. We regret that our original

manuscript may confused the reviewers and readers about the statistical test we used in each analysis. Based on your suggestions, we have added the specific statistical test method in the figure legend of each analysis.

Reviewer #4

Major comments:

Q1. Although one of the main outcomes of this work is the discovery of three subtypes of prostate cancer, the method for stratifying patients, as well as the creation and validation of a panel of 39 proteins, lack sufficient detail (see comments on lines 668-707 below). Also, I did not locate any links to code and raw data used in this article.

Response:

Thanks for your professional comments. Based on your suggestions, we have adjusted the detailed description of 39-protein subtyping panel in the manuscript as “To verify our findings, we firstly constructed a 39-protein subtyping panel, which was filtered by the differential expression analysis and the Random Forest algorithm in the discovery cohort (Figure S3A-C). Then, we additionally employed 119 PCa patients from other centers (details provided in the Methods section) as the validation cohort 1. Based on the proteomic profiling data and the 39-protein subtyping panel, the validation cohort 1 could be successfully categorized into three proteomic subgroups consistent with those identified in the discovery cohort (Figure 3D, Figure S3A)”.(page 8, line 174 to line 180) Besides, we added a flowchart to detailly illustrate the pipeline for the construction and validation of the 39-protein subtyping panel (**Figure S3A**).

The detailed method for the construction and validation of the 39-protein subtyping panel was presented in the Method part: Prior to select protein biomarkers, differential analysis (Wilcoxon test) was performed between one subtype and other two subtypes (considered as one group). $FC \geq 1.4$ & $P\text{-adjusted} < 0.05$ was considered as significant difference. Next, the top 50 differentially expressed proteins were subjected to the following analysis. Random forest algorithm was used to find potential biomarkers. Proteomics data was divided into training and test datasets with 7:3 ratio and cumulative AUC was applied to select minimum number of biomarkers with training dataset (10-fold cross validation). Finally, the model performance was measured using the test dataset. To validate our proteomic subtyping results in the validation cohort 1, the nearest template prediction (NTP), an unsupervised clustering algorithm, was adopted using the 39-protein panel filtered by machine learning. To further analyze the biological characteristics of these subgroups, we performed Gene Set Enrichment Analysis (GSEA) to identify the pathway alterations that underlying each subgroup. (page 28, line 620 to line 626)

The raw data for proteomics analysis have been uploaded to the database according to the submission requirements, and the accession code was added to the Data Availability part as “The raw proteomics and phosphoproteomics data generated in this study is publicly available in the iProX database under the accession ID IPX0009562000”. (page 37, line 827 to line 828) The code for the construction and

validation of 39-protein subtyping panel (including differential expression analysis and the Random Forest analysis) are publicly available in Github at <https://github.com/Diluczhang/Proteome-of-PCA.git>.

Revised Figure S3A. Design of the analytical pipeline for construction and validation of the 39-protein subtyping panel.

Revised Figure S3B. The 39-protein panel for proteomic subtyping of PCa filtered by the Random Forest algorithm, SEG represents subgroup enriched genes.

Revised Figure S3C. The Receiver Operating Characteristic Curves (ROC) of the 39-protein subtyping model in the validation dataset.

Revised Figure 3D. Kaplan-Meier curves showing the bRFS of patients in three proteomic subgroups of the validation cohort 1. P-value is calculated by the Log-rank test.

Q2. What is the relationship between mRNA-based subtypes (Fig. 1B, RNA Subgroup) and proteome-based subgroups 1-3 discovered and investigated in this work? Is the expression of NANS gene/transcript(s) associated with RFS?

Response:

Thanks for your insightful comments. We regret to present the unmodified version of the heatmap with “RNA subgroup” which made you confused. Actually, we have attempted to perform the transcriptomic subtyping analysis in the original version of manuscript which was presented in the heatmap. But we finally decided to mainly focus on the result of proteomic subtyping. We are very sorry that we made a mistake on the heatmap updating. Based on reviewer’s valuable comments above, we fully agree that it is necessary to perform the transcriptomic subtyping analysis and compare different transcriptomic subgroups and proteomic subgroups defined in our manuscript. Therefore, we have taken the RNA subgroups out of the heatmap and performed the comparison between transcriptomic subgroups and proteomic subgroups separately (**Revised Figure 1C**). We also add the detailed transcriptomic subtyping analysis and described as “By contrast, these patients were divided into three distinct subgroups at the transcriptomic level using the NMF method and PAM50 signatures respectively, with mild overlap (15%-18%) between the transcriptomic and proteomic subtypes of PCa, which was consistent with previous report (*Cancer cell*, 2019, PMID: 30889379) (**Figure 1C**).” in the Result part of the revised manuscript. (page 5, line 102 to line 105).

Besides, we performed the supplementary analysis of association between NANS mRNA expression and bRFS. The results showed that the mRNA expression of NANS was not related to the biochemical recurrence of PCa in our discovery cohort (P=0.075, **Response Figure 6**).

Revised Figure 1C. Comparison of the proteomic subtyping (NMF-based) and two kinds of transcriptomic subtyping (NMF-based and PAM50-based) in the discovery cohort. NA represents not available.

Response Figure 6. Kaplan-Meier curves comparing the bRFS between NANS-mRNA higher/lower expression groups in the discovery cohort. P-value was calculated by Log-rank test.

Q3. The authors show that three subgroups have distinct distributions of mutations, T or N stages, GS, etc., but do not evaluate associations of these tumor and patient features with survival. Is it possible that the associations of subgroups with RFS are explained by the associations between clinical features and RFS?

Response:

We appreciate the reviewer’s professional advice and totally agree that clinical parameters should be taken into account when assessing the prognostic role of proteomic subgroups. Based on your suggestion, we added the analysis of univariate and multivariate cox proportional hazard models for independent prognostic predictor evaluation. The results confirmed that the proteomic subgroup 2 was an independent prognostic predictor of PCa (HR=2.53, P=0.021). And this result was added in **Revised Table S2** and described as “Even taking these clinical factors into account, the proteomic subgroup 2 remained an independent prognostic predictor of PCa (Revised Table S2).” in the revised manuscript. (page 5, line 111 to line 113)

Revised Table S2

Variables	HR	95%CI	P-values
Age (years)	0.96	(0.92,1.01)	0.132
PSA (10ng/ml)	1.03	(1.00,1.05)	0.027
Gleason score			
<=7	1.00		
>=8	2.35	(1.11,5.01)	0.026
T-stage			
pT2	1.00		
pT3	1.63	(0.81,3.30)	0.172
N-stage			
pN0	1.00		
pN1	1.25	(0.52,3.03)	0.621
Proteomic subgroups			
Subgroup 3	1.00		
Subgroup 2	2.53	(1.15,5.58)	0.021
Subgroup 1	1.91	(0.75,4.88)	0.174

Revised Table S2. The multivariable cox regression for the biochemical recurrence-free survival (including proteomic subgroups) in the discovery cohort.

Q4. The section Statistics (page 36, lines 803-813) lists all statistical tests used in this work, but it remains difficult for the reader to understand which statistical test was used in each specific analysis. Also, the authors frequently report p-values without mentioning any multiple testing adjustment procedure. I would recommend specifying the statistical test(s) used along with the result description and/or in figure captions.

Response:

Thank you for your precise evaluation of our work. We regret that our original manuscript may confused the reviewers and readers about the statistical test we used in each analysis. Based on your suggestions, we have added the specific statistical test method in the figure legend of each analysis.

Minor comments:

Q5. Fig. 1 B: I would recommend using the same color scales for bar plots comparing the clinicopathological features and annotation of the heatmap and separating panels with the barplots and the heatmap. Why does the upper panel show the barplots for only four out of eight features (RNA subgroup, PSA, T Stage and BCR are not shown)?

Response:

Thank you for your precise evaluation of our work. Based on your suggestions, we have adjusted the color scales of **Figure 1B** as follows. In the revised version of manuscript, we have also adjusted the presentations of clinical features. Three clinical features (PSA, Gleason score and N stage) with $P < 0.05$ were presented in bar plots of **Revised Figure 1F**. Other clinical features with $P > 0.05$ were presented in bar plots of **Revised Figure**

S1E. The RNA subgroup was removed from the heatmap and presented in the **Revised Figure 1C** for detailed comparison with the proteomic subtypes and other transcriptomic subtypes. BCR was not presented as bar plot because there was already the K-M curves comparing the bRFS among three subgroups.

Figure 1B. Heatmap visualizes the three proteomic subgroups of PCa based on the non-negative matrix factorization (NMF) unsupervised clustering algorithm and their corresponding clinical features in the discovery cohort. PSA represents prostate-specific antigen. BCR represents biochemical recurrence. NA represents not available.

Figure 1C. Comparison of the proteomic subtyping (NMF-based) and two kinds of transcriptomic subtyping (NMF-based and PAM50-based) in the discovery cohort. NA represents not available.

Figure 1D. Gene sets enrichment analysis (GSEA) showing distinct molecular characteristics among three proteomic subgroups of the discovery cohort (one subgroup vs the union of other two subgroups).

Figure 1E. Kaplan-Meier curves showing the biochemical recurrence-free survival (bRFS) of patients in three proteomic subgroups of the discovery cohort. P-value is calculated by the Log-rank test.

Revised Figure 1F. Bar plots comparing the clinicopathological features among three

proteomic subgroups of the discovery cohort, including prostate-specific antigen (PSA), Gleason score (GS) and lymph node metastasis (N stage). P-values are determined using one-way ANOVA test.

Figure 1G. GSEA showing the activation of the androgen response pathway in subgroup 2 compared to the union of subgroup 1 and 3 of the discovery cohort.

Revised Figure S1E. Bar plots comparing the clinicopathological features among three proteomic subgroups of the discovery cohort, including age, tumor stage and risk group. P-values are determined using one-way ANOVA test.

Q6.

Fig. 1E. This figure shows GSEA results for HALLMARK_ANDROGEN_RESPONSE gene set. However, this gene set is not displayed in Fig. 1C (the closest match I see in Fig. 1C is the "Androgen_response" gene set). Please explain this or highlight HALLMARK_ANDROGEN_RESPONSE gene set in Fig. 1C.

Response:

Thank you for your precise evaluation of our work. We are sorry that our original presentation of the “Androgen_response” gene set was not clear and may confuse the readers. Actually, the “Androgen_response” gene set was included in the hallmark gene sets of **Figure 1C**. Based on your suggestions, we have highlighted the “Androgen_response” gene set in **Figure 1C (Revised Figure 1D)** as follows.

Figure 1D

Revised Figure 1D. Gene sets enrichment analysis (GSEA) showing distinct molecular characteristics among three proteomic subgroups of the discovery cohort (one subgroup vs the union of other two subgroups).

Q7. page 5, lines 109-111: "Simultaneously, although metabolism subgroup 1 also displayed features of metabolic enrichment, however, this group patients exhibited a favorable prognosis and lower association with AR activation (Figure S1E)." - Please rephrase, because according to Figure S1E, no significant enrichment of the HALLMARK_ANDROGEN_RESPONSE gene set in the comparison Subgroup 1 vs Subgroups 2/3 is observed. For example, "lower association" -> "no statistically significant association"

Response:

Thank you for your critical comments. We fully agree with the reviewer that our original description of **Figure S1E** was inaccurate. Based on your suggestion, we have adjusted the description of **Figure S1E** in the manuscript as "Simultaneously, metabolism subgroup 1 also displayed features of metabolic enrichment, however, patients in this group exhibited a favorable prognosis and no statistically significant association with AR activation (Figure S1F)".(page 6, line 116 to line 119)

Q8.

Fig. 2B: the relationship between the upper and lower panels is not explained.

Fig.2 B and C: the transparent color in bar plots is not explained.

Response:

Thank you for your precise evaluation of our work. We regret that our original presentation of **Figure 2B** and **2C** may confuse the reviewers and readers. The bottom panel of **Figure 2B** was the oncoplot presenting the prevalence of the top 30 recurrent mutation genes in three proteomic subgroups. Among the top 30 recurrent mutation genes, FOXA1 was the only gene whose mutation frequency was differential among three proteomic subgroups. The top panel of **Figure 2B** compared the prevalence of

FOXA1 mutation among three proteomic subgroups and provided detailed statistical data. Besides, the color of **Figure 2B** and **2C** may confuse the readers. To clarify, we adjusted the colors and formats of **Revised Figure 2B** and **2C** to make them easier to understand.

Revised Figure 2B. Bar plot comparing the *FOXA1* mutation frequency among three proteomic subgroups of the discovery cohort. Numbers at the top of the bar represent the counts of each subgroup, with the corresponding proportions at the bottom of the bar. P-values are determined using one-way ANOVA test.

Revised Figure 2C. Bar plots comparing frequency of specific SCNAs and gene fusions among three proteomic subgroups of the discovery cohort. Numbers at the top of the bar represent the counts of each subgroup, with the corresponding proportions at the bottom of the bar. P-values are determined using one-way ANOVA test.

Q9. Fig. 2 D and page 6, lines 126-128: Why does the unadjusted p-value cutoff of 0.05 was applied here? Same as in GSEA, correction for multiple testing is necessary here. What are the results of KSEA for two other subgroups?

Response:

Thank for your critical comments. In the original manuscript, we only presented the results of kinase-substrate enrichment analysis (KSEA) in subgroup 2 patients. Based on your suggestion, we provide supplementary analysis of the phosphoproteome (KSEA) in subgroup 1 and 3 patients (**Revised Figure S2A-B**). The detailed description of the KSEA analysis are added and described as “Meanwhile, the kinase PRKD3 associated with PCa progression was significantly enriched in subgroup 1 (Figure S2A), in consistent with the finding that expression of PRKD3 contributes to mast cell infiltration and angiogenesis in the tumor microenvironment of PCa (Journal of molecular medicine, 2023, PMID: 36843036). By contrast, the atypical inflammatory kinase IKBKE was significantly enriched in the immune subgroup 3 (Nature immunology, 2003, PMID: 12692549; Science, 2003, PMID: 12702806) (Figure S2B), which has been reported to promote PCa tumor growth through modulating the Hippo pathway (Nucleic acids research, 2020, PMID: 32324216). Therefore, the molecular features in the phosphoproteomic level are potentially associated with its proteomic subgroups.” in the Result part (page 7, line 137 to line 144) and as “Protein kinases and phosphatases play crucial roles in the reversible phosphorylation-dephosphorylation cycle, regulating tumor development. The progression of PCa may be attributed to abnormal activation or overexpression of kinases. For instance, excessive activation of pathways like the mitogen-activated protein kinase (MAPK) and the nuclear factor- κ B (NF- κ B) pathway promotes the progression of PCa (Clinical cancer research, 2009, PMID: 19638457; Cancer research, 2019, PMID: 30952632; Cancer research, 2014, PMID: 24686169). Recently, PIM kinases have also been identified as key oncogenic drivers in PCa (Signal Transduct Target Ther, 2020, PMID: 32296034). Increasingly, inhibitors targeting phosphorylated kinases are being applied in the treatment of advanced prostate cancer patients (Trends Cancer, 2024, PMID: 38341319). In the present study, the enrichment of AR-related kinases and immune-related kinases in different subgroups respectively correlates with their proteomic features, potentially aiding in the risk stratification of PCa patients” in the Discussion part of revised manuscript. (page 14, line 308 to line 316).

Besides, we have actually performed the correction for multiple testing of KSEA results as we have done in the GSEA analysis. The kinases presented in **Figure 2D** were all $FDR < 0.05$. We are sorry about the inaccurate figure annotations which might confuse reviewers and readers. To clarify, we have adjusted the annotations and legends of **Figure 2D** as follows:

Revised Figure S2A. Kinase-Substrate Enrichment Analysis (KSEA) showed significantly up-regulated and down-regulated kinases in subgroup 1. Red bars refer to

FDR < 0.05 and z score > 0, blue bars refer to FDR < 0.05 and z score ≤ 0.

Revised Figure S2B. Kinase-Substrate Enrichment Analysis (KSEA) showed significantly up-regulated and down-regulated kinases in subgroup 3. Red bars refer to FDR < 0.05 and z score > 0, blue bars refer to FDR < 0.05 and z score ≤ 0.

Figure 2D. Kinase-substrate enrichment analysis (KSEA) showing significantly up-regulated and down-regulated kinases in subgroup 2 patients of the discovery cohort (subgroup 2 vs subgroup 1 and subgroup 3 combined). Red bars refer to FDR < 0.05 and z score > 0, blue bars refer to FDR < 0.05 and z score ≤ 0.

Q10. Fig 2SA caption: shouldn't it be ssGSEA?

Response:

Thank you for your precise evaluation of our work. We regret that we made a mistake in the caption of Fig 2SA in our original manuscript. To clarify, we have adjusted it as “ssGSEA of immune-related pathways among three proteomic subgroups”.

Q11. Fig. S2B,C,E, Fig.3A: Was correction for multiple testing performed here?

Response:

Thanks for your valuable comments. For **Figure S2B (Revised Figure S2C)**, correction for multiple testing has already been done, FDR-values shown in this figure were determined using Kruskal-Wallis test and B-H correction (N.S. $p > 0.05$; $*p < 0.05$ and $***p < 0.001$). The detailed P-values and FDR-values between groups were updated in the Source Data.

For the GSEA of metabolism-related pathways in **Figure S2C (Revised Figure S2E)** and **Figure 3A**, actually we have already performed the correction for multiple testing, which was not shown in the figure. According to the results, key pathways in metabolic subgroups were significantly enriched in corresponding proteomic subgroups with FDR < 0.05: linoleic acid metabolism pathway (P=0.0003, FDR=0.0003 in subgroup 1); arachidonic acid metabolism pathway (P=0.002, FDR=0.04 in subgroup 1); amino sugar and nucleotide sugar metabolism pathway (P=0.006, FDR=0.04 in subgroup 2). But there were still some metabolic pathways in the top10 list with FDR > 0.05 but P < 0.05, which we did not focus on in the following analysis and

experiments. Therefore, we did not present all FDR results in **Figure S2C (Revised Figure S2E)** and **Figure 3A**. The detailed GSEA results data (including P-value and FDR) of **Figure S2C (Revised Figure S2E)** and **Figure 3A** was updated in the Source Data.

For the volcano plot of **Figure S2E (Revised Figure S2G)**, we did not do the correction for multiple testing in the original manuscript. Based on your kind suggestion, we performed the correction for multiple testing of Figure S2E. According to the results, we updated the **Revised Figure S2G** in which the ALOX15 remained the most significantly differentially expressed protein between subgroup1 and other two subgroups (FC=1.77, FDR= 1.99e-06).

Revised Figure S2C. Single-sample gene set enrichment analysis (ssGSEA) of immune-related pathways among three proteomic subgroups in the discovery cohort. FDR values are determined using Kruskal-Wallis test and Benjamini-Hochberg (B-H) correction.

Revised Figure S2E. GSEA showing the top 10 up-regulated metabolic pathways in subgroup 1 patients of the discovery cohort (subgroup 1 vs subgroup 2 and subgroup 3 combined).

Figure 3A. GSEA showing the top 10 up-regulated metabolic pathways in subgroup 2

patients of the discovery cohort (subgroup 2 vs subgroup 1 and subgroup 3 combined).

Revised Figure S2G. Volcano plot showing higher expression of ALOX15 protein in subgroup 1 compared to subgroup 2 and 3 of the discovery cohort. FDR values are determined using two-sided Wilcoxon signed-rank test and Benjamini-Hochberg (B-H) correction.

Q12. Fig. S2F: What are "SEG1-3" in color legend?

Response:

Thank you for your precise evaluation of our work. We regret that our original figure legend of Figure S2F may confused the reviewer and readers. “SEG” is the abbreviation for “subgroup enriched genes”. To clarify, we have adjusted the figure legend of Figure S2F as “The 39-protein subtyping panel for proteomic subtyping of PCa filtered by the Random Forest algorithm (SEG: subgroup enriched genes)”.

Figure S3B. The 39-protein panel for proteomic subtyping of PCa filtered by the Random Forest algorithm, SEG represents subgroup enriched genes.

Q13. page 9, lines 159-161: "This cohort could be successfully subtyped into three proteomic subgroups by a 39-protein subtyping panel, which filtered by the Random Forest algorithm and validated in our cohort (Figure S2F)." - Please provide more details on how the panel was constructed and validated, and explain how the values shown in the heatmap in Fig. S2F were obtained.

Response:

Thanks for your professional comments. Based on your suggestions, we have adjusted the detailed description of 39-protein subtyping panel in the manuscript as “To verify our findings, we firstly constructed a 39-protein subtyping panel, which was filtered by the differential expression analysis and the Random Forest algorithm in the discovery cohort (Figure S3A-C). Then, we additionally employed 119 PCa patients from other centers (details provided in the Methods section) as the validation cohort 1. Based on the proteomic profiling data and the 39-protein subtyping panel, the validation cohort 1 could be successfully categorized into three proteomic subgroups consistent with those identified in the discovery cohort (Figure 3D, Figure S3A)”. (page 8, line 174 to line 180) Besides, we added a flowchart to detailly illustrate the pipeline for the construction and validation of the 39-protein subtyping panel (**Figure S3A**).

The detailed method for the construction and validation of the 39-protein subtyping panel was presented in the Method part: Prior to select protein biomarkers, differential analysis (Wilcoxon test) was performed between one subtype and other two subtypes (considered as one group). $FC \geq 1.4$ & $P\text{-adjusted} < 0.05$ was considered as significant difference. Next, the top 50 differentially expressed proteins were subjected to the following analysis. Random forest algorithm was used to find potential biomarkers. Proteomics data was divided into training and test datasets with 7:3 ratio and cumulative AUC was applied to select minimum number of biomarkers with training dataset (10-fold cross validation). Finally, the model performance was measured using the test dataset. To validate our proteomic subtyping results in the validation cohort 1, the nearest template prediction (NTP), an unsupervised clustering algorithm, was adopted using the 39-protein panel filtered by machine learning. To further analyze the biological characteristics of these subgroups, we performed Gene Set Enrichment Analysis (GSEA) to identify the pathway alterations that underlying each subgroup. (page 28, line 620 to line 626)

The raw data for proteomics analysis have been uploaded to the database according to the submission requirements, and the accession code was added to the Data Availability part as “The raw proteomics and phosphoproteomics data generated in this study is publicly available in the iProX database under the accession ID [IPX0009562000](https://www.iprox.org/entry/10009562000)”.(page 37, line 827 to line 828) The code for the construction and validation of 39-protein subtyping panel (including differential expression analysis and the Random Forest analysis) are publicly available in Github at <https://github.com/Diluczhang/Proteome-of-PCA.git>.

Revised Figure S3A. Design of the analytical pipeline for construction and validation of the 39-protein subtyping panel.

Revised Figure S3B. The 39-protein panel for proteomic subtyping of PCa filtered by the Random Forest algorithm, SEG represents subgroup enriched genes.

Revised Figure S3C. The Receiver Operating Characteristic Curves (ROC) of the 39-protein subtyping model in the validation dataset.

Revised Figure 3D. Kaplan-Meier curves showing the bRFS of patients in three proteomic subgroups of the validation cohort 1. P-value is calculated by the Log-rank test.

Q14. Fig. 3C caption: "expression of N-Acetylneuraminic acid" - the term "expression" is usually applied to genes, transcripts, and proteins.

Response:

Thank you for your careful evaluation and we regret the confusing presentation. Based on your suggestion, the **Figure 3C** caption has been described as "Comparison of the relative abundance of N-Acetylneuraminic acid among three proteomic subgroups based on the metabolomics data" instead of "Expression Level" (page 48 to page 49, line 1140 to line 1141).

Figure 3C

Figure 3C. Comparison of the relative abundance of N-Acetylneuraminic acid among three proteomic subgroups of the discovery cohort based on the targeted metabolomics analysis. The middle lines represent the median, and the lower and upper hinges denote the 25-75% IQR, with whiskers extending up to a maximum of 1.5 times IQR. P-values are determined using the two-sided Wilcoxon rank-sum test.

Q15. Fig. 3E: The caption does not describe the figure.

Response:

Thank you for your careful evaluation and we regret the confusing caption. Based on your advice, the **Figure 3E** caption has been adjusted as "GSEA showing the up-regulation of the amino sugar and nucleotide sugar metabolism pathway in subgroup 2 patients of the validation cohort 1 (subgroup 2 vs subgroup 1 and subgroup 3 combined)." (page 49, line 1147 to line 1149).

Figure 3E

Figure 3E. GSEA showing the up-regulation of the amino sugar and nucleotide sugar metabolism pathway in subgroup 2 patients of the validation cohort 1 (subgroup 2 vs subgroup 1 and subgroup 3 combined).

Q16. Fig. 4B: Please provide more details on how this figure was obtained (specify algorithm, its input and parameters) and what the cell color corresponds to.

Response:

Thanks for your valuable comments. The proteomic classification of cell lines was performed by application of the 39-protein subtyping panel to the newly generated proteomic data of these cell lines. We used 8 kinds of PCa cell lines and 3 biological replicates of every cell line for proteomic profiling. The generated proteomic data of all cell lines were uploaded to the database according to the submission requirements. The algorithm used for cell lines subtyping is Nearest template prediction (NTP). Its an unsupervised clustering algorithm to make category predictions, using only a gene list and a test dataset to evaluate the prediction confidence calculated in each patient's gene expression data. The input data is the proteomic expression matrix of 8 PCa cell lines. The detailed code including parameters were publicly available in Github at <https://github.com/Diluczhang/Proteome-of-PCA.git>. Besides, the cell color corresponded to different proteomic subgroups.

Figure 4B. Heatmap visualizes the proteomic subtypes of 8 PCa cell lines based on the nearest template prediction (NTP) algorithm. Three replicates for each cell line.

Q17. lines 183-194 and Fig. 4 C-G: please introduce sgNANS in the main text and in the figure caption.

Response:

Thank you for your careful evaluation and we regret the omission of the description here. Based on your suggestion, we have added the description of sgNANS as “NANS-knockout PCa cells were constructed based on the corresponding single guide RNA (sgRNA)” in the revised manuscript (page 10, line 214 to line 215). The sgRNA sequences were listed in Method part: The sgRNA sequences for 22Rv1 cells: sgNANS-1 (5'-TATGTGACGTTCCAACACCT-3'), sgNANS-2 (5'-TCATGCCAGAATACCCTAT-3'). The sgRNA sequences for Myc-CaP cells: sgNans-1 (5'-TCGTGCCCGGAATACCCGAT-3'), sgNans-2 (5'-GGGCTGTAGTGGGTACGCGC-3'). (page 33, line 754 to line 758) Besides, the figure legend of Figure 4C was adjusted as “Knockout (KO) of NANS in 22Rv1 and Myc-CaP cells via single guide RNA (sgRNA) as determined by qRT-PCR. Empty

vector is used as control. P-values are determined using two-tailed Student's t test.”. (page 51, line 1180 to line 1182).

Q18. Fig. 4E: The color scale is not labeled. Not fully clear what the rows of the heatmap correspond to.

Response:

Thank you for your careful evaluation and we regret the confusing caption of **Figure 4E (Revised Figure 4F)**. The rows of the heatmap present the differentially expressed sialylated sites between the control group and *Nans* knockout group. Based on your suggestions, we have labeled the color scale and adjusted the figure legend as “Differentially expressed sialylated sites in Myc-CaP cells between *sgNans* and control group”.

Revised Figure 4F. The heatmap showing differentially expressed sialylated sites in Myc-CaP cells between *sgNans* and control group.

Q19. page 9, lines : "Recent research has suggested the sialoglycan-Siglec axis as a novel immune checkpoint which can be targeted to augment the anti-tumor immune response." - please add the reference.

Response:

Thank you for your careful evaluation and we regret the omission of the reference here. Based on your suggestions, we have added the reference (Annual Review of Immunology, 2020, PMID: 31986070; Cancer Immunology Research, 2022, PMID: 36264237) in the manuscript. (page 10, line 221 to line 223)

Q20. page 9, lines 200-202: To dissect the immune microenvironment of PCa, the EcoTyper analysis was performed and revealed increased infiltration of M2 macrophages in subgroup II (Fig.4H) - please specify which cohort was used in this

analysis. The same concerns Fig. 4I.

Response:

We thank the reviewer's professional comment and regret to make you confused. The EcoTyper analysis in **Figure 4H (Revised Figure 4G)** and the immunofluorescence (IF) staining in **Figure 4I (Revised Figure 4H)** were both performed in the discovery cohort. To clarify, we have adjusted the description of these two figures in the manuscript as “To dissect the immune microenvironment of PCa, the EcoTyper analysis was performed in the discovery cohort and revealed that increased infiltration of M2 macrophages in subgroup II (Fig. 4G). In line with this finding, multiplex immunofluorescence (IF) staining in the discovery cohort further confirmed the immunosuppressive microenvironment with increased infiltration of M2 macrophages and decreased infiltration of CD8+ T cells in the subgroup II (Fig. 4H)”. (page 10, line 223 to line 228)

Revised Figure 4G. Immune microenvironment analysis of three proteomic subtypes by EcoTyper in the discovery cohort..

Figure 4H

Revised Figure 4H. (upper) Representative multiplex immunofluorescence (IF) staining images of M2 macrophages and CD8+ T cells among three proteomic subtypes in the discovery cohort. The scale bar represents 50um. (bottom) Bar plots shows the mean intensity of IF stainings. P-values were determined using two-tailed Student's t test.

Q21. Figs. 5F,G: Please explain how exactly these figures were obtained and label the color scale.

Response:

Thank you for your precise evaluation of our work. We regret that our original presentation on **Figure 5F** and **5G** may confused the reviewer and readers. **Figure 5F** was the dot plot depicting the expression of detected Siglec genes in each identified cell subtype, where dot size and color represent percentage of marker gene expression (pct. exp) and the averaged scaled expression (avg. exp. scale) value, respectively. This figure as applied by gene expression based on the Seurat Log normalization method and the gene expressed rate calculated by Seurat function. The shade of the red color in dot plot represented the average expression of specific genes and the size of the dot represented the percent expression of specific genes. **Figure 5G** was the heatmap depicting the enrichment of classical immune-related pathways in each identified macrophage cell cluster. This figure was applied by the gene activity score calculated based on the gene expression and Qusage package which was described in method and data was scaled by row for visualization. Based on your suggestions, we have adjusted the color scale of **Figure 5F (Revised Figure 5H)** and **Figure 5G (Revised Figure 5I)** as follows:

H

Revised Figure 5H. Dot plot depicting the expression of detected Siglec genes in each identified cell subtype, where dot size and color represent percentage of marker gene expression (pct. exp) and the averaged scaled expression (avg. exp. scale) value, respectively.

Revised Figure 5I. Heatmap depicting the enrichment of classical immune-related pathways in each identified macrophage subtype.

Q22. lines 257-258: "26% of PCa cannot be classified³¹". - according to reference 31, a driver abnormality was not identified in 26% of tumors, but these tumors can still be classified based on mRNA expression.

31: "Despite this detailed molecular taxonomy of primary prostate cancers, 26% of all tumors studied appeared to be driven by still-occult molecular abnormalities or by one or more frequent alterations that co-occur with the genomically defined classes."

Response:

Thank you for your precise evaluation of our work. We regret that our original statement about the reference may confused the reviewer and readers. Based on your suggestions, we have adjusted the description as "For example, seven localized PCa subgroups have been genomically defined by fusions of ETS family genes such as *ERG*, *ETV*, *ETV4* or *FLII*, or mutations in *SPOP*, *FOXA1* or *IDH1* from the Cancer Genome Atlas (TCGA) database. However, 26% of PCa still cannot be classified with these seven common

genomic alterations". (page 13, line 279 to line 282)

Q23. page 26, line 587: Why TCR-enriched libraries were used in single-cell sequencing protocol?

Response:

Thank you for your precise evaluation of our work. We regret that our original presentation may confused the reviewers and readers. We have actually conducted the V(D)J/TCR sequencing when performing the single-cell sequencing. But in the present study, the V(D)J/TCR sequencing data was indeed not involved. It was a written mistake to involve the method of V(D)J/TCR sequencing in the original manuscript. To obviate misunderstanding, we have removed the content about V(D)J/TCR sequencing in Methods part of the revised manuscript.

Typos/mistakes:

line 209: "In compared ..."

lines 212-213: "cells in the absent/present of Nans"

line 232: "to remoulade"

line 265: "In the current study, we defined PCa into three distinct subgroups"

lines 376,530: "metabonomic"

line 651: "EcoTyper were constructed"

line 605: "in default parameters"

line 610: "Nomal-panal"

line 634: "significantly different expressed genes."

line 663: "PCa", should be "PCs"

Response:

Thank you for your precise evaluation of our work. Based on your suggestions, we have adjusted these grammatical mistakes in the corresponding places of the manuscript.

Reviewer #2 (Remarks to the Author):

Q1、 The analysis of proteomics data revealed three subgroups of tumors/patients, with metabolism pathways enrichment subgroup 1 and subgroup 2 and immune-related pathways subgroup 3 that was stated to have exhibited a more 'hot' overall immune profile, with significant upregulation of immune activation pathways such as pro-inflammation and lymphocyte infiltration. It was also shown that these subgroups have different survivals with group 2 being the most aggressive. However, it is hard to make a complete picture of how the proteomic based subgroups relate to immune cells infiltration and levels of NANS.

Response:

Thanks for your constructive comments. In our manuscript, we firstly used the NMF-clustering to stratify PCa patients into three proteomic subgroups. Interestingly, these three subgroups exhibited diverse prognosis and subgroup 2 has the poorest prognosis. Furthermore, we employed GSEA analysis to define three proteomic subgroups as the arachidonic acid metabolic subgroup 1, sialic acid metabolic subgroup 2 and immune subgroup 3. Among them, the immune subgroup 3 presented a more 'hot' overall immune profile with activation of immune-related pathways such as pro-inflammation and lymphocyte infiltration. Meanwhile, the most aggressive subgroup 2 showed the activation of the sialic acid synthesis pathway and higher NANS protein expression, which is the key enzyme of sialic acid synthesis. Strikingly, the IF staining results showed increased sialylation level and immunosuppressive microenvironment in subgroup 2 tumors. High tumor sialylation level has been reported to be relevant to immunosuppressive microenvironment and unfavorable prognosis in many malignancies (*Nature communications*, 2021, PMID: 33627655; *Sci Transl Med*, 2022, PMID: 36322632). Therefore, we used *in vitro* and *in vivo* experiments to validate whether NANS-induced tumor sialylation resulted in immunosuppressive microenvironment and tumor development of PCa. Our *in vitro* experiments found that NANS-knockdown in PCa cells exerts no direct influence on tumor proliferation, indicating the essential involvement of other components in tumor microenvironment. Furthermore, our *in vivo* experiments in Myc-cap cell-derived orthotopic transplanted mouse model and *Pten*^{PC-/-}; *Trp53*^{PC-/-} mouse model both confirmed that higher NANS expression enhanced tumor cell sialylation, which in turn led to the immunosuppressive microenvironment via sialoglycan-Siglec axis and promotes tumor progression in PCa.

Q2、 The new IF data shown in figure 4H indicate immune infiltration in subgroup 1 being rather similar to subgroup 3 and low immune cell presence only in subgroup 2 which is somewhat in contract to proteomics assignment of the groups with only subgroup 3 exhibits higher activities of immune -related pathways. These results need to be discussed in relation to groups assignment and potential biological differences in alteration of immune infiltration.

Response:

We thank the reviewer for bringing this issue to our attention. The IF staining results showed the suppressive immune microenvironment and reduced infiltration of CD4+ and CD8+ T cells in subgroup 2 tumors. While, the IF staining actually indicated no significant difference of immune cell infiltration between subgroup 1 and 3 tumors (**Figure 4H**). Actually, GSEA analysis based on the proteomic data indicated the activation of immune-related pathways in subgroup 3 (such as the Interferon-gamma response, Interferon-alpha response and TNF- α signaling pathway), which suggested the enhanced immune function of immune cells in subgroup 3 than subgroup 1 via producing more functional cytokines (**Figure 1D**). To further address this question, we performed additional IHC staining of functional markers of immune cells (GZMB, INF- γ , TNF- α) in subgroup 1 and 3 tumors using the consecutive paraffin sections of patients in Figure 4H. The IHC results showed higher intensity of functional markers of immune cells, which indicates the enhanced immune activity in immune subgroup 3 than subgroup 1 (**Response Figure 1**). Collectively, despite the similar numbers of infiltrated immune cells in subgroup 1 and subgroup 3, the immune activity of immune cells are stronger in subgroup 3 than that in subgroup 1, which is consistent with our proteomics results.

Figure 4H. Multiplex immunofluorescence (IF) staining and quantification of immune cells in three proteomic subgroups of the discovery cohort. n=10 for each group. The scale bar represents 50um. Data are presented as mean \pm SD. P-values are determined using two-tailed Student's t test.

Figure 1D. Gene sets enrichment analysis (GSEA) showing distinct molecular characteristics among three proteomic subgroups of the discovery cohort (one subgroup vs the union of other two subgroups).

Response Figure 1. Immunohistochemistry (IHC) staining and quantification for GZMB, INF- γ and TNF- α in tumor regions of subgroup 1 and subgroup 3 patients. The

scale bars represent 500/50um in low/high-power field. Data are presented as mean \pm Standard deviation (SD). P-values are determined using two-tailed Student's t test.

Q3、It is also somewhat difficult to make association between the proteomics subgroups with NANS levels as the newly presented data (levels of NANS in adjacent normal and tumor samples in the whole cohort and in the subgroups Figure 3 and Sfigure3), show that levels of NANS are rather similar in group 1 and 2 tumors.

Response:

We appreciate the reviewer's kind evaluation of our work. To detailly address this question, we compared the NANS protein expression level of tumor tissues between subgroup 1 and subgroup 2 patients. The results showed higher NANS protein expression in subgroup 2 tumors than that in subgroup 1 tumors, corrected by corresponding NANS expression of adjacent normal tissues (**Revised Figure 3F**). Besides, IHC staining results further confirmed our proteomic data that subgroup 2 tumors had higher NANS protein expression than subgroup 1 and 3 tumors (**Figure 3G**). We have adjusted the description of NANS expression in the Results part as "To this end, our proteomic data firstly showed that NANS expression was significantly elevated in tumor tissues compared to paired adjacent normal tissues in the discovery cohort, especially in subgroup 2." (page 9, line 186-188)

Revised Figure 3F. Comparison of relative NANS protein abundance among tumors of three subgroups in the discovery cohort, corrected by corresponding NANS expression of adjacent normal tissues. The middle lines represent the median, and the lower and upper hinges denote the 25-75% IQR, with whiskers extending up to a maximum of 1.5 times IQR. P-value is determined using the two-sided Wilcoxon rank-sum test.

Figure 3G. Immunohistochemistry (IHC) staining and quantification for NANS in tumor regions of three proteomic subgroups in the discovery cohort (n=145). The scale bar represents 100µm. Data are presented as mean \pm Standard deviation (SD). P-values are determined using two-tailed Student's t test.

Q4、 Last but not least, the authors state that subgroup 2 exhibits higher activity of androgen receptor signaling when compared to combined subgroups 1 and 3. But is the AR signaling different between subgroups 1 and 2?

Response:

Thanks for your constructive comments. Following the reviewer's recommendation, we further performed the GSEA analysis of Hallmark-Androgen Response pathway. The results showed that subgroup 2 tumors exhibited higher activity of androgen receptor signaling than subgroup 1 tumors (FDR=1.606, P=0.008, **Response Figure 2**).

Response Figure 2. GSEA showing the activation of the androgen response pathway in subgroup 2 tumors compared to subgroup 1 tumors.

Q5、 The authors addressed the potential concordance/discordance between the

proteomics and transcriptomics subgroups. However; what would be of interest is to evaluate transcriptome of the proteomics subgroups.

Response:

Thanks for your constructive comments. Based on your suggestion, we performed the GSEA analysis in the transcriptomic level of the three proteomic subgroups. Proteome-based stratification of localized PCa revealed three subgroups with distinct molecular features: immune subgroup, arachidonic acid metabolic subgroup and sialic acid metabolic subgroup. Interestingly, the global transcriptomic molecular features of the three proteomic subgroups were similar to their proteomic molecular features. Specifically, in the transcriptomic level, the immune subgroup 3 was also enriched in classical immune-related pathways, such as interferon- γ response, interferon- α response, complement pathways (**Response Figure 3**). Meanwhile, the metabolic subgroup 1 was enriched in proliferation-related pathways and lipid metabolism pathways, especially the arachidonic metabolism pathway (**Response Figure 3**). The metabolic subgroup 2 was enriched in the androgen response pathway, proliferation-related pathways, sugar and glycogen metabolism-related pathways, especially the sialic acid synthesis pathway (**Response Figure 3**). However, there were still some discordances between the transcriptomic and proteomic features of three proteomic subgroups. For instance, in the transcriptomic level, the immune subgroup 3 also enriched in several lipid metabolism pathways and the lipid metabolic subgroup 1 also exhibited a certain degree of glucose metabolic characteristics. Consequently, the concordance/discordance between global transcriptomic and proteomic molecular features of different subgroups highlight the complex signal coordination in the regulation of gene expression and activities in PCa.

Response Figure 3. Gene sets enrichment analysis (GSEA) showing distinct transcriptomic molecular characteristics among three proteomic subgroups of the discovery cohort.

6、 *Based on additional analysis using data from TCGA, the authors concluded that results from Chinese patients are not recapitulated in Caucasian cohort data available. However, the manuscript includes evaluation of a panel of 8 human PCa cell lines, and their stratification into three proteomic subtypes using the nearest template prediction (NTP) algorithm similarly to human PCa tissues (Fig 4B). These assignments of the subgroups do not correspond to the aggressiveness assignment to the clinical samples, as PC-3 and DU 145, aggressive prostate cancer cell lines do not fall into the aggressive category (PC-3 subgroup 1 and DU 145 subgroup 3). Similarly, it does not appear that cell lines assigned to subgroup 2 have higher AR signaling, and levels of NANS for these cell lines were not provided. In light of these results, it is not clear how to incorporate the results using 22Rv1, where NANS knockdown in 22Rv1 similarly to Myc-CaP alters sialylation. Also, the sialylated sites altered appear to be different between these two cell lines. These issues need to be addressed.*

Response:

Thank you for your insightful feedback and I greatly appreciate the time and effort you invested in the work. In our proteomic subtyping results of cell lines, PC3 and DU 145 cells did not fall into the aggressive subgroup 2. We consider the reasons for this may attribute to the following aspects: Firstly, PC3 cells originate from bone metastases and DU 145 cells originate from brain metastases. While, the 39 signature proteins used for cell line subtyping in our manuscript were generated from the localized PCa. The tumor biology and molecular features between localized and metastatic PCa are widely different. Therefore, the 39 signature proteins may only account for part of tumor characteristics of metastatic cell lines (PC3 and DU 145), which lead to the present proteomic subtyping results. While, 22Rv1 cells originate from prostate tissues rather than metastases so that this cell line can adequately reflect the proteome of primary PCa. Secondly, the proteomic subtyping of eight PCa cell lines was performed using the nearest template prediction (NTP) algorithm, which was mainly based on the expression pattern of 39 signature proteins, rather than only based on the aggressiveness of cell lines. PC-3 and DU145, aside from their shared high aggressiveness, may also exhibit differences in other tumor biology aspects such as metabolism, proliferation and oxidative stress. This may also explain why they were stratified into subgroup 1 and subgroup 3, respectively.

The major molecular feature of subgroup 2 tumor is the activation of the sialic acid synthesis metabolism. Meanwhile, we found the AR signaling activity was generally higher in subgroup 2 tumors compared to subgroups 1 and 3 tumors. However, when we checked the status of AR signaling in each subgroup 2 patients, we found there were still some patients in subgroup 2 exhibiting low AR signaling (**Response Figure 4**). This phenomenon indicated that the 39-protein subtyping panel we used for cell lines was not strictly AR-related. Therefore, we considered the different AR signaling status in subgroup 2 cell lines may also attributed to the complex signal coordination in the tumor microenvironment of PCa. These observations warrant further exploration in future research.

Furthermore, based on your kind suggestion, we added a comparison of NANS protein expression across different cell lines by Western blot (**Response Figure 5**). The results showed that the NANS expression levels in the Myc-CaP and 22Rv1 cell lines, which correspond to subgroup 2, are indeed higher. Combining the NTP stratification results with NANS expression data, we propose that Myc-CaP and 22Rv1 cells exhibit greater concordance with the tumor expression profiles of subgroup 2 patients, rendering them appropriate models for subsequent target validation studies.

Moreover, the changes of sialyated sites in the two cell lines were inconsistent after NANS deletion. This occurs because NANS functions as a sialic acid synthase, and after its deletion, the reduction in sialic acid synthesis leads to random effects on downstream sialyated sites. Besides, the two cell lines originate from different species (Myc-CaP is mouse-derived, whereas 22Rv1 is human-derived), which reasonably explains the observed differences in the sites of reduced sialic acid modification.

Response Figure 4. Comparison of AR score (the transcriptomic activity of AR) in three proteomic subgroups and adjacent normal samples.

Response Figure 5. Comparison of NANS protein expression among 8 PCa cell lines by Western blot.

Q7. In the revised manuscript, the authors introduced data with Pten/Trp53 double knockout (PtenPC^{-/-} ; Trp53PC^{-/-}) mouse model, and they state that this model partially recapitulates the features of aggressive PCa. It is this reviewer's opinion that characteristics of this model, relevant to the topic of this manuscript, should be provided in more details than the statement "partially recapitulates" characteristics of aggressive cancer. For example, does this model exhibit subgroup 2 characteristics and

what are the levels of NANS in these tumors?

Response:

Thanks for your critical comments. We fully agree with the reviewer's suggestion that it's necessary to provide more details on the subgroup 2 characteristics of the *Pten*^{PC-/-}; *Trp53*^{PC-/-} mouse model. To address this question, the proteomic profiling of the formalin fixed paraffin-embedded (FFPE) tissue samples of *Pten*^{PC-/-}; *Trp53*^{PC-/-} mouse model and Myc-cap cell-derived orthotopic transplanted mouse model in the *Nans* knockout (KO) group and control groups were performed. The proteomic matrix of these twelve tumor samples were deposited in the Source Data. Based on the 39-protein subtyping panel, these tumor samples were classified into three proteomic subgroups by the Nearest template prediction (NTP) algorithm. Similar to the subtyping results in Myc-cap cell-derived mouse model, 3 tumor samples from the control group of *Pten*^{PC-/-}; *Trp53*^{PC-/-} mouse model were divided into subgroup 2, while 3 tumor samples from the *Nans*-KO group were divided into subgroup 1 and 3 (**Revised Figure S6A**). Besides, GSEA analysis and Western blot further confirmed the subgroup 2 molecular features, higher expression of NANS protein and the upregulation of the sialic acid synthesis pathway in the control group of *Pten*^{PC-/-}; *Trp53*^{PC-/-} mouse model (**Response Figure 6-7**). Consequently, the *Pten*^{PC-/-}; *Trp53*^{PC-/-} mouse model actually presented the molecular characteristics of subgroup 2 PCa, which could be used in the *in vivo* experiments to recapitulate aggressive PCa.

Based on your suggestion, we added the detailed description of *Pten*^{PC-/-}; *Trp53*^{PC-/-} mouse model in the Results part as “To further validate our findings, we next constructed the *Pten/Trp53* double knockout (*Pten*^{PC-/-}; *Trp53*^{PC-/-}) mouse model, which partially recapitulates features of subgroup 2 PCa confirmed by the proteomic subtyping results.” (page 12, line 259-262)

A

Revised Figure S6A. Heatmap visualizes the proteomic subtype assignments of murine tumor tissues of the control and *Nans* knockout (KO) group in *Pten*^{PC-/-}; *Trp53*^{PC-/-} and Myc-cap cell-derived mouse models. Three tumor samples for each group.

Response Figure 6. Comparison of NANS protein expression between the control group and *Nans*-KO group in *Pten*^{PC-/-} ; *Trp53*^{PC-/-} mouse tumor tissues by Western blot.

Response Figure 7. GSEA showing upregulation of the sialic acid synthesis pathway in the control group compared to the *Nans*-KO group in *Pten*^{PC-/-} ; *Trp53*^{PC-/-} mouse tumors.

Q8. While the point how was the cut off for assignment to NANS low and NANS high groups selected was addressed, it is not clear whether the cut off from the discovery cohort was used in the validation cohorts or whether the cut offs were calculated was each cohort independently, as the rebuttal states “the comparison of NANS expression was performed based on the entire discovery cohort or corresponding validation cohorts”. If different cut off was calculated for each cohort, that would significantly decrease translational relevance of the results.

Response:

We appreciate the reviewer’s kind evaluation of our work. We totally understand the reviewer’s concern about the consistency of NANS proteomic cut-off point. In our manuscript, proteomic data of NANS for the discovery cohort and the validation cohort 1 originated from diverse tissue samples, with fresh snap-frozen tissue samples used for the discovery cohort and formalin fixed paraffin-embedded (FFPE) tissue samples used for the validation cohort 1. Because there are large differences in global protein expression of FFPE samples and fresh samples, the distribution of NANS expression in two datasets was so different that a uniform cut-off point for NANS expression could not be directly applied in two cohorts (**Response Figure 8**). We attempted to eliminate the scale differences of NANS proteomic data between the discovery and validation cohorts by using various normalization methods. However, the distribution differences of NANS expression still remain after data standardization due to different sample sources. Therefore, we considered the selection of independent cut-off points in two cohorts to be necessary to ensure the biological rationality of patient assignment within each cohort. Despite different cut-off points for NANS expression, our results indicated consistent prognostic differences between high and low NANS expression subgroups in two cohorts, which satisfied the core requirements of the validation. The survival analysis of NANS in the discovery and validation cohort 1 were based on their

corresponding proteomic data. In addition, we also performed the survival analysis based on the IHC staining of NANS in the validation cohort 2, which further verified the prognostic implication of NANS protein.

In our manuscript, we firstly revealed the prognostic implication of NANS protein in the discovery cohort based on the proteomic data from fresh snap-frozen PCa samples. Then we retrospectively collected the FFPE tissue samples of PCa patients with long-term follow-up data from other centers to further validate the prognostic implication of NANS. In consideration of the limited follow-up period, the prospectively collected fresh snap-frozen PCa samples were not included in the present manuscript. Based on your kind suggestion, we plan to prospectively collect fresh snap-frozen PCa samples with adequate follow-up time from other centers to further verify the prognostic implication of NANS protein.

Response Figure 8. The distribution of NANS protein expression in the discovery and validation cohort 1.

Q9、 *One of the previous comments was why Myc-CaP cells were used based on proteomic subtyping if the intention was to evaluate the impact of NANS on the tumor immunophenotype, and that it would be of considerable interest to evaluate the impact of NANS regardless of the subgroup. This comment was not addressed. The response was justification of why Myc-Cap was used but no additional data were provided about impact of NANS on tumor immunophenotype regardless of the subgroups.*

Response:

We appreciate your valuable comments. According to your suggestion, we examined the expression levels of NANS by Western blot across 8 PCa cell lines and found a strong correlation between NANS expression and proteomic subgroups. Specifically, NANS expression is higher in subgroup 2 cell lines (22Rv1, C4-2B, Myc-cap) compared to cell lines in other two subgroups (**Response Figure 5**). Additionally, we utilized another mouse-derived PCa cell line, RM1, which belongs to subgroup 1, to establish the orthotopic transplanted mouse model. Interestingly, *NANS* depletion in the RM1-derived orthotopic model did not lead to notable alteration of the tumor immune microenvironment (**Response Figure 9**). This result suggests that targeting NANS in

non-subgroup 2 tumors has minimal impact on their immunophenotype. We propose the reasons for this observation are as follows: Firstly, non-subgroup 2 tumors exhibit relatively low expression of NANS. Thereby, deletion of NANS does not induce significant biological alterations in these tumors. Secondly, NANS-based regulation of tumor immune microenvironment is via the sialic acid metabolism and tumor sialylation level, which are not prominent in non-subgroup 2 tumors. Consequently, targeting NANS presents a promising anti-tumor therapy in aggressive PCa with features of sialic acid metabolism. We hope these results may provide insights on the subgroup-specific role of NANS in modulating the tumor immune microenvironment and address your query. Thank you for bringing up this important question.

Response Figure 5. Comparison of NANS protein expression among 8 PCa cell lines by Western blot.

Response Figure 9. Immunofluorescence staining and quantification of immune cells in murine tumor tissues of the *sgNans* and control group. $n=3$ for each group. The scale bar represents 50 μ m. Data are presented as mean \pm SD. P-values are determined using two-tailed Student's t test.

Q10, The newly provided single cell analysis, showing three control and three Nans knockdown tumors, shows that three control tumors are rather separated into different clusters (suppl figure 5C) any explanation of this separation of control tumors?

Response:

Thank you for your insightful question. After detailed data check, we are sincerely sorry to present the initial version of Supplementary Figure 5C before batch correction in the original manuscript. Actually, we have done the batch correction in all the following scRNA-seq data analysis and other scRNA-seq figures presented in the manuscript were based on the data after batch correction. Based on your kind suggestion, we presented the adjusted version of Supplementary Figure 5C after batch correction (**Revised Supplementary Figure 5C**). The analysis revealed that even after batch correction, tumor epithelial cells exhibit some degree of heterogeneity. Previous studies have demonstrated that cell plasticity under different environmental conditions contributes to this variability. Alterations in the tumor microenvironment, alongside genotypic factors, conjointly determine tumor cell fate and transcriptional states (*Cell*, 2021, PMID: 34890551). We propose that this phenomenon is largely driven by host-specific individual variations after cell lines transplanted. Besides, marker genes of tumor epithelial cells included *Wfdc12*, *Cldn3* and *Hspb1* from another scRNA-seq research of PCa, which also used the Myc-cap cell-derived transplanted mouse model (*Nature Communications*, 2024, PMID: 39406723, **Response Figure 10**). In our scRNA-seq data, these marker genes were also significantly highly expressed in tumor epithelial cells, which indicated the tumor biological homogeneity across different research (**Response Figure 11**). Consequently, we speculated that host-specific individual variations mainly contributed to the heterogeneous clustering among tumor cells in the control group.

Revised Supplementary Figure 5C. UMAP visualization of 29515 single cells, (left panel) colored by sample origin (right panel) colored by group origin (control group or Nans KO group).

Response Figure 10. Marker gene expression of tumor epithelial cells of Myc-cap cell-derived transplanted mouse model in Nature Communications.

Response Figure 11. Marker gene expression of tumor epithelial cells of Myc-cap cell-derived transplanted mouse model.

Reviewer #4 (Remarks to the Author):

I would like to thank the authors for revising the manuscript and for addressing most of my comments. While the manuscript has shown improvement after the revision, I still have some questions regarding the data availability and code provided to replicate the computational analysis. Since this is essential for the reproducibility and transparency of the research, I cannot recommend this manuscript for publication until these questions are resolved.

Q1 、 The authors state that: "The raw proteomics and phosphoproteomics data generated in this study is publicly available in the iProX database under the accession ID IPX0009562000" "The whole exome sequencing (WES) and RNA-seq raw data that support the findings of this study have been deposited in the Genome Sequence Archive (GSA) under accession ID HRA008293". However, I could not find either IPX0009562000 on <https://www.iprox.cn/>, or HRA008293 on <https://ngdc.cncb.ac.cn/gsa/> .

Response:

Thank you for your attention and we're sincerely sorry for causing any confusion. Since we have set these datasets to be public after the paper is published, reviewers are currently unable to find them using the accession IDs. We have generated new links

through the website for reviewers to access in advance (The raw proteomics and phosphoproteomics data: <https://www.iprox.cn/page/PSV023.html?url=1736302661405L6ko>; Password: rFK8, the WES and RNA-seq raw data: <https://ngdc.cncb.ac.cn/gsa-human/s/dR6LUGj2>). Furthermore, we affirm that the data will be designated as 'public' concurrently with the publication of the paper, thereby ensuring its accessibility to all researchers through the accession IDs (IPX0009562000 and HRA008293).

Q2、 Also, I could not find any link to the raw metabolomics data used in this study. To ensure data availability, I recommend the authors provide direct links for each dataset used in this study.

Response:

Thank you for your rigorous work. We fully agree that direct links facilitate more efficient access to data. We have conducted a thorough review of all the data and provide direct links of each dataset used in this study for reviewers to access in advance as below:

PCa patients:

The raw proteomics and phosphoproteomics data:

<https://www.iprox.cn/page/PSV023.html?url=1736302661405L6ko>, Password: rFK8

The WES and RNA-seq raw data:

<https://ngdc.cncb.ac.cn/gsa-human/s/dR6LUGj2>

The raw metabolomics data:

<https://ngdc.cncb.ac.cn/omix/preview/7s2OX3fe>

The RNA-seq matrix:

<https://ngdc.cncb.ac.cn/omix/preview/xKwuJZE7>

PCa cell lines:

The raw proteomics data:

<https://www.iprox.cn/page/PSV023.html?url=173630308090915ZY>, Password: 7HeB

The raw metabolomics data:

<https://ngdc.cncb.ac.cn/omix/preview/J6fxWY6S>

PCa orthotopic transplanted mice:

The raw single-cell RNA sequencing data:

<https://ngdc.cncb.ac.cn/gsa/s/JXy16uIZ>

The raw proteomics data:

<https://www.iprox.cn/page/PSV023.html?url=17363032882896maZ> Password: gKPU

** Due to the limitations of the websites, the links for reviewers to access in advance are valid for approximately 90 days. Should the links have expired by the time you receive the response, please do not hesitate to contact us at any time and we would be delighted to provide you with new links.

All the datasets have been uploaded to the corresponding database and will be made publicly accessible upon the publication of the article. And the corresponding accession IDs have been included in the “Data availability” part of the manuscript as “The raw proteomics and phosphoproteomics data generated during this study have been archived

in the iProX database (<https://www.iprox.cn>) and are available via the following accession IDs: PXD056748 (also available as IPX0009562000) for PCa patients, PXD058635 (also available as IPX0010395000) for PCa cell lines, and PXD058636 (also available as IPX0010400000) for mice models. The raw whole exome sequencing (WES) and RNA-seq data of PCa patients have been deposited in the Genome Sequence Archive (GSA-human, <https://ngdc.cncb.ac.cn/gsa-human>) under accession ID HRA008293; also available under accession number PRJCA029068. The raw single cell RNA sequencing data of mice models are available at the Genome Sequence Archive (GSA, <https://ngdc.cncb.ac.cn/gsa>) under accession ID CRA021269; also available under accession number PRJCA033071. The raw metabolomics data used in this study have been deposited in the OMIX database (<https://ngdc.cncb.ac.cn/omix>) and are available via the following accession IDs: OMIX008183 for PCa patients, and OMIX008185 for PCa cell lines. The processed matrix of patients' RNA-seq was also deposited on OMIX (accession number: OMIX008531). The codes generated in this study are publicly available in GitHub at <https://github.com/Diluczhang/ProstateCancerProteins.git>. The remaining data are available within the Article or Supplementary Information.”

Q3 、 "Data access can be obtained by contacting Dr. Wang [wangzr27@mail.sysu.edu.cn]" To enable anonymous and quick peer review, I kindly ask the authors to include a password for reviewer access in the future.

Response:

Thank you for your thoughtful suggestions. We have provided direct links for each dataset, allowing reviewers to access them without the need for additional passwords. These datasets will be made publicly accessible to all researchers upon the publication of the article. By then, the proteomic, phosphoproteomic, and metabolomic data are publicly available for online viewing and download. The WES and transcriptomic data of PCa patients can be publicly viewed online; but researchers wishing to download these data need to contact the corresponding author via email. This procedure is in accordance with the regulatory guidelines established by the Chinese government for the management of genomic and transcriptomic raw sequencing data involving Chinese populations. Researchers will be granted download access following the approval of their request in accordance with the standard procedure.

Q4、 The authors write that "the codes generated in this study are publicly available in Github at <https://github.com/Diluczhang/Proteome-of-PCA.git>."

Currently, the repository provides only the code (with minimal documentation) for Random Forest, differential protein expression analysis, NTP and for some plots. I could not find the code for other analyzes, incl. the analysis of variants, differential expression with DESeq2, metabolomics, or single-cell data.

Response:

Thank you for your rigorous work and we are sorry for this omission. We have added the corresponding codes in GitHub. Now the codes with corresponding documentation and input data are publicly available in GitHub at <https://github.com/Diluczhang/ProstateCancerProteins.git>, including the analyses of variants, tumor mutation burden (TMB), differential expression with DESeq2, ssGSEA, MCP-counter, NMF-clustering, differential expression of metabolites, QuSAGE analysis for single cell RNA and so on.

Q5 、 Moreover, the code for RF model learning and feature selection found in RadomForecast/radomforest.py appears inconsistent with the analysis description in the manuscript. For example, I could not locate 10-fold CV and it seems that only a simple train-test split is implemented. Since input data used by the authors are not available, I cannot replicate this analysis and test the code directly. I recommend the authors ensure that the code they used and provided in the repository aligns with the analysis described in the manuscript. Moreover, it is recommended to provide the readers with the code and inputs necessary to replicate all essential analysis steps.

Response:

Thank you for your meticulous work and we're sorry to make you confused. Due to our mistake, the code for RF model we uploaded in the manuscript was not the latest version. We re-uploaded the latest version of code for RF model (including 10-fold CV) which we actually used in the analysis of our manuscript. Besides, we rechecked all codes provided in the GitHub and ensured that the codes are consistent with that we used in the analyses, as well as aligns with the descriptions in the manuscript. Moreover, we also added corresponding inputs data and documentation for replications of essential analysis steps. The latest version of codes with corresponding documentation and input data are publicly available in GitHub at <https://github.com/Diluczhang/ProstateCancerProteins.git>

Minor questions and corrections:

Fig. 1A: labels are difficult to read, at least 200% zoom is required.

Fig. 2D: lacks x-axis label.

Fig. S3B: the color bar lacks a label. Does it present averaged protein expressions or z-scores? In which dataset?

Response:

Thank you for your rigorous work. Based on your kind suggestion, we have adjusted these mistakes in Figure 1A, Figure 2D and Figure S3B. In Figure S3B, the color bar presents z-scores and this analysis was performed in the dataset of the discovery cohort.

Figure 1A. Study schematic. The left panel shows the multi-omic experimental design of the discovery cohort. The right panel shows the type of validation experiments, including formalin fixed paraffin-embedded (FFPE) proteome and immunohistochemistry (IHC) in two validation cohorts. LC-MS/MS represents liquid chromatography tandem mass spectrometry. WES represents whole exome sequencing, IF represents immunofluorescence. Created with BioRender.com, released under a Creative Commons Attribution-NonCommercial-NoDerivs 4.0 International license.

Figure 2D. Kinase-substrate enrichment analysis (KSEA) showing significantly up-regulated and down-regulated kinases in subgroup 2 patients of the discovery cohort (subgroup 2 vs subgroup 1 and subgroup 3 combined). Red bars refer to $FDR < 0.05$ and z score > 0 , blue bars refer to $FDR < 0.05$ and z score ≤ 0 .

Figure S3B. The 39-protein panel for proteomic subtyping of PCa filtered by the Random Forest algorithm, SEG represents subgroup enriched genes.

Typos:

"metabonomics"

"Uniformmanifold"

Response:

Thank you for your rigorous work. Based on your kind suggestion, we have adjusted these mistakes in the manuscript.

Reviewer #2 (Remarks to the Author):

This reviewer feels that the authors have addressed the comments provided by the reviewers in the rebuttal. However, it is also this reviewer's opinion that more of the responses from the rebuttal should have been integrated into the manuscript.

Response: Thank you for your insightful feedback and I greatly appreciate the time and effort you invested in the work. Based on your suggestion, we have also integrated the responses (including figure updating and corresponding detailed description) to the manuscript as follows to better improve our manuscript:

- Firstly, we have updated the new IHC staining results in Revised Figure S4D and corresponding description in Results part as **“Besides, IHC staining also confirmed the enhanced immune activity of tumors in the immune subgroup 3 compared to subgroup 1”.** (page 11, line 233-234)

Revised Figure S4D. Immunohistochemistry (IHC) staining and quantification for GZMB, INF- γ and TNF- α in tumor regions of subgroup 1 and subgroup 3 patients. The scale bars represent 500/50um in low/high-power field. Data are presented as mean \pm Standard deviation (SD). P-values are determined using two-tailed Student's t test. (page 11, line 133-136 in Supplementary Information)

- Secondly, we have added the GSEA analysis in the transcriptomic level in Revised Figure S1E and corresponding description in Results part as **“Interestingly, the**

global transcriptomic molecular features of three proteomic subgroups were similar to their proteomic molecular features”. (page 5, line 108-109)

Revised Figure S1E. Gene sets enrichment analysis (GSEA) showing distinct transcriptomic molecular characteristics among three proteomic subgroups of the discovery cohort. (page 7, line 71-71 in Supplementary Information)

- Thirdly, we have added the Western blot results in Revised Figure S4A and corresponding description in Results part as “Besides, the western blot results confirmed the higher expression of NANS in subgroup 2 cell lines of PCa”. (page 10, line 214-215)

Revised Figure S4A. Comparison of NANS protein expression among 8 PCa cell lines by Western blot. (page 11, line 127 in Supplementary Information)

Reviewer #4 (Remarks to the Author):

I sincerely appreciate the authors' efforts in revising the manuscript and for providing links to the data and up-to-date code used in their study. In response to Q3, the authors state: "We have provided direct links for each dataset, allowing reviewers to access them without the need for additional passwords. These datasets will be made publicly accessible to all researchers upon the publication of the article. By then, the proteomic, phosphoproteomic, and metabolomic data are publicly available for online viewing and download". However, I was unable to find download links for the following datasets:

** PCa patients, raw metabolomics data:
<https://ngdc.cncb.ac.cn/omix/preview/7s2OX3fe>*

** RNA-seq matrix: <https://ngdc.cncb.ac.cn/omix/preview/xKwuJZE7>*

** PCa cell lines, raw metabolomics data:
<https://ngdc.cncb.ac.cn/omix/preview/J6fxWY6S>*

I would appreciate it if the authors could confirm that all datasets will be publicly accessible upon the publication of the article.

Response:

We appreciate the reviewer's kind evaluation of our work. Regarding the issue about the inability to download datasets from the OMIX database, we have re-examined the upload status of these datasets and confirmed that they have been fully uploaded to the OMIX platform (see the attached screenshots below for details). And we also identified that due to inherent limitations of the OMIX database, the preview links generated for reviewers do not include download permissions. Depending on the rules of OMIX platform, the download permissions will be publicly available after the official publication of the article because of data privacy and supervision. We would like to clarify that datasets currently marked as "Open-access" on the OMIX platform are indeed publicly accessible to all researchers (see the screenshots below). We have verified that the permissions for our datasets have already been set to "Open-access". Therefore, we confirm that once the article is officially published, these datasets will be publicly accessible on the OMIX platform. And before that, we provide preview links for reviewers and editors to check the uploaded status and detailed information of these raw data:

Datasets details and upload status from OMIX platform:

① PCa patients, raw metabolomics data

(OMIX008183, <https://ngdc.cncb.ac.cn/omix/preview/7s2OX3fe>)

OMIX008183 - Files

Files & Download

The backend will take some time to ensure the integrity of each file by checking MD5 hash values.
The time increases linearly with file size. Thanks for your patience.

File ID	File Title	Number/Samples	File Type	File Size	File Suffix	Download Times	Status	Operation
OMIX008183-01	HM650_HILIC_mzML	27	Metabolome Data by Mass Spectrometry (MS)	16.9 MB	zip	0	HTTP Archived	
OMIX008183-02	HM650_C18_mzML	27	Metabolome Data by Mass Spectrometry (MS)	11.3 MB	zip	0	HTTP Archived	

② RNA-seq matrix (OMIX008531, <https://ngdc.cncb.ac.cn/omix/preview/vCA3VMVp>)

OMIX008531 - Files

Files & Download

The backend will take some time to ensure the integrity of each file by checking MD5 hash values.
The time increases linearly with file size. Thanks for your patience.

File ID	File Title	Number/Samples	File Type	File Size	File Suffix	Download Times	Status	Operation
OMIX008531-01	RNA_fpkm	1	Expression Profiling by NGS	10.0 MB	zip	0	HTTP Archived	

③ PCa cell lines, raw metabolomics data
(OMIX008185, <https://ngdc.cncb.ac.cn/omix/preview/J6fxWY6S>)

OMIX008185 - Files

Files & Download

The backend will take some time to ensure the integrity of each file by checking MD5 hash values.
The time increases linearly with file size. Thanks for your patience.

File ID	File Title	Number/Samples	File Type	File Size	File Suffix	Download Times	Status	Operation
OMIX008185-01	Myccap_HM650_HILIC_mzML	20	Metabolome Data by Mass Spectrometry (MS)	10.8 MB	zip	0	HTTP Archived	
OMIX008185-02	Myccap_HM650_C18_mzML	20	Metabolome Data by Mass Spectrometry (MS)	8.6 MB	zip	0	HTTP Archived	
OMIX008185-03	22Rv1_HM650_HILIC_mzML	20	Metabolome Data by Mass Spectrometry (MS)	10.8 MB	zip	0	HTTP Archived	
OMIX008185-04	22Rv1_HM650_C18_mzML	20	Metabolome Data by Mass Spectrometry (MS)	8.6 MB	zip	0	HTTP Archived	

Download links of “Open-access” datasets on OMIX platform after the article published:

OMIX001130

Summary

Title	Mechanism of intercellular protein homeostasis regulation
Description	Unfolded protein response (UPR) was activated for maintain protein homeostasis in cellular level. However, how UPR function across multiple cells and tissues in the organism is largely unknown. Here we propose to study the molecular mechanism of cell non-autonomous UPR between mammalian cell. We anticipate a better understanding of the protein homeostasis at the multicellular level, and to provide new therapeutic strategies for related diseases such as cancer, obesity and fatty liver disease.
Organism	Geospiza fortis
Data Type	Clinical information
Data Accessibility	Open-access
BioProject	PRJCA009523
Release Date	2025-02-15
Submitter	Yazhen Huo (yazhenhuo@ibp.ac.cn)
Organization	Institute of Biophysics, Chinese Academy of Sciences
Submission Date	2022-05-11

Files & Download

HTTP download speed may be slow. It is highly recommended that you download the dataset using a dedicated FTP tool (such as FileZilla Client).

File ID	File Title	Number/Samples	File Type	File Size	File Suffix	Download Times	Download
OMIX001130-40-02	Ctrl FT mice serum lipidomics	1	Clinical information	35.6 KB	.xlsx	0	FTP HTTPS
OMIX001130-40-03	WT OB mice serum lipidomics	1	Clinical information	25.1 KB	.xlsx	0	FTP HTTPS
OMIX001130-04	Ctrl CM lipidomics	1	Clinical information	49.5 KB	.xlsx	0	FTP HTTPS
OMIX001130-05	3T3-L1 AML12 cell line lipidomics	1	Clinical information	56.2 KB	.xlsx	0	FTP HTTPS